# MultiGATE: integrative analysis and regulatory inference in spatial multi-omics data via graph representation learning

Jishuai Miao [1,7], Jinzhao Li[1,7], Jingxue Xin[1], Jiajuan Tu[2], Muyang Ge[1], Ji Qi[1], Xiaocheng Zhou [1], Ying Zhu [3], Can Yang [4,5] ✉ & Zhixiang Lin [1,6] ✉

New spatial multi-omics technologies, which jointly profile transcriptome and epigenome/protein markers for the same tissue section, expand the frontiers of spatial techniques. Here, we introduce MultiGATE, which utilizes a two-level graph attention auto-encoder to integrate the multi-modality and spatial information in spatial multi-omics data. The key feature of MultiGATE is that it simultaneously performs embedding of the spatial pixels and infers the cross-modality regulatory relationship, which allows deeper data integration and provides insights on transcriptional regulation. We evaluate the performance of MultiGATE on spatial multi-omics datasets obtained from different tissues and platforms. Through effectively integrating spatial multi-omics data, MultiGATE both enhances the extraction of latent embeddings of the pixels and boosts the inference of transcriptional regulation for cross-modality genomic features.

In recent years, the advent of spatial transcriptomic technologies has revolutionized our understanding of gene expression patterns within complex tissues and organs. These techniques profile gene expression levels in their native spatial context, offering rich insights into the organization and heterogeneity of biological tissues. New spatial multi-omics technologies, which jointly profile spatial transcriptome and epigenome/protein markers for the same tissue section, have expanded the frontiers of spatial techniques: spatial ATAC–RNA–seq and spatial CUT&Tag-RNA-seq[1] jointly profile spatial transcriptome and epigenome; Slide-tags[2] jointly profiles open chromatin, RNA, and T-cell receptor sequences in single nuclei, together with the spatial barcode of the nuclei; Spatial Protein and Transcriptome Sequencing (SPOTS)[3] is a technology that jointly profiles RNA and protein markers. Compared to single-omics technologies, spatial multi-omics provides additional molecular modality for more comprehensive profiling of the tissue and allows a deeper understanding of the regulatory relationship across different molecular modalities in the central dogma of molecular biology, within the native spatial context of the tissue.

Methods have been developed for spatial transcriptomic data, including SpaGCN[4], STAGATE[5], and SpatialPCA[6]. These methods, utilizing either graph neural networks or spatially-aware matrix factorization, have achieved precise spatial clustering of the pixels, thereby facilitating our understanding of spatial gene expression patterns. However, there is a pressing need to utilize the information embedded within additional omics layers in spatial multi-omics data. Methods have been developed for single-cell multi-omics data[7-12]. Among them, Seurat WNN[7] provides an unsupervised framework to learn the relative importance of each molecular modality in each cell, enabling integrative analysis of single-cell multi-omics data. However, it was not designed for spatial multi-omics and does not account for the spatial information, which is crucial in the study of spatial data. MOFA+[8] is

[1]Department of Statistics and Data Science, The Chinese University of Hong Kong, Hong Kong SAR, China. [2]School of Science, Hubei University of Technology, Wuhan, China. [3]State Key Laboratory of Brain Function and Disorders, MOE Frontiers Center for Brain Science, Institutes of Brain Science and Department of Neurosurgery, Huashan Hospital, Fudan University, Shanghai, China. [4]Department of Mathematics, The Hong Kong University of Science and Technology, Hong Kong SAR, China. [5]State Key Laboratory of Nervous System Disorders, The Hong Kong University of Science and Technology, Hong Kong SAR, China. [6]CUHK Shenzhen Research Institute, Shenzhen, China. [7]These authors contributed equally: Jishuai Miao, Jinzhao Li. ✉e-mail: macyang@ust.hk; zhixianglin@cuhk.edu.hk

designed for the integration of single-cell multi-modal data, using a linear factor model that decomposes the input matrices into the product of low-rank matrices (weight matrices and low-dimensional representation matrices); this linear factor model does not consider spatial information and the cross-modality regulatory relationship. totalVI[9] is designed for CITE-seq data (RNA + surface protein); it uses the variational autoencoder (VAE) framework to model gene expression raw counts with negative-binomial (NB) distribution and the protein counts as an NB mixture of foreground and background signal; the VAE framework in totalVI does not incorporate spatial information in the spatial multi-omics data. MultiVI[10] is based on a conditional VAE and models each modality using a specific distribution: NB distribution for gene expression raw counts and Bernoulli distribution for chromatin accessibility; this conditional VAE does not consider spatial information and the cross-modality regulatory relationship. JSNMF[11] employs a joint semi-orthogonal nonnegative matrix factorization (NMF) model to learn separate latent factor matrices for each modality and then fuses them via a consensus cell-cell similarity graph, and the NMF model does not incorporate the spatial information. In contrast, a more recent method, SpatialGlue[13], performs an integrated analysis of spatial multi-omics data using a graph convolution network to obtain embeddings for each molecular modality separately, and then employs attention mechanisms to integrate information from different modalities. However, SpatialGlue uses principal components as the input and does not model the cross-modality regulation in the embedding of pixels. To fully utilize the information in spatial multi-omics data and realize its potential in uncovering the regulatory relationship across different molecular modalities, it is crucial to model the connection of the cross-modality features.

In response to the critical need for integrative tools in spatial multi-omics, we have developed MultiGATE, and its core is a two-level graph attention autoencoder designed to (i) infer cross-modality regulatory relationships (including cis-regulation, trans-regulation and protein–gene interactions) and (ii) embed spatial pixels into a latent space for clustering/spatial domain identification and visualization. By modeling regulatory links (such as peak–gene associations, protein–gene interactions, and enzyme–metabolite associations) directly into its graph attention mechanism, MultiGATE learns more informative low-dimensional representations of each spatial pixel/spot by deeper integrating different modalities. In turn, these refined embeddings sharpen the attention scores between cross-modality features, yielding more accurate inference of cross-modality regulation. The analysis of the cross-modality regulatory relationship enabled by MultiGATE provides unique insight in studying transcriptional regulation in the native tissue context powered by spatial multi-omics data (Detailed methodological innovations of MultiGATE are described in Supplementary Note S1). We demonstrate the superior performance of MultiGATE through various spatial multi-omics datasets generated from different tissues and technologies. Our results highlight the ability of MultiGATE to effectively capture the regulatory relationships across different molecular modalities while providing enhanced latent embeddings of the pixels for accurate spatial clustering.

## Results

### Method overview
The core of MultiGATE is a two-level graph attention autoencoder that simultaneously embeds the pixels/spots in a low-dimensional space and models the cross-modality feature regulatory relationships (e.g., peak–gene, protein–gene, and metabolite–gene). At the first level, a cross-modality attention mechanism is employed to model the cross-modality regulatory relationship; at the second level, a within-modality attention mechanism is utilized to incorporate spatial information, which encourages the embedding of neighboring pixels to be similar. To further improve the cross-modality data integration, MultiGATE

also incorporates a Contrastive Language-Image Pretraining (CLIP) loss, which encourages the alignment of the embeddings from different modalities.

The holistic structure of MultiGATE enables the aggregation and alignment of spatial information and molecular features from each modality. MultiGATE effectively extracts the low-dimensional representations of the pixels for clustering/spatial domain identification and data visualization, and simultaneously unravels the regulatory relationships of the cross-modality features for cis-regulation, trans-regulation, and protein-gene interactions (Fig. 1).

### MultiGATE unveils the functional landscape of the adult human hippocampus
We first employed MultiGATE to analyze the Spatial ATAC–RNA–seq dataset[1] generated from the adult human brain hippocampus, where chromatin accessibility and gene expression are simultaneously profiled at different spatial pixels. We manually annotated the hippocampus layers and white matter (Fig. 2a) based on guidance from previous studies[14–16] and marker genes. These annotations served as the ground truth for evaluating the performance of different methods. We calculated the Adjusted Rand Index (ARI) for each method and compared their clustering accuracy. The results were obtained using pixels that underwent preprocessing in SpatialGlue[13], which filters out low-quality pixels, and we removed the pixels that were filtered out by SpatialGlue when computing the ARI for each method to ensure a fair comparison. Of all the methods evaluated, MultiGATE achieved the highest accuracy in detecting the layer structure within the human hippocampus (ARI: MultiGATE 0.60, SpatialGlue 0.36, Seurat WNN 0.23, MOFA+ 0.10, and MultiVI 0.14; Fig. 2b and Supplementary Fig. 1A) and superior performance in other quantitative metrics (Supplementary Fig. 2A). Upon visually inspecting the clustering results in Fig. 2b (A unified color scheme is applied across methods using label matching, as detailed in Supplementary Note S2), MultiGATE provided a clearer deciphering of the molecular layer (ML) and the choroid plexus compared to SpatialGlue. On the other hand, Seurat WNN does not incorporate the spatial information, resulting in a clustering outcome that lacks spatial organization.

MultiGATE simultaneously models the peak–gene association in embedding the pixels in a low-dimensional space, distinguishing itself from SpatialGlue[13] and Seurat WNN[7], which rely on principal components as input and do not model peak–gene associations. By incorporating a graph-based representation of regulatory interactions, MultiGATE employs a Bayesian-like approach that combines prior knowledge of genomic distance with the observed spatial multi-omics data to estimate cross-modality attention, allowing for modeling spatial information and the multi-omics genomic features in the attention mechanism simultaneously ("Methods").

To assess whether the attention mechanism in MultiGATE can accurately capture the cis-regulatory interactions, we compared the attention score estimated by MultiGATE with external eQTL data[17], which has been widely used to identify genetic variants that influence gene regulation[18]. For the human hippocampus data, the attention scores estimated by MultiGATE decrease as genomic distance increases, whereas the peak–gene pairs supported by eQTL data from the human hippocampus demonstrate higher attention scores, compared to those that are not supported (Fig. 2c). This suggests that MultiGATE effectively captures genuine cis-regulatory interactions. Next, we used the human hippocampus eQTL data[17] as the benchmark, and compared the performance for identifying peak–gene association between MultiGATE and other methods (Fig. 2d): MultiGATE achieved the best AUROC score (0.703), compared to Cicero[19] (AUROC = 0.530), Spearman correlation (AUROC = 0.515), and LASSO regression (AUROC = 0.501). To further showcase the biological relevance of the peak–gene interactions identified by MultiGATE, we examined genes with well-established functional importance in the hippocampus, including *CA12*

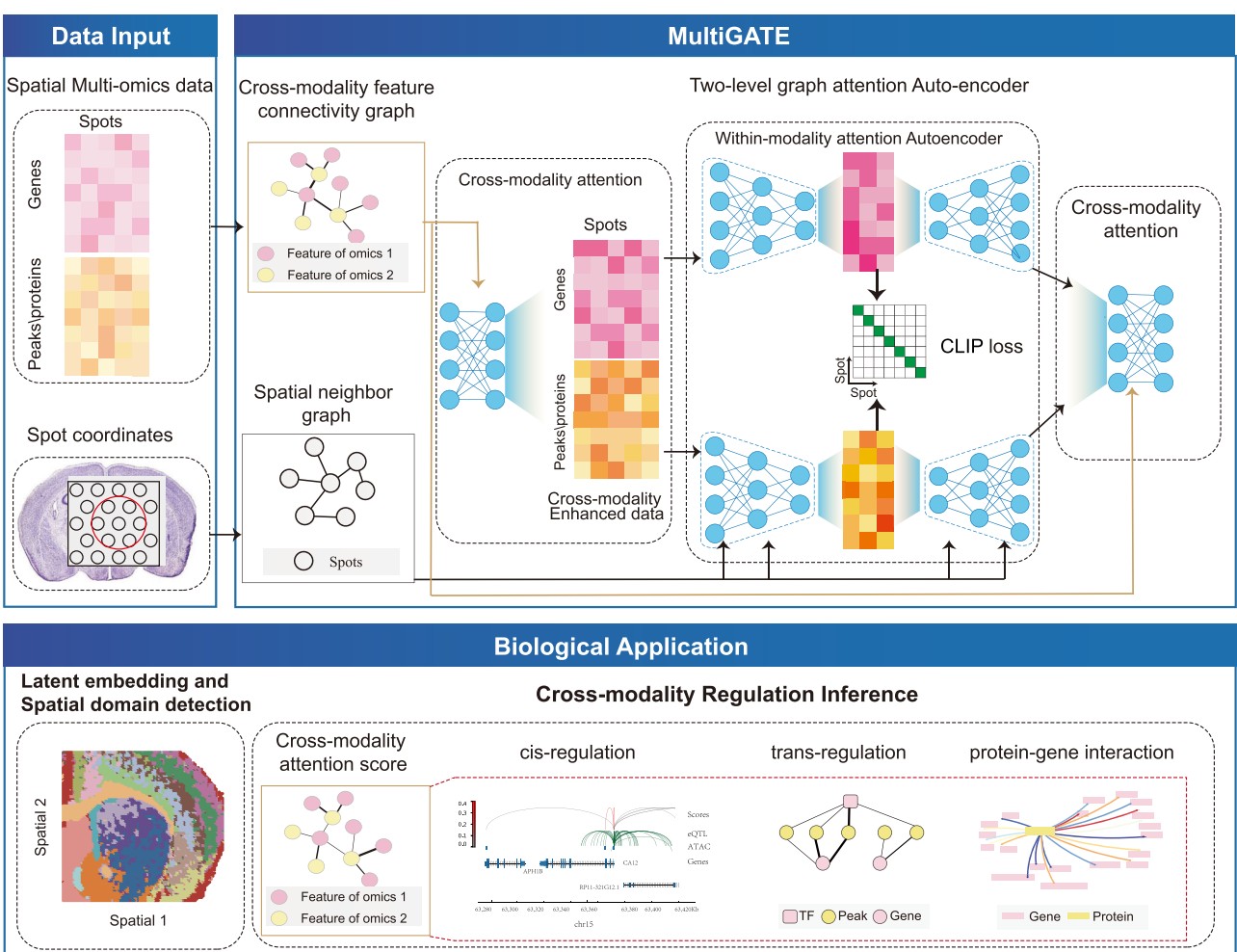

**Fig. 1 | Overview of MultiGATE.** MultiGATE is a two-level graph attention auto-encoder designed for spatial multi-omics analysis. It extracts the latent embeddings of the pixels/spots in spatial multi-omics data, while simultaneously incorporating the regulatory relationship of the cross-modality features through the cross-modality attention mechanism and the spatial relationship of the pixels/spots through the within-modality attention mechanism. In addition to reconstruction loss, a CLIP contrastive loss aligns embeddings across modalities. MultiGATE yields (i) latent representations of pixels for clustering and visualization and (ii) cross-modality attention scores for cross-modality regulatory inference.

and *PRKD3*. *CA12* is a carbonic anhydrase in hippocampal neurons, regulates pH, and supports neuronal excitability[20,21]. *PRKD3* is associated with synaptic function and Alzheimer's disease[22,23]. For each of these genes, MultiGATE successfully identified peak–gene associations that are supported by hippocampus-specific eQTLs (Fig. 2e), whereas non-significant associations are generally not supported by eQTL signals. Moreover, MultiGATE can be extended to detect long-range interactions between enhancers and promoters: across eight distance bins (0–150 kb, 150–300 kb, …, >1.25 Mb), MultiGATE can distinguish the brain-specific enhancer-promoter contacts from those in other tissues across 24 diverse human tissues (Supplementary Note S3). In addition, MultiGATE has been extended to capture trans-regulatory interactions by incorporating TF binding priors and learning TF–peak–gene attention scores. Using ChIP-seq data for SOX2 in brain tissue as ground truth, our method outperformed motif-only, TF binding potential, and cosine similarity baselines in predicting SOX2 binding, achieving an AUC of 0.8669 and AUPR of 0.4906 (Supplementary Note S4). In summary, MultiGATE enables regulatory inference and enhances our understanding of cis-regulatory and trans-regulatory interactions by effectively mining multi-omics spatial data to precisely infer regulatory networks.

Finally, to explore the biological relevance of the identified clusters, we examined differentially expressed genes (DEGs) for each cluster identified by MultiGATE in Supplementary Fig. 3B. For instance, *PPFIA2* displayed differential expression and showed high expression levels in the granule cell layer, aligning with previous findings[24]. Known molecular markers of the hippocampus, such as *SLC1A2*, were specifically expressed in the ML[14] identified by MultiGATE. Furthermore, *PLP1* encodes a major component of myelin protein in the central nervous system[25]. And it is highly expressed in the stratum lacunosum-moleculare region identified by MultiGATE. Gene Ontology (GO) analysis of the DEGs from this region revealed significant enrichment for myelination (Supplementary Fig. 3C), aligning with the known function of *PLP1*. These findings underscore the biological relevance and accuracy of the clusters identified by MultiGATE.

## MultiGATE reveals a layered pattern that mimics the annotated structure in the mouse brain while enabling regulatory inference

We further employed MultiGATE to analyze the spatial ATAC–RNA-seq dataset[1] generated from mouse postnatal day 22 (P22) brains. We utilized a P56 coronal annotation provided by the Allen Mouse Brain Atlas as a reference in Fig. 3a.

We aligned the clustering results at the profiled region within the whole tissue slide in Fig. 3b to enhance clarity. To quantitatively evaluate clustering results, we computed the Intraclass Correlation

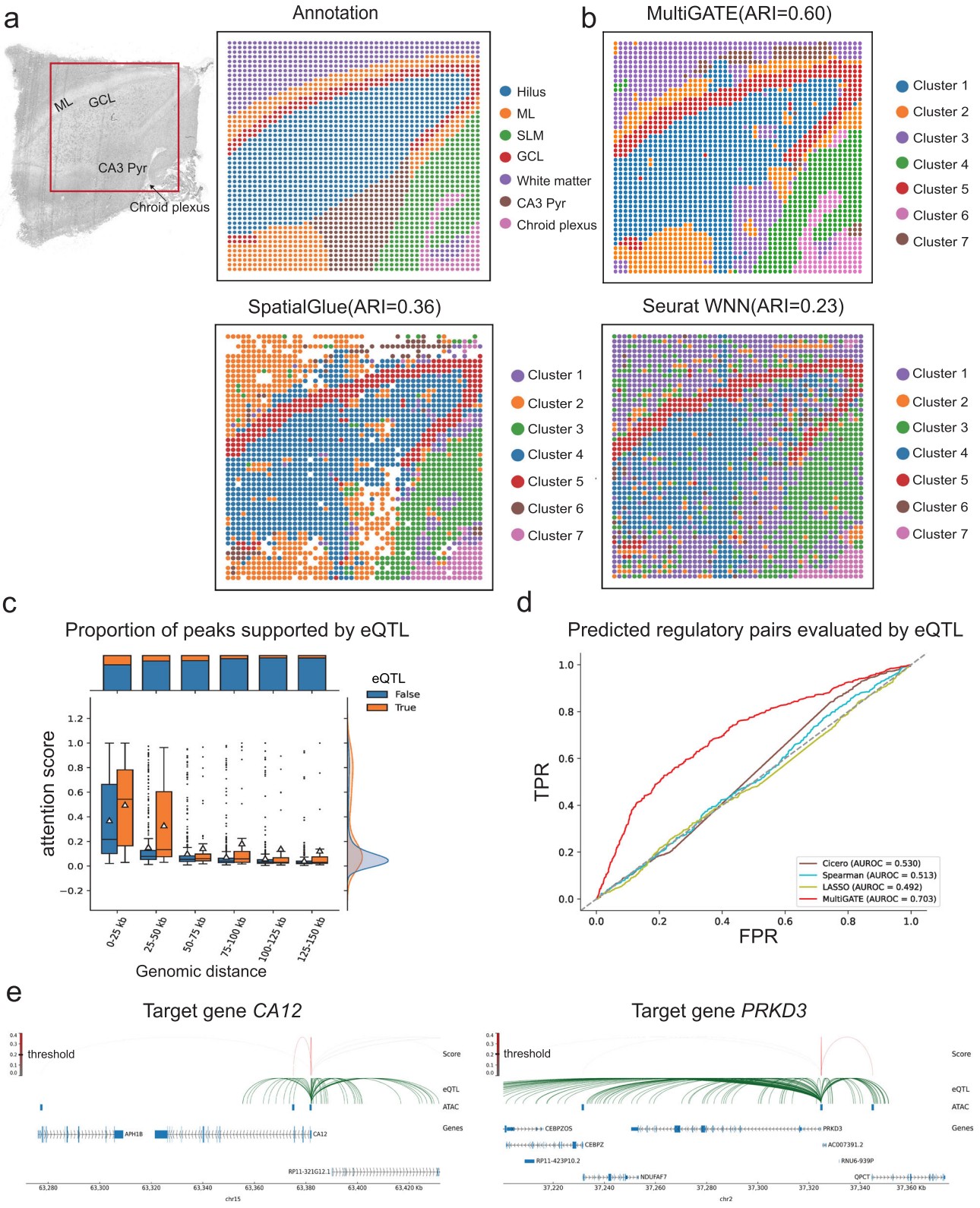

Coefficient (ICC)[26]. The ICC quantifies the consistency of observations within each cluster. Higher ICC values reflect greater intra-cluster homogeneity for a given modality, indicating improved clustering accuracy[27]. MultiGATE achieves the highest combined ICC score among all methods compared, outperforming SpatialGlue, Seurat WNN, MOFA+, and MultiVI (Supplementary Figs. 2C, 1B, and 4A). Both MultiGATE and SpatialGlue exhibit the ability to detect the genu of corpus callosum (ccg), lateral ventricle (VL), caudoputamen (CP),

olfactory limb (aco), and lateral preoptic area, as well as accurately identifying the six layers present in the cortex. MultiGATE more accurately discriminates the outermost cortical layer (Cluster 5, shown in red). In contrast, Cluster 5 in SpatialGlue shows no cluster-enriched markers (Supplementary Note S5), indicating a heterogeneous mixture of multiple cell types. More specifically, Cluster 5 in SpatialGlue contains pixels in the LS nucleus region[1], which are molecularly different from the other pixels in Cluster 5 (Fig. 3b and Supplementary Note S5).

**Fig. 2 | MultiGATE enables precise detection of spatial domains and authentic integrative regulatory inference in the human hippocampus. a** Bright-field image and manually annotated segmentation of hippocampus layers and white matter (WM) in the human hippocampus. **b** Spatial clustering of hippocampal regions using MultiGATE, SpatialGlue, and Seurat WNN. Clustering performance is assessed using the Adjusted Rand Index (ARI), with higher values indicating greater clustering accuracy. **c** Box plots representing attention scores for peak–gene pairs across different genomic distances, grouped based on whether they are supported by expression quantitative trait loci (eQTL) evidence. The box plots indicate the medians (centerlines), means (triangles), first and third quartiles (bounds of boxes), and 1.5 × interquartile range (whiskers). Sample sizes per bin (False/True): 0–25 kb (621/222), 25–50 kb (479/88), 50–75 kb (461/78), 75–100 kb (469/44), 100–125 kb (446/30), 125–150 kb (405/29). **d** Receiver operating characteristic (ROC) curves comparing the performance of MultiGATE and other methods in predicting eQTL-associated regulatory interactions. **e** Visualization of MultiGATE-predicted cis-regulatory interactions for the target genes *CA12* and *PRKD3* along with eQTL evidence. Source data are provided as a Source Data file.

This highlights the enhanced precision and reliability offered by MultiGATE in cortical layer analysis. In contrast, Seurat WNN does not consider the spatial information and the clustering result does not have a clear spatial organization of the regions (Fig. 3b).

To validate MultiGATE's attention mechanism in inferring peak–gene regulation, we utilized two validation datasets: EnhancerAtlas[28] and EGAS[29] for enhancer-gene regulation. In Fig. 3c, d, the attention scores estimated by MultiGATE decrease as genomic distances increase; Moreover, higher attention scores were observed for the peak–gene pairs where the peak overlaps with an enhancer that regulates the corresponding gene. These results are attributed to the adaptive learning of attention, which incorporates prior weights that depend on genomic distance and the observed multi-omics data. Next, we compared the performance on identifying peak–gene regulatory pairs between MultiGATE and other methods. MultiGATE outperformed Cicero[19], Spearman correlation and lasso regression, as indicated by the highest AUROC scores in Fig. 3c, d. We next demonstrate the regulatory relationship for the gene *Xrcc5*. The gene *Xrcc5* is known for its involvement in DNA repair mechanisms in the mouse brain[30]. MultiGATE identified a peak that regulates *Xrcc5* (indicated by the red curve) and this peak is about 90 kb away from the gene body of *Xrcc5*; This peak also overlaps with an enhancer that regulates *Xrcc5* in the Enhancer Atlas database; The other peaks near *Xrcc5* are not predicted to regulate *Xrcc5* in MultiGATE and neither of these peaks is supported in the Enhancer Atlas dataset (Fig. 3e). This observation and the overall distribution of attention scores in Fig. 3c, d demonstrate the superior performance of MultiGATE in regulatory inference.

To demonstrate that MultiGATE effectively combines the information in multiple modalities, we also inspected the clustering result for each molecular modality separately[5]. There are substantial differences between the clustering outcomes of the two individual modalities: spatial clusters generated by ATAC-seq provide more informative patterns regarding contextual layering and internal structure (Supplementary Fig. 3D); RNA-seq also contributes valuable information, especially the identification of the outermost layer (Supplementary Fig. 3D). MultiGATE integrates both modalities to produce more accurate and comprehensive clustering results. By incorporating multi-modal information, MultiGATE capitalizes on the strength of ATAC-seq and RNA-seq, leading to enhanced performance in clustering analysis (Fig. 3b).

Furthermore, we have successfully identified marker genes associated with the spatial clusters identified by MultiGATE (Supplementary Note S6), including marker genes for the regions ccg, CP, and VL, as shown in Fig. 3f. *Pde10a* is highly expressed in CP, which is part of the striatum dorsal region, consistent with the previous findings[31]. *Hmgb2* was shown to be expressed in neural stem/progenitor cells, which are highly active in the VL zone and contribute to the balance between neurogenesis and gliogenesis[32], consistent with the observed high expression of *Hmgb2* in the VL region in the spatial multi-omics data (Fig. 3f). *Top2a* is highly expressed in the VL region as shown in Fig. 3f, which supports previous findings that *Top2a* is associated with mitosis and cell cycle regulation and is highly expressed in the VL region[33]. *Nr4a2* is highly expressed in the EPd region (Fig. 3f), consistent with previous findings that *Nr4a2*-expressing cells migrate to

form the dorsal endopiriform nucleus[34]. *Fth1* is highly expressed in the corpus callosum, particularly in oligodendrocytes, where it plays a critical role in neuroprotection and the maintenance of axonal health[35], which aligns well with our observation that *Fth1* is highly expressed in the CP region.

These expression patterns and the above regulatory inference derived from spatial analysis contribute to a more profound understanding of the diverse regions within the brain.

## MultiGATE reveals spatial patterns in diverse spatial multi-omics technologies

SPOTS[3] is a spatial multi-omics technology that jointly profiles whole transcriptome and protein markers while preserving tissue architecture. We first analyzed a SPOTS dataset[3] generated from murine spleen, which consists of distinct cellular populations organized into well-defined structures such as germinal centers (GCs), marginal zones (MZs), and red pulp (The bright-field image is shown in Fig. 4a). The clustering results of MultiGATE, SpatialGlue, and Seurat WNN are shown in Fig. 4b. MultiGATE achieves the highest combined ICC score among the five methods, outperforming SpatialGlue, Seurat WNN, MOFA+, and totalVI (Supplementary Figs. 2D, 7C, and 8B). MultiGATE identified five clusters: T cell, B cell, and three distinct macrophage subtypes. Spatially, the T cells (purple) are located at the center, surrounded by B cells (green), and further encircled by successive layers of White Pulp Macrophages (WPM), Marginal Zone Macrophages (MZM), and Red Pulp Macrophages (RPM) (Supplementary Note S7 and Supplementary Fig. 5A, B). This anatomically consistent layering reflects spleen structure and highlights MultiGATE's ability to distinguish spatially organized immune populations.

Notably, MultiGATE demonstrates a more precise clustering of T cells and B cells compared to SpatialGlue. We evaluated this by comparing the ADT expression of the CD3 protein, a canonical T cell marker, across T cell and B cell clusters identified by each method (Fig. 4c). Since CD3 is not expressed in B cells, a method that better separates these populations should yield a greater difference in CD3 expression between the two clusters. Indeed, the clustering of MultiGATE shows significantly greater separation in CD3 expression (Wilcoxon $P = 1 \times 10^{-6}$, rank-biserial $r = -0.953$, and Cliff's $\delta = 0.953$) than SpatialGlue (Wilcoxon $P = 1 \times 10^{-6}$, rank-biserial $r = -0.866$, and Cliff's $\delta = 0.866$; Supplementary Note S8). Additionally, within the T cell cluster itself, CD3 expression is significantly higher in the clustering produced by MultiGATE than SpatialGlue (Wilcoxon $P = 1.2 \times 10^{-5}$; Supplementary Note S8), suggesting more precise identification of T cells. To visually compare clustering performance, we examined three annotated regions in the bright-field image (Fig. 4a). The T cell clusters (purple pixels) identified by MultiGATE show stronger concordance with CD3 protein expression than those identified by SpatialGlue (Fig. 4d). Furthermore, SpatialGlue identifies a significantly higher number of B cells (outliers in Fig. 4c) with elevated CD3 expression–likely T cells misclassified as B cells. In regions 2 and 3 (Fig. 4d), the highlighted pixels show high CD3 expression and low CD19 expression, indicating that these pixels should be categorized as

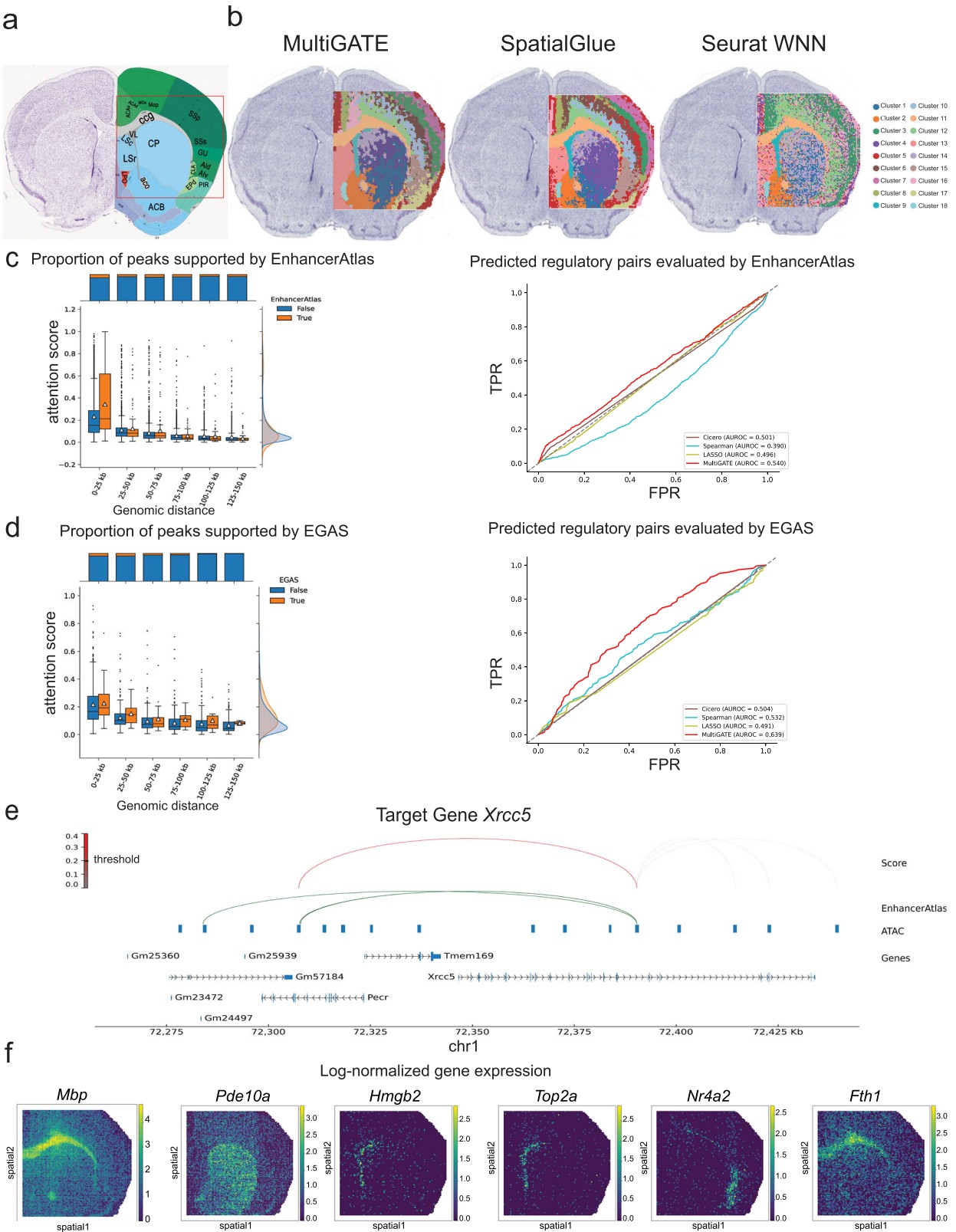

a

b    MultiGATE          SpatialGlue          Seurat WNN

c    Proportion of peaks supported by EnhancerAtlas

Predicted regulatory pairs evaluated by EnhancerAtlas

d    Proportion of peaks supported by EGAS

Predicted regulatory pairs evaluated by EGAS

e    Target Gene *Xrcc5*

f    Log-normalized gene expression

*Mbp*    *Pde10a*    *Hmgb2*    *Top2a*    *Nr4a2*    *Fth1*

T cells, which MultiGATE correctly identifies. In summary, these results suggest that MultiGATE achieves a more precise separation of T cells and B cells. Its spatial clustering result also aligns well with the protein-based deconvolution (Supplementary Note S9). In Fig. 4e, the T cell cluster shows strong CD3, CD4, and CD8 expression, consistent with known T cell profiles: CD8 for cytotoxic T lymphocytes[36], and CD3/CD4 crucial for anti-tumor responses[37]. Additionally, CD169-Siglec is

highly expressed in the MZM cluster (Fig. 4e), aligning with its known function as an I-type lectin that recognizes gangliosides and captures viruses[38].

We also implemented Seurat WNN on this dataset (Fig. 4b). Even at the lowest resolution of 0.1 for Louvain clustering, Seurat WNN identified seven clusters, roughly capturing the T cell (Cluster 4) and B cell (Cluster 3) populations. However, because it

**Fig. 3 | MultiGATE enhances structural identification in the P22 mouse brain. a** Annotated coronal section of a P56 mouse brain from the Allen Mouse Brain Atlas. **b** Spatial clustering of brain regions in a P22 mouse brain using MultiGATE, SpatialGlue, and Seurat WNN, displaying distinct regional segmentation by each method. **c** Box plots showing attention scores for peak–gene pairs across genomic distances, grouped by whether they are supported by EnhancerAtlas, a multi-species database linking enhancers to target genes. The ROC curves compare the predictive performance of MultiGATE and other methods in identifying enhancer-gene interactions supported by EnhancerAtlas. The box plots indicate the medians (centerlines), means (triangles), first and third quartiles (bounds of boxes), and 1.5 × interquartile range (whiskers). Sample sizes per bin (False/True): 0–25 kb (1714/236), 25–50 kb (1735/161), 50–75 kb (1509/167), 75–100 kb (1344/132),

100–125 kb (1265/101), 125–150 kb (1075/93). **d** Similar analysis as in (**c**) for peak–gene pairs evaluated by EGAS, an enhancer-gene association study employing a permutation-based approach to minimize false positives, validated through chromatin conformation and enhancer perturbation experiments. The box plots indicate the medians (centerlines), means (triangles), first and third quartiles (bounds of boxes), and 1.5 × interquartile range (whiskers). Sample sizes per bin (False/True): 0–25 kb (380/39), 25–50 kb (342/35), 50–75 kb (324/21), 75–100 kb (304/17), 100–125 kb (276/6), 125–150 kb (223/2). **e** Visualization of cis-regulatory interactions predicted by MultiGATE for the target gene *Xrcc5*, supported by evidence from EnhancerAtlas. **f** Spatial expression patterns of differentially expressed genes (DEGs) in specific brain clusters of the P22 mouse brain. Source data are provided as a Source Data file.

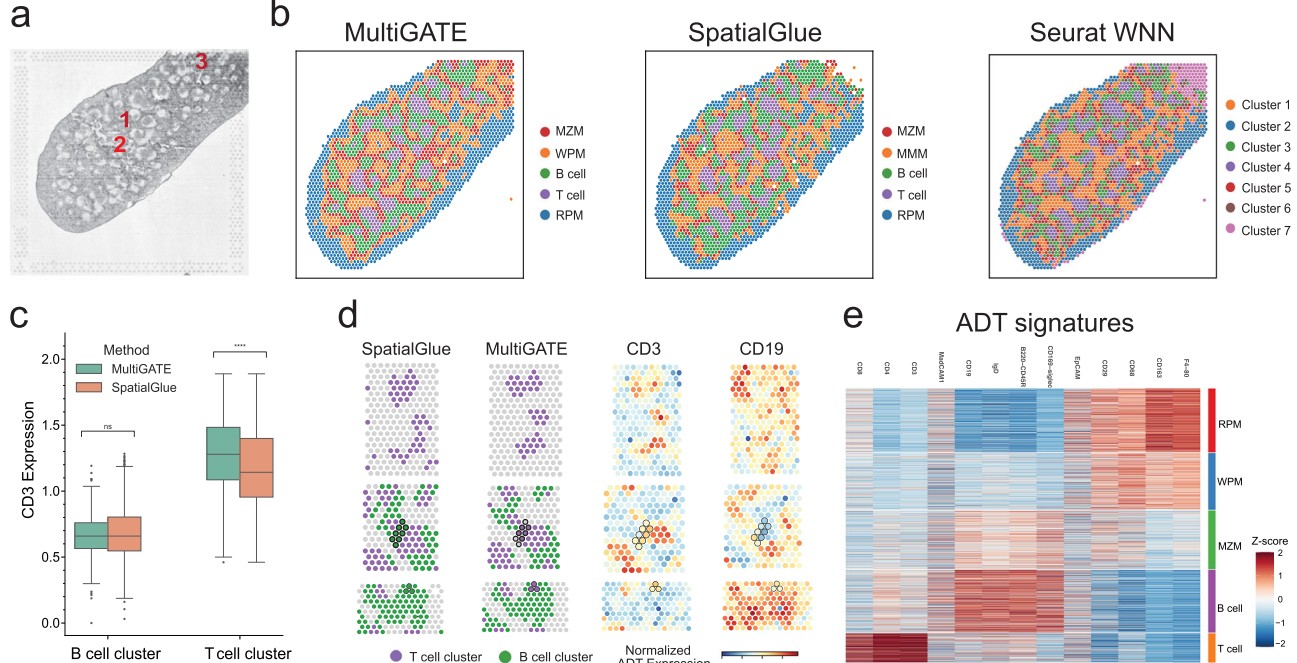

**Fig. 4 | MultiGATE discerns clear patterns in the SPOTS dataset. a** Histology image of the mouse spleen replicate 1 sample, highlighting regions corresponding to magnified comparison region in (**d**). **b** Spatial clustering of immune cells in the mouse spleen as identified by MultiGATE, SpatialGlue, and Seurat WNN. **c** Box plots showing CD3 expression levels in T cells and B cells across clusters identified by MultiGATE and SpatialGlue. Box plots display the median (centerlines), interquartile range (box bounds), and 1.5 × interquartile range (whiskers). *n* (MultiGATE/ SpatialGlue): T cells 259/360; B cells 579/705. Statistical significance was assessed

using two-sided Mann–Whitney–Wilcoxon tests. Exact *p* values: MultiGATE vs. SpatialGlue in B cell cluster: *P* = 2.81 × 10⁻¹; MultiGATE vs. SpatialGlue in T cell cluster: *P* = 1.22 × 10⁻⁵. No correction for multiple comparisons was applied. Significance: ns: *P* > 0.05; *P* ≤ 0.05; **P* ≤ 0.01; ***P* ≤ 0.001; ****P* ≤ 0.0001. **d** Magnified comparison of CD3 and CD19 expression in regions identified by MultiGATE and SpatialGlue. **e** ADT (Antibody-Derived Tags) signatures associated with each spatial cluster, revealing distinct immune cell identities. Source data are provided as a Source Data file.

does not consider the spatial information, its overall clustering result lacks the layered anatomical pattern in murine spleen-particularly for the macrophage clusters. For example, Cluster 5 likely corresponds to MZM, yet its pixels are scattered and mixed with other clusters, lacking clear layered pattern. In contrast, MultiGATE recovers the expected spatial pattern of macrophage subtypes relative to GCs: WPM appear closest, followed by MZM, and RPM farthest (Supplementary Fig. 5B and Supplementary Note S7). Beyond spatial clustering, MultiGATE identifies regulatory associations by linking proteins to related gene sets via cross-modality attention. Applied to CD3, it assigned higher attention scores to CD3-related genes than to B cell-related, macrophage-related, or random controls, confirming its ability to recover meaningful gene-protein relationships (Supplementary Note S10).

Next, we employed MultiGATE to analyze a Slide-tags dataset[2] generated from metastatic melanoma samples, where

open chromatin and RNA expression were profiled in single nuclei, along with spatial barcodes providing the spatial location of the nuclei. This dataset primarily consisted of two tumor clusters, as annotated in the original study[2] (Supplementary Fig. 6A). Consistent with these annotations, MultiGATE, along with SpatialGlue, and Seurat WNN accurately divided the tumor cells into two clusters (Supplementary Fig. 6B and D). Further analysis of DEGs identified by MultiGATE between tumor Cluster 1 and Cluster 2 revealed distinct expression patterns (Supplementary Fig. 6C and E). Cluster 1 was characterized by a mesenchymal-like state, with elevated expression and chromatin accessibility of *TNC*, a marker of invasion and metastasis, alongside *PLCB4* and *CPEB2*, which support YAP signaling–a key driver of epithelial-mesenchymal transition and cellular plasticity in melanoma[39–43]. Cluster 1 also showed downregulation of MHC genes (*CTSB*, *HSP90AA1*, *HSP90AB1*, and *LGMN*; Supplementary Fig. 6E), suggesting immune evasion and potential resistance to

immunotherapy, a trait often linked to aggressive tumor behavior[41]. In contrast, Cluster 2 displayed a melanocytic-like state with high expression of *DCT* and *APOE*, both implicated in treatment resistance via maintenance of melanocyte identity and ferroptosis evasion[44,45]. This melanocytic state was further supported by the significant enrichment of Cluster 2 upregulated genes (in Supplementary Fig. 6E) in pigmentation (GO:0043473; Hypergeometric $P = 1.38 \times 10^{-4}$) and in MITF-regulated pathways (R-HSA-9730414; Hypergeometric $P = 6.47 \times 10^{-6}$), where MITF is a master regulator of melanocyte identity and drives melanocyte development. Additionally, Cluster 2 showed a relative upregulation of MHC genes within the KEGG antigen processing and presentation pathway, with 7 out of 27 genes showing a fold change greater than 1.2 (Hypergeometric test, $P = 4.63 \times 10^{-6}$), indicating enhanced antigen presentation. This increased immune visibility likely promotes greater interaction with cytotoxic T cells, contributing to a more proliferative and therapy-sensitive phenotype in Cluster 2[46,47]. Together, these findings reveal that heterogeneity in melanoma identified by MultiGATE reflects distinct cell states with divergent therapeutic vulnerabilities.

To further demonstrate MultiGATE's versatility across measurement technologies and tissue types, we performed three additional evaluations: (1) integration of spatial transcriptomics and metabolomics (Supplementary Fig. 7 and Supplementary Note S11), showing that MultiGATE can jointly analyze modalities without genomic distance priors; (2) analysis of spatial RNA + protein data in human breast cancer (Supplementary Fig. 8 and Supplementary Fig. 2B); and (3) a simulated breast cancer-patterned spatial ATAC + RNA dataset (Supplementary Fig. 9 and Supplementary Note S12), confirming its applicability to tissues lacking stereotypical structure. In all cases, MultiGATE outperformed competing methods, accurately recovering spatial domains.

## Discussion

Spatial multi-omics analysis has emerged as a powerful approach for understanding the interplay between genes, chromatin accessibility, protein, and spatial organization in biological systems. In this study, we propose MultiGATE, a two-level graph attention framework that leverages the graph attention mechanism to integrate spatial multi-omics data.

Compared to existing methods, MultiGATE provides a unique advantage in simultaneously embedding the pixels and inferring the cross-modality regulatory relationship, which is not available in alternative approaches such as SpatialGlue and Seurat WNN. The attention scores incorporate genomic distances as prior knowledge to model cross-modality regulatory interactions, providing new insights into transcriptional regulation. Furthermore, the latent representations learned by MultiGATE enable accurate spatial clustering and enhanced visualization.

In conclusion, MultiGATE is a significant approach in the analysis of spatial multi-omics data. Its integration of multi-omics and spatial information, generation of high-quality embeddings, and inference of regulatory interactions make it a valuable tool for unraveling gene regulation and spatial organization. We anticipate that MultiGATE will facilitate discoveries and advancements across various fields of biological research.

Beyond modeling cis-regulation, MultiGATE can also capture trans-regulatory interactions by incorporating TF binding priors and learning TF–peak–gene attention scores. While initial results are promising, large-scale validation remains challenging due to the limited availability of high-quality ChIP-seq datasets in brain tissues. Future efforts will aim to broaden trans-regulatory validation as more tissue-specific datasets become accessible.

Currently, MultiGATE does not incorporate histology images into pixel embeddings, as only bright-field images were provided in the spatial ATAC–RNA–seq datasets[1]. Previous studies have demonstrated that combining histology images with gene expression data improves cell segmentation[48,49] and enables high-resolution gene expression[50,51]. As a future direction, subsequent iterations of MultiGATE could integrate an additional pre-trained encoder designed to process histology images.

## Methods

### The framework of MultiGATE

MultiGATE integrates spatial multi-omics data and the key feature of MultiGATE is that it provides both the cross-modality connection of the features, which reveals the regulatory relationship across modalities (for example, the peak–gene association in spatial ATAC–RNA–seq data), and the latent embeddings of the spatial spots, which can be used for downstream analysis including clustering and visualization of the spatial spots. These are achieved through a two-level graph attention auto-encoder that seamlessly integrates the cross-modality features and the spatial proximity of the spots. The first-level auto-encoder employs the feature connectivity graph to integrate the features of different modalities in spatial multi-omics data. It effectively combines the multi-modality information and learns attention scores, which capture the cross-modality connection of the features, facilitating inference of the regulatory relationships. The second-level auto-encoders use the neighboring graph of the spots to capture the spatial information of the spots in each modality, respectively, which encourages spatial smoothness for the enhancement of the latent embeddings of the spots. The latent embeddings and cross-modality regulatory scores provided by MultiGATE play a vital role in facilitating subsequent integration and downstream analysis for spatial multi-omics data, thereby enabling a more comprehensive and insightful exploration of the data.

### The two-level graph attention auto-encoder

Let $\mathbf{X}_1 \in \mathbb{R}^{N \times p}$ be the preprocessed data matrix from one modality with $N$ spots and $p$ features, $\mathbf{X}_2 \in \mathbb{R}^{N \times q}$ be the preprocessed data matrix from another modality with $N$ spots and $q$ features.

The first-level auto-encoder in MultiGATE, referred to as the cross-modality graph attention auto-encoder, employs the feature connectivity graph as a prior to integrate cross-modality features of different modalities in spatial multi-omics data. The feature connectivity graph is an adjacency matrix denoted by $\mathbf{A}$, where each entry $A_{ij}$ represents the connection between feature $i$ and feature $j$. The features in the adjacency matrix are the union of two modalities in spatial multi-omics data. For spatial ATAC–RNA–seq data[1], features are the union of genes and chromatin peaks. The value of $A_{ij}$ lies in the range of $(1, e]$ and is determined by the base pair distance between peak $i$ and gene $j$, with the value decreasing as the base pair distance increases, while being restricted within 150 kb. For SPOTS data[3], features encompass the union of genes and proteins. In this scenario, $A_{ij} = 1$ if and only if the gene $i$ encodes the protein $j$ or the subunits of protein $j$. In the feature connectivity graph, self-loops are added for each feature: $A_{ii} = 1$, for all $i$. The encoder in the first-level cross-modality auto-encoder generates the integrated feature matrix by aggregating information from the features that are connected across modalities through the feature connectivity graph $\mathbf{A}$. More specifically, we transpose $\mathbf{X}_1$ and $\mathbf{X}_2$, concatenate $\mathbf{X}_1^T$ and $\mathbf{X}_2^T$ along the dimension of the features and obtain the merged data matrix $\mathbf{X}^T \in \mathbb{R}^{(p+q) \times N}$. Let $\mathbf{X}_{(f)}^T$ be the data vector for feature $f$, $\forall f \in \{1, 2, \cdots, p + q\}$. The encoder integrates the cross-modality features as follows:

$$\bar{\mathbf{X}}_{(f)}^T = \sigma_1 \left( \sum_{g \in \mathbf{A}_f} \text{att}_{gf} \mathbf{X}_{(g)}^T + \text{att}_{ff} \mathbf{X}_{(f)}^T \right), \tag{1}$$

where $\bar{\mathbf{X}}_{(f)}^T$ is the integrated data for feature $f$, $\sigma_1$ is the ReLU activation function, $\mathbf{A}_f$ represents the set of features that are connected to feature $f$ in the feature connectivity graph $\mathbf{A}$, $\text{att}_{gf}$ is the attention score between feature $g$ and feature $f$, and $\text{att}_{ff}$ is the self-attention for feature $f$. These attention scores are learned adaptively based on a self-attention mechanism[52]. For cross-modality attentions, we first calculate the attention coefficient from feature $f$ to feature $g$ as follows:

$$e_{fg} = \text{Sigmoid}\left(\mathbf{v}_1^T\left(\mathbf{X}_{(f)}^T\right) + \mathbf{v}_2^T\left(\mathbf{X}_{(g)}^T\right)\right), \tag{2}$$

where $\mathbf{v}_1$ and $\mathbf{v}_2$ are trainable parameters, and the Sigmoid activation function is used. Next, the attention scores are computed with a Bayesian-like approach:

$$\text{att}_{fg} = \frac{A_{fg}\exp(e_{fg})}{\sum_{h \in \mathbf{A}_f} A_{fh}\exp(e_{fh})}, \tag{3}$$

where

$$A_{fg} = \exp\left(\left(\frac{\text{dist}_{fg} + \text{bp\_width}}{\text{bp\_width}}\right)^{-0.75}\right). \tag{4}$$

$\text{dist}_{fg}$ is the genomic distance between features $f$ and $g$. bp_width is the bandwidth in the kernel set as 400 (A sensitivity analysis in Supplementary Note S13). $A_{fg}$ represents the prior knowledge that quantifies the connection between features $f$ and $g$ based on their genomic distance, and its value decays when the genomic distance increases; The attention coefficient $e_{fg}$ represents the connection between the features derived from the observed data. In addition to the cross-modality attention $\text{att}_{fg}$, we also incorporate a self-attention $\text{att}_{ff}$ in computing $\bar{\mathbf{X}}_{(f)}^T$, to preserve the information in feature $f$ itself. $\text{att}_{ff}$ is computed the same as $\text{att}_{fg}$, with $\text{dist}_{ff}$ set to 0 in computing $A_{ff}$. In summary, the first-level auto-encoder integrates the cross-modality features, and the learned attention scores $\text{att}_{fg}$ reflect the strength of the connections between the cross-modality features and can be used for inferring their regulatory relationships.

Following the acquisition of integrated features $\bar{\mathbf{X}}^T$, we proceed to the second-level auto-encoders in MultiGATE, which are the within-modality graph attention auto-encoders. The second-level auto-encoders employ the spatial neighborhood graph to aggregate information from neighboring spots in spatial multi-omics data. The spatial neighborhood graph captures the similarity of neighboring spots using an adjacency matrix denoted as $\mathbf{S}$, where $S_{ij} = 1$ if spot $j$ is an immediate spatial neighbor of spot $i$ (and $S_{ii} = 1$ for all $i$). Neighborhoods are defined based on direct spatial adjacency[4,53]. The spatial RNA +protein[3] technique uses the 10× Visium platform, each spot has six neighbors, following a hexagonal grid layout[53]. For Spatial ATAC−RNA−seq data, neighborhood size reflects pixel resolution: in the 50 μm human hippocampus dataset, each pixel has four adjacent neighbors (up/down/left/right), while in the 20 μm P22 mouse brain dataset, each pixel has eight neighbors in a 3 × 3 grid. The average number of neighbors per spot across datasets is visualized in Supplementary Note S14. The spatial neighborhood graph allows us to effectively represent the spatial relationships between spots and incorporate their similarity in the auto-encoder. The output from the first-level cross-modality encoder, $\bar{\mathbf{X}}^T$, is first transposed to $\bar{\mathbf{X}}$ and then split to $\bar{\mathbf{X}}_1$ and $\bar{\mathbf{X}}_2$ according to the modality. $\bar{\mathbf{X}}_1$ and $\bar{\mathbf{X}}_2$ are the input for the second-level auto-encoders. For the $k$-th modality, we utilize an auto-encoder with the structure described below to capture the information from neighboring spots, where $k \in \{1, 2\}$. Let $\bar{\mathbf{x}}_{ki}$ be the integrated feature vector for modality $k$ of spot $i$, $\forall i \in \{1, 2, \cdots, N\}$. Let $L$ be the number of layers in the encoder. The output of the encoder in

the $l$-th layer ($l \in \{1, 2, \cdots, L-1\}$) is calculated as follows:

$$\bar{\mathbf{x}}_{ki}^{(l)} = \sigma_2\left(\sum_{j \in \mathbf{S}_i} \text{att}_{ij}^{(l)}\left(\mathbf{W}_k^l \bar{\mathbf{x}}_{kj}^{(l-1)}\right)\right), \tag{5}$$

where $\mathbf{W}_k^L$ is the trainable projection matrix in layer $l$ of modality $k$, $\sigma_2$ is the ELU activation function, $\mathbf{S}_i$ is the set of neighbors for spot $i$ in the spatial neighborhood graph $\mathbf{S}$ (including spot $i$ itself) and $\text{att}_{ij}^{(l)}$ is the spatial attention score between spot $i$ and spot $j$ in the $l$-th graph attention layer. To obtain $\text{att}_{ij}^{(l)}$, we first calculate the attention coefficient from spot $i$ to neighboring spot $j$ in the $l$-th encoder as follows:

$$e_{ij}^{(l)} = \text{Sigmoid}\left(\mathbf{v}_{\text{self}}^T\left(\mathbf{W}_k^l \bar{\mathbf{x}}_{ki}^{(l-1)}\right) + \mathbf{v}_{\text{nei}}^T\left(\mathbf{W}_k^l \bar{\mathbf{x}}_{kj}^{(l-1)}\right)\right), \tag{6}$$

where $\mathbf{v}_{\text{self}}$ and $\mathbf{v}_{\text{nei}}$ are trainable vectors. The attention score $\text{att}_{ij}^{(l)}$ is then computed as softmax

$$\text{att}_{ij}^{(l)} = \frac{\exp(e_{ij}^{(l)})}{\sum_{d \in \mathbf{S}_i}\exp(e_{id}^{(l)})}. \tag{7}$$

Our proposed attention scores differ from the standard graph attention auto-encoder[52] in the following two important aspects. Firstly, the calculation of cross-modality attention scores incorporates prior information on the connection of cross-modality features: it considers the prior knowledge that a peak is less likely to regulate a gene when the genomic distance is larger. Secondly, the trainable projection matrices in the cross-modality attention coefficients in equation (2) are fixed as identity matrices: we set $\mathbf{v}_1^T(\mathbf{X}_{(f)}^T)$ instead of $\mathbf{v}_1^T(\mathbf{W}_1\mathbf{X}_{(f)}^T)$. This decision is based on the intrinsic strong connection of the cross-modality features: regulatory regions/peaks regulate gene expression, and gene expression directly affects protein level. Performing an early projection could potentially disrupt this information, hence, we avoid including the projection matrices.

The last encoder layer (i.e., $L$-th layer) does not employ the attention layer, and its output is calculated as follows:

$$\bar{\mathbf{x}}_{ki}^{(L)} = \sigma_2\left(\sum_{j \in \mathbf{S}_i} \mathbf{W}_k^L \bar{\mathbf{x}}_{kj}^{(L-1)}\right), \tag{8}$$

where the output $\bar{\mathbf{x}}_{ki}^{(L)}$ is the latent embedding learned for the $k$-th modality.

After the input passes through the cross-modality encoder and the within-modality encoder, we obtain the latent embeddings $\bar{\mathbf{X}}^{(L)}$. Next, these latent embeddings undergo reconstruction first through the within-modality decoder and then through the cross-modality decoder, where both decoders are symmetric to their corresponding encoders. More specifically, the within-modality decoder takes the latent embeddings $\bar{\mathbf{X}}^{(L)}$ as the input (i.e., $\tilde{\mathbf{x}}_{ki}^{(L)} = \bar{\mathbf{x}}_{ki}^{(L)}$), its output in the $(l-1)$-th layer ($l \in \{L, \cdots, 2\}$) is calculated as follows:

$$\tilde{\mathbf{x}}_{ki}^{(l-1)} = \sigma_2\left(\sum_{j \in \mathbf{S}_i} \hat{\text{att}}_{ij}^{(l-1)}\left(\hat{\mathbf{W}}_k^l \tilde{\mathbf{x}}_{kj}^{(l)}\right)\right). \tag{9}$$

Symmetric to the within-modality encoder, the output in the outermost within-modality decoder layer is calculated as follows:

$$\tilde{\mathbf{x}}_{ki}^{(0)} = \sigma_2\left(\sum_{j \in \mathbf{S}_i} \hat{\mathbf{W}}_k^1 \tilde{\mathbf{x}}_{kj}^{(1)}\right). \tag{10}$$

MultiGATE sets the parameters in the within-modality decoder to be the same as those in the within-modality encoder to avoid overfitting: $\hat{\mathbf{W}}_k^l = (\mathbf{W}_k^l)^T$, for $l = 1, \cdots, L$ and $\hat{\text{att}}_{ij}^{(l)} = \text{att}_{ij}^{(l)}$, for $l = 1, \cdots, L-1$. We concatenate the output of the within-modality decoder for both modalities and denote it as $\tilde{\mathbf{X}}$. Next, the cross-modality decoder takes $\tilde{\mathbf{X}}$ as the input, and outputs the reconstructed data:

$$\hat{\mathbf{X}}_{(f)}^T = \sigma_2 \left( \sum_{g \in \mathbf{A}_f} \hat{\text{att}}_{gf} \tilde{\mathbf{X}}_{(g)}^T + \hat{\text{att}}_{ff} \tilde{\mathbf{X}}_{(f)}^T \right), \tag{11}$$

where $\hat{\mathbf{X}}_{(f)}^T$ is the reconstructed data for feature $f$. MultiGATE sets the cross-modality decoder to be symmetric to the cross-modality encoder to avoid overfitting: $\hat{\text{att}}_{gf} = \text{att}_{gf}$ and $\hat{\text{att}}_{ff} = \text{att}_{ff}$.

## The overall architecture of MultiGATE

In all datasets, the cross-modality mechanism consists of one GAT (Graph Attention Network) layer with two learnable vectors to compute attention scores between cross-modality features. In the cross-modality attention mechanism, we integrate the information of each gene with its neighboring peaks (or proteins) according to attention scores, and each peak (or protein) with its neighboring genes. The cross-modality encoder then outputs two enhanced data matrices–one of size (Peaks × Spots) and one of size (Genes × Spots)–which integrate information from the other modality and are passed on to the within-modality attention step.

For the within-modality attention autoencoders, each encoder has two hidden layers (dimensions 512 and 30), and the decoder is symmetric to the encoder. A sensitivity analysis for a latent dimension of 30 is presented in Supplementary Note S15.

## Loss function and training details

There are two components in the loss function. The first component is the reconstruction loss for the modalities.

$$\sum_{i=1}^N ||\mathbf{X}_{1i} - \hat{\mathbf{X}}_{1i}||_2 + \sum_{i=1}^N ||\mathbf{X}_{2i} - \hat{\mathbf{X}}_{2i}||_2. \tag{12}$$

The second component is the CLIP loss[54], which encourages the latent embeddings for different modalities, $\bar{\mathbf{X}}_1^{(L)}$ and $\bar{\mathbf{X}}_2^{(L)}$ to be similar and further facilitates cross-modality integration, in addition to the cross-modality graph attention. The CLIP loss is calculated as follows:

$$-\frac{1}{2N} \sum_{i=1}^N \log \frac{\exp(\text{sim}(\mathbf{f}_i, \mathbf{g}_i) \exp(\text{temp}))}{\sum_{j=1}^N \exp(\text{sim}(\mathbf{f}_i, \mathbf{g}_j) \exp(\text{temp}))} + $$
$$-\frac{1}{2N} \sum_{i=1}^N \log \frac{\exp(\text{sim}(\mathbf{g}_i, \mathbf{f}_i) \exp(\text{temp}))}{\sum_{j=1}^N \exp(\text{sim}(\mathbf{g}_i, \mathbf{f}_j) \exp(\text{temp}))}, \tag{13}$$

where the function *sim* represents cosine similarity, $\mathbf{f}_i = \mathbf{M}_1^T \bar{\mathbf{x}}_{1i}^{(L)}$, $\mathbf{g}_i = \mathbf{M}_2^T \bar{\mathbf{x}}_{2i}^{(L)}$, $\mathbf{M}_1$ and $\mathbf{M}_2$ are trainable projection matrices for modality 1 and modality 2, respectively. The embeddings $\mathbf{f}_i$ and $\mathbf{g}_i$, $\forall i \in \{1, 2, \cdots, N\}$ serve as the final output latent embeddings generated by MultiGATE, which are subsequently utilized for downstream analysis. temp is the temperature parameter in CLIP loss, which is set to 1 by default.

MultiGATE employs an Adam optimizer to minimize the loss with an initial learning rate of 1e-4. The weight decay is set as 1e-4. There are two kinds of activation functions: $\sigma_1$ is ReLU, while $\sigma_2$ is ELU. The default number of iterations is set to 1000. For details on runtime performance and memory usage across datasets, please refer to Supplementary Notes S16 and S17.

## MultiGATE attention-based *cis*-regulatory inference and validation

We use the following procedure to get peak–gene regulatory pairs from MultiGATE's cross-modality attention scores and evaluate the peak–gene regulatory pairs:

1. Cross-modality attention computation

    In the cross-modality autoencoder, for each peak $f$ and gene $g$, we first compute an unnormalized attention coefficient

    $$e_{fg} = \text{Sigmoid} \left( \mathbf{v}_1^T \mathbf{X}_{(f)}^T + \mathbf{v}_2^T \mathbf{X}_{(g)}^T \right),$$

    where $\mathbf{X}_{(f)}^T$ and $\mathbf{X}_{(g)}^T$ are the observed data for peak $f$ and gene $g$, and $\mathbf{v}_1$, $\mathbf{v}_2$ are learned vectors.

    We then incorporate a genomic distance prior

    $$A_{fg} = \exp \left( \left( \frac{\text{dist}_{fg} + \text{bp\_width}}{\text{bp\_width}} \right)^{-0.75} \right) \quad (\text{bp\_width} = 400 \text{ bp}),$$

    and normalize $e_{fg}$ to obtain the cross-modality attention score

    $$\text{att}_{fg} = \frac{A_{fg} \exp(e_{fg})}{\sum_{h \in \mathcal{N}_f} A_{fh} \exp(e_{fh})},$$

    where $\mathcal{N}_f$ is the set of features connected to peak $f$ in the feature connectivity graph.

2. Rescaling and thresholding

    First, we pooled all inter-feature attention scores (i.e., excluding self-attention) and performed a linear transformation to map them onto the interval [0, 1]. The resulting distribution was bimodal (Supplementary Note S18). We fitted a two-component Gaussian mixture model to these normalized scores and defined our threshold $\theta$ as the intersection point of the two Gaussian densities-i.e., the value at which the posterior probability of belonging to either component is equal (Supplementary Note S18). This data-driven cutoff separates low, noise-driven attention scores from high, biologically meaningful ones. Empirically, we obtained

    $$\theta_{\text{Human hippocampus}} = 0.204, \quad \theta_{\text{Mouse P22}} = 0.141.$$

    These thresholds were then used to call peak–gene links.

To validate the peak–gene regulatory pairs identified in the human hippocampus data, we used the human hippocampus eQTL[17] data. To validate the peak–gene regulatory pairs identified in the mouse brain data, we used the EnhancerAtlas[28] and EGAS[29] data obtained from the mouse cortex and striatum: The EnhancerAtlas obtains regulatory information by predicting consensus enhancers based on high-throughput datasets, including histone modification, CAGE, GRO-seq, transcription factor binding, and DHS; The EGAS dataset utilizes single-cell sequencing data and a non-parametric permutation-based procedure to predict the genome-wide enhancer-gene associations. However, mouse eQTL data was not employed due to the absence of the corresponding tissues of interest. During validation, we restricted our analysis to genes present in these external datasets and evaluated whether their linked peaks were supported by independent evidence.

A peak–gene pair is considered to be supported by EnhancerAtlas or EGAS if the peak overlaps with a genomic region that regulates the corresponding gene. A peak–gene pair is considered to be supported by eQTL if there is an eQTL locus within the peak and the locus is associated with the corresponding gene.

## Data description

We utilize MultiGATE to analyze diverse types of spatial multi-omics data, including spatial ATAC–RNA-seq, spatial snRNA-ATAC-seq, spatial protein-RNA-seq, and spatial RNA + metabolomics. In the context of spatial ATAC–RNA-seq, Spatial Epigenome-Transcriptome Co-profiling[1] generates data from the juvenile mouse brain as well as the adult human brain. This technique allows for the simultaneous profiling of epigenetic and transcriptomic characteristics, providing insights into the regulatory relationships between the genome and gene expression within a spatial context. For spatial snRNA-ATAC-seq, Slide-tags[2] generates multi-omics measurements capturing open chromatin and RNA within the same cells derived from metastatic melanoma. This approach enables the investigation of both chromatin accessibility and gene expression profiles within individual cells, facilitating a deeper understanding of the molecular characteristics of melanoma cells. In the case of spatial protein-RNA-seq, SPOTS[3] generates high-throughput data that enables simultaneous profiling of spatial transcriptomics and protein expression within the mouse spleen. By integrating transcriptomic and proteomic information, SPOTS offers a comprehensive view of the spatially resolved molecular landscape of the spleen, shedding light on the interplay between gene expression and protein abundance. For spatial RNA and metabolomics data, the SMA[55] protocol enabled the generation of mouse brain spatial RNA + metabolomics profiles by integrating spatial transcriptomics (SRT) and MALDI-MSI on the same tissue section. For the breast cancer spatial RNA + protein dataset, the Visium CytAssist Spatial Gene and Protein Expression assay[56] was used to simultaneously profile gene and protein expression with spatial resolution in formalin-fixed, paraffin-embedded tissue samples, generating the human breast cancer dataset.

## Data preprocessing

For the scRNA-seq or snRNA-seq data, we first log-transform and normalize the raw gene expression by library size using Scanpy[57]. Next, we select the top 3000 highly variable genes as input. For the scATAC-seq or snATAC-seq data, we first select the top 50,000 highly variable peaks and then run TF-IDF on the chromatin accessibility matrix. Next, we log-transform and normalize the processed chromatin accessibility matrix by sequencing depth. For the protein data, we apply CLR (centered log-ratio transformation) to the protein matrix using muon[58].

## Clustering

We employ two clustering strategies for the latent embeddings obtained from MultiGATE. The first strategy involves averaging the latent embeddings for each spot and subsequently applying Louvain clustering[59], as utilized in Slide-tags tumor dataset or mclust[60], to the averaged embeddings in SPOTS spleen. The second strategy involves employing Seurat WNN (FindMultiModalNeighbors function)[7] to construct a weighted graph based on the latent embeddings, and then we apply Louvain clustering using this graph in the mouse brain as well as the adult human brain from spatial ATAC–RNA–seq analyses. In general, both clustering methods yield similar results, with the WNN approach providing more refined details. A sensitivity analysis of the Louvain clustering resolution is presented in Supplementary Note S19.

## Identifying differentially expressed genes

Wilcoxon test is employed to discover DEGs for each identified spatial domain (Benjamini–Hochberg adjustment).

## Gene Ontology enrichment analysis

For the Spatial ATAC–RNA–seq dataset[1] generated from the adult human brain hippocampus, we conducted the gene set enrichment analysis implemented in the GSEAPY package[61] to discover the enriched GO terms for spatially variable genes in the detected domain with adjusted $P$ value $< 0.01$.

## Parameter settings for comparison methods

For all comparing methods, we used default hyperparameters as specified in their official documentation: SpatialGlue (https://spatialglue-tutorials.readthedocs.io/en/latest/), Seurat WNN (https://satijalab.org/seurat/articles/weighted_nearest_neighbor_analysis). Full parameter settings are detailed in Supplementary Note S20.

## Quantitative metrics calculation

We employed multiple clustering evaluation metrics to quantitatively evaluate the performance of spatial clustering results across different methods.

**Datasets with ground truth labels.** For the adult human hippocampus dataset (spatial ATAC–RNA–seq), the mouse brain dataset (spatial transcriptomics + metabolomics), the breast cancer-patterned spatial ATAC + RNA dataset (Supplementary Fig. 9), which have ground truth spatial domain annotations, we computed the following metrics to assess clustering agreement: Rand Index (RI), ARI, Adjusted Mutual Information, Normalized Mutual Information, Homogeneity, Completeness, V-measure, Fowlkes-Mallows Index.

**Datasets without ground truth labels.** For the P22 mouse brain dataset (spatial ATAC–RNA–seq), the mouse spleen dataset (spatial RNA + protein), and human breast cancer (spatial RNA + protein), where no ground truth labels are available, we adopted the ICC[27] to evaluate clustering quality (Supplementary Note S21).

## Reporting summary

Further information on research design is available in the Nature Portfolio Reporting Summary linked to this article.

# Data availability

All datasets analyzed in this paper have already been published and are available on public websites. All data utilized in the paper are accessible for open download. The Spatial epigenome-transcriptome, including the human hippocampus and mouse brain[1] is provided by the UCSC cell browser (https://brain-spatial-omics.cells.ucsc.edu/). The Slide-tags dataset is collected from the Single Cell Portal (https://singlecell.broadinstitute.org/single_cell/study/SCP2176). The mouse spleen data (spatial RNA + Protein) generated by SPOTS can be found at (https://www.ncbi.nlm.nih.gov/geo/query/acc.cgi?acc=GSE198353). The mouse brain spatial RNA + metabolomics dataset can be found at (https://data.mendeley.com/datasets/w7nw4km7xd/1). The human breast cancer data (spatial RNA + Protein) can be found at (https://www.10xgenomics.com/datasets/gene-and-protein-expression-library-of-human-breast-cancer-cytassist-ffpe-2-standard). The processed datasets used in this study have been deposited to Figshare and can be accessed at: https://doi.org/10.6084/m9.figshare.27978765. Source data are provided with this paper.

# Code availability

The code used to develop the model, perform the analyses, and generate the results of this study has been deposited in the GitHub repository cuhklinlab/MultiGATE (https://github.com/cuhklinlab/MultiGATE) under the Apache License 2.0. The specific version of the code associated with this publication (v0.1.0) has been archived on Zenodo and is accessible via 10.5281/zenodo.16310792[62].

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

## Acknowledgements

We thank Prof. Hongyu Zhao from Yale University for his valuable and constructive suggestions on the manuscript. This work is supported by the Chinese University of Hong Kong startup grant (4930181 to Z.L.), the Chinese University of Hong Kong Science Faculty's Collaborative Research Impact Matching Scheme (CRIMS 4620033 to Z.L.), the Chinese University of Hong Kong direct grants (4053540 to Z.L., 4053586 to Z.L.), and Research Grants Council, University Grants Committee (GRF 14301120 to Z.L., GRF 14300923 to Z.L.). This work is supported in part by the Innovation and Technology Commission (ITCPD/17-9 to C.Y.); Hong Kong Research Grants Council Grants (16301419 to C.Y., 16308120 to C.Y., 16307221 to C.Y., 16307322 to C.Y., 16302823 to C.Y., 16309424 to C.Y.); The Hong Kong University of Science and Technology Startup Grants (R9405 to C.Y.) and the Big Data Institute (Z0428 to C.Y.). This work is supported in part by the National Key Research and Development Project of China (Grant No. 2023YFF1204802 to Y.Z.), National Science and Technology Innovation 2030 Major Program (Grant No. 2021ZD0200100 to Y.Z.). This work is supported in part by the Open Research Fund of Key Laboratory of Mathematical Sciences (Central China Normal University), P. R. China (MPL2025ORG013 to J.T.).

## Author contributions

Z.L. and C.Y. supervised this study. J.L., J.M., and Z.L. proposed and developed MultiGATE's computational method. J.M., J.L., and J.X. conducted data analysis. C.Y., Y.Z., J.T., M.G., J.Q., and X.Z. provided advice and guided data analysis. J.M., J.X., and J.L. developed and released MultiGATE software package. J.M., J.L., J.X., Y.Z., C.Y., and Z.L. drafted the manuscript. J.M., C.Y., Y.Z., J.T., M.G., J.Q., and X.Z. revised the manuscript.

## Competing interests

The authors declare no competing interests.
