## [Transparent Peer Review file · Nature Communications]

MultiGATE: Integrative Analysis and Regulatory Inference in Spatial Multi-Omics Data via Graph Representation Learning

Corresponding Author: Dr Zhixiang Lin

Version 0:

Reviewer comments:

Reviewer #1

(Remarks to the Author)

The authors present MultiGATE, a graph attention auto-encoder framework for spatial multi-omics data integration. The method uniquely combines cross-modality regulatory inference with spatial pixel embedding, validated across diverse datasets. MultiGATE demonstrates superior spatial clustering accuracy and regulatory inference while providing biologically interpretable results. Here are my comments.

1. The current framework relies on genomic distance as a prior. How might MultiGATE adapt to modalities like spatial metabolomics, where genomic distance cannot be used as a prior? A brief discussion would clarify its potential for broader multi-omics integration.
2. The current framework focuses on cis-regulatory interactions. Can MultiGATE model trans-regulation? A discussion here would set expectations.
3. Parameter sensitivity and justification. For example, the choice of $bp_width=400$ for genomic distance decay (Eq. 4) lacks biological or statistical justification. Is this value empirically determined or borrowed from prior studies? A sensitivity analysis may help clarify robustness.
4. Scalability and practical use. How does runtime scale with dataset size? What hardware is required for typical analyses? A table summarizing these metrics would help users assess feasibility.
5. Can MultiGATE detect long-range interactions such as enhancer-promoter loops or genomics distance priors limit such discoveries?

(Remarks on code availability)

Reviewer #2

(Remarks to the Author)

Overall summary:

Jinzhai Li, et al, developed a graph-based deep learning model named MultiGATE to integrate spatial omics data for identifying tissue architecture. MultiGATE implements two graph attention auto-encoders to process data spatial multimodalities and spatial information. The results show MultiGATE outperforms SpatialGlue and Seurat WNN method using published data (e.g., Spatial ATAC-RNA-seq, SPOTS, and Slide-tags) with histological annotation. Overall, the method design is interesting, and three different spatial omics data are used to evaluate performance. The framework's innovation, rigor, and robustness, as well as the biological insights, are superficial and unclear. Comments are provided below:

Major comments:

1. It is difficult to identify the framework's novelty based on the current manuscript, as the innovative aspects are not clearly highlighted in the main text. Meanwhile, the final output (is it embedding?) and downstream tasks presented in Figure 1 are not clear.
2. The method of using H&E images is not clear. Is it only to determine the radius of the neighbor when constructing a graph? In addition, the description of line 346 is vague, "When the pixel size decreases from 50 μm to 20 μm , spots become spatially denser, and we set the radius parameter d such that the numbers of neighbors increase from four to eight on

average.” This claim needs data to support it.

3. For the benchmarking on mouse brain data, the parameter settings for SpatialGlue and Seurat WNN appear to be missing from the Methods section. Could authors specify these parameters to ensure a fair comparison among approaches? Furthermore, regarding the attention mechanism, would authors please provide data or a discussion illustrating which modality carries more weight in identifying the mouse brain tissue architecture?
4. The description of how the attention score is applied to predict peak–gene associations is unclear. Why did the authors choose not to validate these predictions using peak–motif–TF and gene relationships? Additionally, the insights from Figures C and D are not evident. Specifically, what is the innovation or importance of identifying RPS27L as a regulator, and how does it contribute to the study’s significance? Presenting more substantial biological examples would help demonstrate the practical value of this approach.
5. It is hard to disguise the overall performance between multiGATE and SpatialGlue on tissue architecture prediction; the quantitative score is needed. In addition, why do those three clustering results share the same color legend, which is unclear to me. In line 151, “SpatialGlue misassigned pixels in the inner layer to the same cluster as the outermost layer” is not convincing and confuses the pixels with spots.
6. The biology story and evidence in Figure 4 are not convincing and come across as somewhat confusing. First, SPOT’s measure is at the cellular level, which explains why the authors label tissue structures by cell types. However, it would be advisable to incorporate a spatial deconvolution step rather than making cell-level conclusions without single-cell resolution. Additionally, it is unclear what “Macrophages I, II, and III” (i.e., not typical Macrophage name) specifically refer to from a biological standpoint. Figure 4C should include a metric or value to indicate the significance level. Lastly, the way of presenting Figure 4E heatmap is weak in illustrating the variation in protein intensities for each cluster.
7. The biology insight from the Slide-Tags data set is not clear, and Extended Data Figure 2D is not cited.

Minor comments:

1. The color legend in Figure 3F should show the value rather than low and high.
2. Color legend in Figures 3F, 4D, and 4E should claim which values are used (e.g., score, normalized, and raw data).

(Remarks on code availability)

Reviewer #3

(Remarks to the Author)

The authors present MultiGATE, a novel method for integrating spatial multi-omics data that utilizes a two-level graph attention autoencoder. The key innovation is that it combines CLIP loss and two-level GATE to model the cross-modality regulatory relationship for regulatory inference, and to do spatial embedding for clustering/spatial domain identification. The method seems reasonable for this task and is well-described. The authors benchmarked against several popular methods on 3 datasets, showing MultiGATE’s better performance.

Below are my questions and comments that may improve this paper:

1. All the benchmarking datasets are from brain or spleen, both of which exhibit a stereotypical structure. Can it be applied to tissues without such stereotypical structure, such as tumors, skeletal muscle, colon, liver, lung, and adipose tissue? Benchmarking on a dataset from a tissue without a stereotypical structure would make a stronger case, to be more representative of the full breadth of spatial multi-omics datasets.
2. The authors include spatialGlue and Seurat WNN in their benchmark. However, several other popular methods have been proposed to integrate multi-omics datasets. I would like to suggest the authors introduce more state-of-the-art methods in the introduction such as totalVI, MultiVI, MOFA etc. Also, more methods should be included in the benchmarking analyses.
3. It’s interesting that attention mechanism in MultiGATE can capture the cis-regulatory interactions and identify the peak-pair regulation. Is it possible to model trans-regulation, and if so, why or why not? Additionally, could transcriptional regulation be modeled by incorporating gene-protein associations (i.e. how does gene expression affect protein level)?
4. In Figure 2A, the authors calculated ARI for each method and compared their clustering accuracy. What parameters are used for clustering (i.e. resolution of Louvain clustering)? Is the ARI presented the highest value each method could achieve? Also, the number of spots appears to be inconsistent across the different methods. Could the authors also clarify the clustering parameters used for the resulting clusters in Figure 3B?
5. Figure 3F illustrates several marker genes that are highly spatially enriched in certain brain regions, as reported in previous studies (line 183-196 in the manuscript). However, the identified clusters do not perfectly align with these brain regions (Figure 3A, B). How, then, is the marker gene Pde10a (Figure 3F), which is highly expressed in the CP region, identified through DEG analysis, given that the CP region appears to be separated into several distinct clusters?
6. The Methods section states that in the cross-modality autoencoder, the dimensions are set to match the number of spots. What is the rationale behind this choice? As is known, high-resolution spatial multi-omics techniques can generate tens of thousands of spots in a single slice. How can the autoencoder dimensions be set to match such a large number?
7. Line 399 says the dimension of the hidden layers of encoders are 512, 30 respectively. Does this mean the latent

dimension is set to be 30? Why is this number chosen? How does this affect downstream results?

8. For peak-gene connection, why is the base pair distance restricted within 150k? Line 311 says " $A_{i,j} = 1$ if and only if the gene name i matches the protein name of protein j ". How is the case handled when a single protein is encoded by multiple genes?

9. Is there a weighting factor to balance the components in the loss function?

10. It would be helpful to mention the computational time and memory usage in the paper.

(Remarks on code availability)

The authors provide an easily installable package, accompanied by comprehensive documentation. This includes step-by-step instructions for installing the MultiGATE package, along with tutorials on how to analyze various spatial multi-omics datasets using the tool. These resources enable users to replicate the provided results and examples with ease.

Version 1:

Reviewer comments:

Reviewer #1

(Remarks to the Author)

No more comments

(Remarks on code availability)

No more comments

Reviewer #2

(Remarks to the Author)

The authors solved all my concerns. I have no further questions.

(Remarks on code availability)

Reviewer #3

(Remarks to the Author)

The authors have sincerely addressed all the questions, thoroughly resolving all the unclear points. All my concerns have been fully addressed. They have successfully implemented the necessary revisions, and the manuscript is suitable for publication in Nature Communications as it is. I appreciate the authors' dedicated effort in improving the paper.

(Remarks on code availability)

Responses to Reviewers' Comments for Manuscript NCOMMS-24-80805

MultiGATE: Integrative Analysis and Regulatory Inference in Spatial Multi-Omics Data via Graph Representation Learning

We sincerely thank the reviewers for their constructive comments and suggestions, which have significantly improved our manuscript. We have carefully addressed each of the raised points and incorporated the necessary revisions. Below is a summary of the key changes made in response to Reviewers 1–3:

1. **Methodological Innovation of MultiGATE**

MultiGATE simultaneously models cross-modality regulatory interactions and extracts the latent representation of each pixel/spot in a single, unified framework. By directly modeling regulatory links (such as peak-gene associations, protein-gene interactions, and enzyme-metabolite associations) into its graph attention mechanism, MultiGATE learns more informative low-dimensional representations of each spatial pixel/spot by deeper integrating different modalities. In turn, these refined embeddings sharpen the attention scores between cross-modality features, yielding more accurate inference of cross-modality regulation in the native tissue context powered by spatial multi-omics data. The methodological innovation that distinguishes MultiGATE from existing work is elaborated in the main text and Supplementary Note S1.

2. **Extension to Trans-Regulatory Inference**

We have extended MultiGATE to capture trans-regulatory interactions by incorporating TF binding priors and learning TF–peak–gene attention scores. Using ChIP-seq data for SOX2 in brain tissue as ground truth, our method outperformed motif-only, TF binding potential, and cosine similarity baselines in predicting SOX2 binding, achieving an AUC of 0.8669 and AUPR of 0.4906 (Supplementary Note S2).

3. **Detection of Long-Range Enhancer–Promoter Interactions**

We have extended MultiGATE to detect long-range interactions between enhancers and promoters: across eight distance bins (0–150 kb, 150–300 kb, . . . , > 1.25 Mb), MultiGATE can distinguish the brain-specific enhancer–promoter contacts from those in other tissues across 24 diverse human tissues (Supplementary Note S4).

4. **Modeling of Protein–Gene Interactions**

We have extended MultiGATE to model gene-protein association. We have demonstrated that MultiGATE can discriminate CD3-related genes from three negative control sets using the cross-modality attention scores (Supplementary Note S3).

5. **Expanded Benchmarking and Method Comparison Across Multiple Datasets**

We evaluated MultiGATE on 7 datasets (3 newly added), covering five different spatial multi-omics platforms/technologies: Spatial ATAC-RNA, Slide-tags, Visium-CytAssist RNA-protein, SMA and SPOTS. These datasets include stereotypical tissues (brain, spleen) and a non-stereotypical sample (breast cancer), with various modality combinations (ATAC+RNA, RNA+protein/ADT, RNA+metabolite). We compared MultiGATE against 5 other methods (3 added in this revision): SpatialGlue, Seurat WNN, MOFA+, totalVI, and MultiVI (Extended Data Figs. 3-8).

6. **Multi-Metric Performance Evaluation**

When ground-truth tissue annotations are available, we report 8 clustering metrics (ARI, RI, AMI, NMI, Homogeneity, Completeness, V-measure, FMI); otherwise we use the combined intra-class correlation (ICC) to evaluate the clustering results. Regulatory accuracy is assessed by AUROC/AUPR for peak-gene, TF-peak links (Extended Data Figs. 3-8).

7. **Biological insights**

In spleen data (generated by SPOTS technology), through integrative analysis of protein expression (ADT), gene expression (RNA), and spatial localization, we accurately mapped three macrophage clusters to canonical macrophage subsets within the spleen: Red Pulp Macrophages (RPM), Marginal Zone Macrophages (MZM) and White Pulp Macrophages (WPM). These annotations were supported by distinct and consistent molecular signatures at both the protein and transcriptomic levels, along with their anatomically appropriate spatial distribution. This integrative approach demonstrates the biologically meaningful and interpretable results achieved through the multi-modal framework.

For the Slide-tags dataset generated from metastatic melanoma samples, MultiGATE accurately identified two major tumor clusters consistent with prior annotations and further distinguished them by distinct marker gene expression. One cluster exhibited a mesenchymal-like, immune-evasive state, associated with invasion, metastasis, and reduced

MHC expression. In contrast, the other cluster displayed a melanocytic-like, immune-visible phenotype, characterized by pigmentation pathways and MITF activity. These molecular differences suggest divergent potential responses to immunotherapy, demonstrating the framework's ability to reveal tumor heterogeneity.

In the adult human hippocampus dataset (spatial ATAC–RNA–seq), MultiGATE successfully linked regulatory elements to genes such as CA12 and PRKD3, which are involved in neuronal excitability and synaptic function, supported by hippocampus eQTLs. Beyond proximal interactions, MultiGATE identified long-range enhancer–gene links using HiChIP-augmented priors. Notably, it connected a distal enhancer 883.9 kb from CAMTA1, a hippocampus-enriched gene essential for neuronal survival, and another 507.9 kb from CSNK1A1, a kinase involved in neurodegenerative disease pathways. The interactions were supported by hippocampus-specific eQTLs and relevant GWAS signals, underscoring their potential functional relevance. MultiGATE also uncovered trans-regulatory TF–peak–gene relationships, exemplified by accurate prediction of SOX2 binding sites in brain tissue, validated against ChIP-seq data. These findings demonstrate MultiGATE's ability to capture both long-range and trans-regulatory mechanisms, offering biologically meaningful insights into hippocampal gene regulation.

We believe that we have addressed each comment and suggestion from the reviewers thoroughly. Our detailed, point-by-point responses—spanning 127 pages—are provided below.

Authors' Response to Reviewer 1

General Comments. The authors present MultiGATE, a graph attention auto-encoder framework for spatial multi-omics data integration. The method uniquely combines cross-modality regulatory inference with spatial pixel embedding, validated across diverse datasets. MultiGATE demonstrates superior spatial clustering accuracy and regulatory inference while providing biologically interpretable results.

Response:

Thank you very much for your encouraging comments and thorough review, which have significantly helped us improve our manuscript. Our responses to each of your comments are provided below.

Comment 1

The current framework relies on genomic distance as a prior. How might MultiGATE adapt to modalities like spatial metabolomics, where genomic distance cannot be used as a prior? A brief discussion would clarify its potential for broader multi-omics integration.

Response:

We thank the reviewer for this insightful suggestion. While our original MultiGATE implementation leverages a genomic distance-based prior to model peak-gene interactions, MultiGATE's cross-modality attention architecture is agnostic to the particular form of the prior—it can incorporate any biologically meaningful network that links features across modalities, including peak-gene, protein-gene and metabolite-gene. To demonstrate the flexibility of our framework, we have extended MultiGATE by replacing the genomic-distance prior with a biochemical reaction-based network that links metabolites to their cognate enzyme-encoding genes to integrate spatial RNA+ metabolomics data.

Specifically, we constructed a metabolite-gene adjacency matrix by extracting enzyme-metabolite associations from RaMP-DB [1], which integrates curated reaction and pathway annotations

Figure R1: Clustering results for a mouse brain spatial transcriptomics and spatial metabolomics dataset.

A Ground-truth spatial annotation of distinct brain regions (colored by region).

B Bar plot comparing multiple clustering metrics (adjusted Rand score, Rand score, adjusted mutual information, normalized mutual information, homogeneity, completeness, V-measure, and Fowlkes–Mallows index) for MultiGATE, SpatialGlue, and Seurat WNN, respectively. This plot shows that MultiGATE achieves superior performance to alternative approaches across all measures.

C Clustering assignments produced by MultiGATE, SpatialGlue, and Seurat WNN, with each method's Adjusted Rand Index (ARI) indicated in parentheses.

from HMDB, Reactome, WikiPathways, KEGG, ChEBI, LipidMaps, and Rhea. We then built a metabolite–gene adjacency matrix in which each metabolite is uniformly connected to all enzyme-encoding genes catalyzing its conversion. This reaction-derived adjacency replaces the original genomic-distance weights in MultiGATE’s cross-modality attention framework.

We evaluated MultiGATE with the biochemical reaction–based network that links metabolites to their cognate enzyme-encoding genes on a published mouse brain dataset that jointly profiles spatial transcriptomics and spatial metabolomics [2]. As shown in Fig. R1, MultiGATE achieves superior clustering performance—measured by adjusted Rand index, adjusted mutual information, V-measure, and five other metrics—when compared to SpatialGlue and Seurat WNN. These results demonstrate that by employing a reaction-based prior, MultiGATE can integrate spatial metabolomics with transcriptomics.

In summary, by supplying an appropriate biologically meaningful prior network that links features across modalities, including peak-gene, protein-gene and metabolite-gene, MultiGATE offers a general framework for joint analysis of diverse spatial multi-omics datasets, including spatial ATAC+RNA, spatial RNA+protein/ADT, and spatial RNA+metabolomics data.

Manuscript revisions

We have revised the manuscript as follows:

1. Incorporated the spatial RNA + metabolomics application into the Results section (lines 310–312).
2. Detailed the construction of the metabolite–gene adjacency matrix and its integration into MultiGATE in Supplementary Note S21.
3. Added the results for spatial RNA + metabolomics data in Extended Data Figure 3.

Comment 2

The current framework focuses on cis-regulatory interactions. Can MultiGATE model trans-regulation? A discussion here would set expectations.

Response:

We thank the reviewer for raising this important point. In the original version of MultiGATE, the attention mechanism primarily captures cis-regulatory interactions, modeling the influence of proximal regulatory elements (defined as spatial ATAC-seq peaks) on gene expression. To extend MultiGATE toward trans-regulation, we modified the model to incorporate transcription factor (TF) binding information explicitly. This modification enables the model to capture distal regulatory effects mediated by TFs, thus going beyond cis-regulation.

In this revised setup, we introduce a TF binding potential (TFBP) as an additional prior in the cross-modality autoencoder. Formally, for a given TF k and peak i , the prior is defined as:

$$\text{TFBP}_{k,i} = M_{k,i} \times X_i \quad (1)$$

where $M_{k,i} \in \{0, 1\}$ is the binarized motif binding score for TF k on peak i . The motifs were scanned using HOMER [3], with motif-to-TF mapping curated from [4]. It is set to 1 if at least one binding site for TF k is found in peak i , and 0 otherwise. X_i denotes the average chromatin accessibility of peak i across all spots. This score integrates both motif presence and chromatin openness, serving as a biologically informed prior for TF binding potential. The original peak–gene (excluding the TFs) attention mechanism remains unchanged, allowing MultiGATE to simultaneously learn both cis and trans regulations within a unified attention framework. To quantify predicted TF binding to peaks, we define the TF–peak integrated attention score as:

$$S_{k,i} = \alpha_{k,i} \times \sum_{j \in \mathcal{G}(i)} \text{Att}_{i,j} \quad (2)$$

where $\alpha_{k,i}$ is the learned attention score from TF k to peak i , $\text{Att}_{i,j}$ is the attention score from peak i to gene j , and $\mathcal{G}(i)$ is the set of genes linked to peak i . This metric prioritizes TF–peak pairs that are both accessible and functionally linked to gene expression.

We tested this framework using the spatial ATAC–RNA–seq dataset from the adult human hippocampus and evaluated the predictions for SOX2, a well-studied brain-specific TF. The ChIP-seq data for SOX2 in brain tissue were downloaded from the Cistrome database [5] and served as the ground truth for evaluating predicted TF binding peaks. We compared our model against several baseline methods: The motif-only baseline was constructed using the

binary motif TF binding score (defined as M in Eq. (1)) identified via HOMER [3], with motif-to-TF mapping curated from [4]; The TF binding potential baseline was defined as the product of binary motif binding and chromatin openness (defined as TFBP in Eq. (1)); The cosine similarity baseline was computed as the cosine similarity between the expression profile of the TF (from spatial RNA data) and the accessibility profile of each peak (from spatial ATAC data) across all the spots.

Figure R2: Comparison of methods for predicting SOX2 binding in the adult human hippocampus. Motif_Binding denotes the binary motif TF binding score. TFBinding_Potential is computed as the product of binary motif binding score and chromatin openness (defined as TFBP in Eq. (1)). The Cosine_TF_Peak is computed as the cosine similarity between the expression profile of the TF and the accessibility profile of each peak across all the cells. Attention_Score demonstrates the TF–peak integrated attention score defined as $S_{k,i}$ in Eq. (2).

Our method's learned attention scores (defined as $S_{k,i}$ in Eq. (2)) were highly predictive of TF binding events, with an AUC of 0.8669 and AUPR of 0.4906, substantially outperforming motif-only (AUC = 0.4906, AUPR = 0.1934), TF binding potential baseline (AUC = 0.6326, AUPR = 0.2845), and cosine similarity between TFs and peaks (AUC = 0.6280, AUPR = 0.2752), as shown in Fig. R2.

We believe this extended design makes MultiGATE a flexible and generalizable framework for modeling both cis and trans gene regulation. However, we acknowledge that validation remains limited by the availability of high-quality ChIP-seq datasets in brain tissues, such as the hippocampus. While our proof-of-concept with SOX2 is promising, we recognize the need for broader validation across TFs and plan to explore this further as more tissue-specific datasets are available. These capabilities and limitations are discussed in the revised manuscript (lines 333-337).

Manuscript revisions

We have revised the manuscript as follows:

1. Added trans-regulatory modeling results to the Results section (lines 149–154).
2. Described the TF binding potential prior and TF–peak attention score in Supplementary Note S2.
3. Added the capabilities and limitations of modeling trans-regulation in the Discussion.

Comment 3

Parameter sensitivity and justification. For example, the choice of `bp_width = 400` for genomic distance decay (Eq. 4) lacks biological or statistical justification. Is this value empirically determined or borrowed from prior studies? A sensitivity analysis may help clarify robustness.

Response:

We thank the reviewer for highlighting the importance of justifying the choice of the genomic distance decay parameter `bp_width` in Equation 4. The following paragraph is the description of `bp_width` in the methods section of MultiGATE:

`bp_width` used in MultiGATE:

In the manuscript, we defined our attention prior as

$$A_{fg} = \exp\left(\left(\frac{dist_{fg} + bp_width}{bp_width}\right)^{-0.75}\right), \quad (3)$$

where $dist_{fg}$ is the genomic distance between features f and g , and `bp_width` = 400 is the kernel bandwidth. Intuitively, A_{fg} encodes the biological prior that nearby features tend to interact more strongly, decaying smoothly as distance increases. The full attention score then combines this prior with data-derived evidence:

$$\mathbf{att}_{fg} = \frac{A_{fg} \exp(e_{fg})}{\sum_{h \in \mathbf{A}_f} A_{fh} \exp(e_{fh})}, \quad (4)$$

where e_{fg} are the attention coefficients inferred from the observed measurements.

To demonstrate the robustness of our model to the choice of `bp_width`, we conducted a sensitivity analysis varying `bp_width` from 200 to 550. Figures R3 and R4 display clustering performance on the human hippocampus and mouse P22 datasets, respectively. Across this range, MultiGATE consistently recovers the known anatomical layers, with only minimal fluctuations in clustering accuracy. Based on these results, we selected `bp_width` = 400 as a sensible default.

Manuscript revisions

We have updated a sensitivity analysis of `bp_width` in Supplementary Note S19 (see also lines 390–391 in the main text).

Human hippocampus dataset (spatial ATAC+RNA)

Figure R3: Human Hippocampus dataset clustering results of different settings of `bp_width` (200-550)

Figure R4: Mouse P22 clustering results of different settings of `bp_width` (200-550)

Comment 4

Scalability and practical use. How does runtime scale with dataset size? What hardware is required for typical analyses? A table summarizing these metrics would help users assess feasibility.

Response: Thank you for the comment.

To assess the scalability of MultiGATE, we evaluated its runtime on the spatial multi-omics datasets with varying sizes and complexities (Table 1). Training time increases with the number of spots and the dimensions of the features (genes, peaks, or protein markers). Nonetheless, our results demonstrate that even for larger datasets, the runtime remains feasible on standard high-performance hardware.

Hardware Configuration:

- **GPU:** 1 × Tesla V100-PCIE-32GB
- **CPU:** Intel Xeon Gold 6254 @ 3.10GHz (20 cores)
- **Operating System:** CentOS Linux 7 (Core)

Manuscript revisions

We have updated the detailed runtime and memory usage metrics in Supplementary Note S17 (see also lines 477–478 in the main text).

Table 1: Runtime and GPU Memory Usage for MultiGATE on Spatial Multi-Omics Datasets

Dataset	Dimensions	Memory Usage	Avg. Training Time
Adult Human Hippocampus (Spatial ATAC-RNA-seq [6])	RNA: 2500 pixels \times 7666 genes ATAC: 2500 pixels \times 28270 peaks	8726 MB	5.5 minutes
P22 Mouse Brain (Spatial ATAC-RNA-seq [6])	RNA: 9215 pixels \times 16252 genes ATAC: 9215 pixels \times 120400 peaks	31108 MB	1 hr 50 min
Mouse Spleen (Spatial RNA + ADT) (SPOTS [7])	RNA: 2563 spatial barcodes \times 14371 genes Protein: 2563 spatial barcodes \times 21 markers	1550 MB	2 minutes
Metastatic Melanoma (Spatial ATAC + RNA) (Slide-tags [8])	RNA: 2535 pixels \times 14807 genes ATAC: 2535 pixels \times 13665 peaks	4630 MB	8.5 minutes
Mouse Brain (Spatial RNA + metabolomics) (SMA [2])	RNA: 2820 spots \times 1538 genes Metabolomics: 2820 spots \times 1538 m/z's	1550 MB	1 minute

Comment 5

Can MultiGATE detect long-range interactions such as enhancer-promoter loops or genomics distance priors limit such discoveries?

Response:

We thank the reviewer for this insightful question. In our initial implementation, MultiGATE employs a genomic-distance prior that restricts candidate peak–gene edges to within 150 kb. However, the cross-modality attention framework in MultiGATE places no hard limit on distance and can incorporate more distal links when supported by external data.

To demonstrate MultiGATE’s ability to detect long-range interactions, we reanalyzed the adult human hippocampus Spatial ATAC–RNA-seq dataset. Under the 150 kb genomic distance prior alone, we obtained 16,347 short-range peak–gene candidates. We then augmented this prior by adding 205,627 enhancer–promoter contacts curated from HiChIP experiments database [9] across 24 diverse human tissues (Table 2). Specifically, any ATAC peak overlapping a HiChIP-annotated enhancer that loops to a gene promoter was introduced as a candidate edge in the peak-gene prior network, regardless of genomic distance.

After retraining MultiGATE with these augmented connections, we stratified all candidate edges into eight distance bins (0–150 kb, 150–300 kb, . . . , >1.25 Mb) and compared the attention scores assigned to brain-specific versus non-brain HiChIP loops in each bin. In every distance category, brain-specific HiChIP loops received significantly higher attention scores than other loops (Mann–Whitney U test, $P < 1 \times 10^{-3}$; Fig. R5).

Notably, the enhancer chr1:7669258-7670119 and its target gene CAMTA1, located 883.9 kb apart, were identified by MultiGATE with an attention score of 0.2194 and is supported by the eQTL pair chr1_7669531_G_C - CAMTA1. CAMTA1, a transcription factor enriched in the hippocampus, is crucial for Purkinje cell survival and cerebellar function, with its loss leading to ataxia, motor deficits, and neurodegeneration due to dysregulated neuronal gene expression [10]. Additionally, MultiGATE identified another enhancer at chr5:150000116-150001000 and its target gene CSNK1A1, with a distance of 507.9 kb. This enhancer-gene pair is supported by the hippocampus eQTL chr5_150000595_C_T - CSNK1A1, along with a risk

SNP (rs4705403, $p=1.00e-8$) associated with migraine in GWAS study. $CK1\alpha$ plays a crucial role in the pathogenesis of Alzheimer's and Parkinson's diseases by regulating key proteins involved in neurodegeneration [11].

These observations demonstrate that MultiGATE can detect long-range interactions when supplied with appropriate priors.

Manuscript revisions

We have updated the manuscript as follows:

1. Added long-range interaction analysis with HiChIP-augmented priors to the Results section (lines 146–149).
2. Detailed HiChIP-augmented prior construction and long-range interaction results in Supplementary Note S4.

Table 2: Number of peak–gene pairs per tissue (FitHiChIP, 5 kb resolution).

Tissue	Candidate peak-gene pairs
Aorta	5399
Blood	53084
Brain	10644
Breast	18694
Colon	765
Embryo	8958
Endometrioid endometrial tumor	51
Endometrium	11
Esophagus	16026
Eye	1062
Heart	8221
Kidney	3
Lung	14744
Lymph node	1163
Lymphocyte	28106
Muscle	1616
Ovary	1714
Prostate	227
Skin	26163
Stem cell	44
Stomach	5675
Thyroid	2666
Uterus	591

Figure R5: Tissue-specific trends in gene–peak attention scores across genomic distances. This line plot shows median attention scores for gene–peak pairs in Brain (red) and Other Tissues (blue) across eight distance bins (0–150kb to >1250kb). Shaded regions indicate 95% confidence intervals around each median. Statistical significance between Brain and Other Tissues was assessed using Mann–Whitney U tests, with asterisks above each bin denoting significance levels ($*p < 0.05$, $**p < 0.01$, $***p < 0.001$). “ES” marks the effect size.

References

- [1] J. Braisted, A. Patt, C. Tindall, *et al.*, “Ramp-db 2.0: A renovated knowledgebase for deriving biological and chemical insight from metabolites, proteins, and genes,” *Bioinformatics*, vol. 39, no. 1, btac726, Nov. 2022. DOI: 10.1093/bioinformatics/btac726.
- [2] M. Vicari, R. Mirzazadeh, A. Nilsson, *et al.*, “Spatial multimodal analysis of transcriptomes and metabolomes in tissues,” *Nature Biotechnology*, vol. 42, no. 7, pp. 1046–1050, 2024.
- [3] S. Heinz, C. Benner, N. Spann, *et al.*, “Simple combinations of lineage-determining transcription factors prime cis-regulatory elements required for macrophage and b cell identities,” *Molecular cell*, vol. 38, no. 4, pp. 576–589, 2010.
- [4] Z. Duren, X. Chen, J. Xin, Y. Wang, and W. H. Wong, “Time course regulatory analysis based on paired expression and chromatin accessibility data,” *Genome research*, vol. 30, no. 4, pp. 622–634, 2020.
- [5] R. Zheng, C. Wan, S. Mei, *et al.*, “Cistrome data browser: Expanded datasets and new tools for gene regulatory analysis,” *Nucleic acids research*, vol. 47, no. D1, pp. D729–D735, 2019.
- [6] D. Zhang, Y. Deng, P. Kukanja, *et al.*, “Spatial epigenome–transcriptome co-profiling of mammalian tissues,” *Nature*, vol. 616, no. 7955, pp. 113–122, 2023.
- [7] N. Ben-Chetrit, X. Niu, A. D. Swett, *et al.*, “Integration of whole transcriptome spatial profiling with protein markers,” *Nature Biotechnology*, vol. 41, no. 6, pp. 788–793, 2023.
- [8] A. J. C. Russell, J. A. Weir, N. M. Nadaf, *et al.*, “Slide-tags enables single-nucleus barcoding for multimodal spatial genomics,” *Nature*, vol. 625, no. 7993, pp. 101–109, Jan. 2024. DOI: 10.1038/s41586-023-06837-4.
- [9] W. Zeng, Q. Liu, Q. Yin, R. Jiang, and W. H. Wong, “Hichipdb: A comprehensive database of hichip regulatory interactions,” *Nucleic Acids Research*, vol. 51, no. D1, pp. D159–D166, Oct. 2022. DOI: 10.1093/nar/gkac859.

- [10] C. Long, C. E. Grueter, K. Song, *et al.*, “Ataxia and purkinje cell degeneration in mice lacking the camta1 transcription factor,” *Proceedings of the National Academy of Sciences*, vol. 111, no. 31, pp. 11 521–11 526, 2014.
- [11] S. Jiang, M. Zhang, J. Sun, and X. Yang, “Casein kinase 1 α : Biological mechanisms and theranostic potential,” *Cell Communication and Signaling*, vol. 16, pp. 1–24, 2018.

Authors' Response to Reviewer 2

General Comments. Jinzhai Li, et al, developed a graph-based deep learning model named MultiGATE to integrate spatial omics data for identifying tissue architecture. MultiGATE implements two graph attention auto-encoders to process data spatial multimodalities and spatial information. The results show MultiGATE outperforms SpatialGlue and Seurat WNN method using published data (e.g., Spatial ATAC-RNA-seq, SPOTS, and Slide-tags) with histological annotation. Overall, the method design is interesting, and three different spatial omics data are used to evaluate performance. The framework's innovation, rigor, and robustness, as well as the biological insights, are superficial and unclear.

Response:

We thank the reviewer for recognizing the promise of MultiGATE and for highlighting the need to make its innovation, rigor and robustness, and biological insights more explicit. Below, we summarize how the revised manuscript now makes each aspect—innovation, rigor and robustness, and biological insight—fully explicit:

1 Methodological Innovation

MultiGATE contains a two-level graph attention auto-encoder designed for spatial multi-omics data integration and regulatory inference. At the first level, a cross-modality attention mechanism directly integrates features in different modalities (e.g., peak–gene, protein–gene and metabolite–gene), while the simultaneously estimated attention scores can be used for cis-regulation, trans-regulation and protein–gene interactions. At the second level, a within-modality attention mechanism aggregates information across spatial neighbors. In version 1 of the manuscript, the cross-modality attention mostly captures cis-regulatory relationships, i.e., peak–gene association. In the revised manuscript, we have extended the cross-modality attention module beyond cis-regulatory inference to capture trans-regulatory mechanisms, such as transcription factor–peak–gene interactions, as well as protein–gene associations, and we

demonstrate MultiGATE’s versatility on a spatial transcriptomics–metabolomics dataset by incorporating enzyme–metabolite associations from RaMP-DB [1].

Compared to existing methods, the key methodological innovation of MultiGATE is that it extracts the latent embeddings of the pixels/spots in spatial multi-omics data, while simultaneously incorporates the regulatory relationship of the cross-modality features through the cross-modality attention mechanism and the spatial relationship of the pixels/spots through the within-modality attention mechanism: incorporating the regulatory relationship of the cross-modality features in obtaining the latent embedding of the pixels/spots allows deeper integration of different modalities, and the two (latent embeddings of the pixels/spots and regulatory relationship of the cross-modality features) can foster the estimation of each other. In addition, the analysis of the cross-modality regulatory relationship enabled by MultiGATE provides the unique insight in studying transcriptional regulation in the native tissue context powered by spatial multi-omics data.

The following paragraph describes the details on what distinguishes MultiGATE from existing methods. SpatialGlue [2] uses a graph convolution network (GCN) auto-encoder framework to extract a low-dimensional embedding for each spot/pixel, and the GCN auto-encoder does not incorporate the regulatory inference since the GCN takes the principal components of raw features as input, ignoring the feature-level cross-modality relationship. GLUE [3] can infer cis-regulatory relationships but does not consider the spatial information. In the scenario of spatial multi-omics data integration, MultiGATE can incorporate the spatial information through the within-modality attention mechanism, improving the extraction of latent embeddings and clustering results. Seurat WNN [4] is designed for single-cell multi-omics data. It uses the unsupervised framework (weighted-nearest neighbor) to learn the relative utility of each data type in each cell, and the weighted-nearest neighbor method neither models spatial information nor incorporates cross-modality regulation. MOFA+ [5] is designed for the integration of single-cell multi-modal data, using a linear factor model that decomposes the input matrices into the product of low-rank matrices (weight matrices and low-dimensional representation matrices); this linear factor model does not consider spatial information and the cross-modality regulatory relationship. totalVI [6] is designed for CITE-seq data (RNA+surface protein); it

uses the variational autoencoder (VAE) framework to model gene expression raw counts with negative-binomial (NB) distribution and the protein counts as an NB mixture of foreground and background signal; the variational autoencoder (VAE) framework in totalVI does not incorporate spatial information in the spatial multi-omics data. MultiVI [7] is based on a conditional variational autoencoder and models each modality using a specific distribution separately (negative-binomial (NB) distribution for gene expression raw counts and Bernoulli distribution for chromatin accessibility); this conditional variational autoencoder does not consider spatial information and the cross-modality regulatory relationship.

In contrast, MultiGATE simultaneously captures cross-modality regulatory interactions and extracts the latent representation of each pixel/spot in a single, unified framework. By modeling regulatory links (such as peak-gene associations, protein-gene interactions, and enzyme-metabolite associations) directly into its graph attention mechanism, MultiGATE learns more informative low-dimensional representations of each spatial pixel/spot. In turn, these refined embeddings sharpen the attention scores between cross-modality features, yielding more accurate inference of cross-modality regulation in the native tissue context powered by spatial multi-omics data.

2 Rigor and Robustness

a) Comprehensive benchmarking panel.

Besides the original 4 benchmark datasets (human hippocampus dataset (Spatial ATAC–RNA), P22 mouse brain dataset (Spatial ATAC–RNA), murine spleen dataset (SPOTS), metastatic melanoma dataset (Slide-tags)), in the revised version, we added one spatial RNA + metabolomics dataset (SMA technology) to show MultiGATE can be used to integrate spatial RNA + metabolomics, and we added one breast cancer dataset (Visium-CytAssist RNA–protein) and one simulation dataset (breast cancer-patterned spatial ATAC + RNA) to show MultiGATE can be applied to non-stereotypical tissues.

Overall, we evaluated MultiGATE on 7 datasets spanning five different spatial multi-omics platforms/technologies: Spatial ATAC–RNA, Slide-tags, Visium-CytAssist RNA–protein, SMA and SPOTS, covering both stereotypical tissues (brain and spleen) and a non-

stereotypical sample (breast cancer), and different molecular combinations: spatial ATAC+RNA, spatial RNA+protein/ADT, spatial RNA+metabolite.

b) *Multi-metric evaluation.*

When ground-truth tissue annotations are available, we report eight clustering metrics (ARI, RI, AMI, NMI, Homogeneity, Completeness, V-measure, and FMI); otherwise we use the combined intra-class correlation (ICC) to evaluate the clustering results. Regulatory accuracy is assessed by AUROC/AUPR for peak–gene, TF–peak links, against external orthogonal resource/data:

- *Cis-regulation*: comparison of predicted peak–gene links with human hippocampus eQTLs [8] and mouse EnhancerAtlas [9] & EGAS [10] enhancer–gene maps.
- *Trans-regulation*: evaluation of TF–peak links against SOX2 ChIP-seq peaks derived from adult brain tissue in Cistrome database [11].
- *Protein–gene interactions*: comparison of inferred protein–gene pairs with protein–gene pairs co-occurring in the same pathway in RaMP-DB [1].
- *Spatial clustering*: comparison of MultiGATE’s clusters on SPOTS spleen data with the protein-based deconvolution provided by the SPOTS [12] authors, and analysis of the spatial distribution of each identified cell type relative to germinal centers [12].

c) *A general cross-modality attention framework.*

Our attention mechanism flexibly models peak–gene, protein–gene and metabolite–gene relationships, enabling joint embedding of heterogeneous feature types and integration of spatial and molecular information across modalities.

4 Biological insights

a) **Human hippocampus.**

To demonstrate the biological relevance of regulatory interactions identified by MultiGATE, we analyzed a spatial ATAC–RNA-seq dataset from the adult human hippocampus and uncovered meaningful peak–gene associations. MultiGATE successfully linked

regulatory elements to genes such as *CA12* and *PRKD3*, which are involved in neuronal excitability and synaptic function, supported by hippocampus eQTLs. Beyond proximal interactions, MultiGATE identified long-range enhancer–gene links using HiChIP-augmented priors. Notably, it connected a distal enhancer 883.9 kb from *CAMTA1*, a hippocampus-enriched gene essential for neuronal survival, and another 507.9 kb from *CSNK1A1*, a kinase involved in neurodegenerative disease pathways. The interactions were supported by hippocampus-specific eQTLs and relevant GWAS signals, underscoring their potential functional relevance. MultiGATE also uncovered trans-regulatory TF–peak–gene relationships, exemplified by accurate prediction of *SOX2* binding sites in brain tissue, validated against ChIP-seq data. These findings demonstrate MultiGATE’s ability to capture both long-range and trans-regulatory mechanisms, offering biologically meaningful insights into hippocampal gene regulation.

b) P22 Mouse brain.

In the P22 mouse brain dataset, MultiGATE exhibits the ability to detect the genu of corpus callosum (ccg), lateral ventricle (VL), caudoputamen (CP), olfactory limb (aco), and lateral preoptic area (LPO), as well as accurately identifying the six layers present in the cortex. Beyond spatial clustering, regulatory attention scores identified a single distal peak (≈ 90 kb from *Xrcc5*) as the regulator of *Xrcc5*, and this peak overlaps a validated enhancer in EnhancerAtlas. Neighboring peaks show neither elevated attention nor enhancer support, underscoring the specificity of MultiGATE’s cis-regulatory inference.

c) Spleen SPOTS.

Through integrative analysis of protein expression (ADT), gene expression (RNA), and spatial localization, we accurately mapped three macrophage clusters to canonical macrophage subsets within the spleen: Red Pulp Macrophages (RPM), Marginal Zone Macrophages (MZM) and White Pulp Macrophages (WPM). These annotations were supported by distinct and consistent molecular signatures at both the protein and transcriptomic levels, along with their anatomically appropriate spatial distribution. This integrative approach demonstrates the biologically meaningful and interpretable results achieved through the multi-modal framework.

d) **Slide-tags Tumor.**

For the Slide-tags dataset generated from metastatic melanoma samples, MultiGATE accurately identified two major tumor clusters consistent with prior annotations and further distinguished them by distinct marker gene expression. One cluster exhibited a mesenchymal-like, immune-evasive state, associated with invasion, metastasis, and reduced MHC expression. In contrast, the other cluster displayed a melanocytic-like, immune-visible phenotype, characterized by pigmentation pathways and MITF activity. These molecular differences suggest divergent potential responses to immunotherapy, demonstrating the framework's ability to reveal tumor heterogeneity.

Comment 1

It is difficult to identify the framework's novelty based on the current manuscript, as the innovative aspects are not clearly highlighted in the main text. Meanwhile, the final output (is it embedding?) and downstream tasks presented in Figure 1 are not clear.

Response:

Thank you for pointing out that the novelty of our framework and the interpretation of the final outputs in Figure 1 were not emphasized clearly. We have revised the main text and Figure 1 to address these concerns, as detailed below.

Response to Question 1: It is difficult to identify the framework's novelty based on the current manuscript, as the innovative aspects are not clearly highlighted in the main text.

We thank the reviewer for this valuable feedback. We have listed the methodological innovation of MultiGATE explicitly in the above response to the general comment.

We added these points in the Method Overview (pp. 4) and Introduction (pp. 3), to guide the reader to better know MultiGATE's novelty.

The revised Introduction paragraph now reads:

"In response to the critical need for integrative tools in spatial multi-omics, we have developed MultiGATE, and its core is a two-level graph attention autoencoder designed to (i) infer cross-modality regulatory relationships (e.g., cis-regulation, trans-regulation, and protein–gene interactions) and (ii) embed spatial pixels into a latent space for clustering/spatial domain identification and visualization. By modeling regulatory links (such as peak-gene associations, protein-gene interactions, and enzyme-metabolite associations) directly into its graph attention mechanism, MultiGATE learns more informative low-dimensional representations of each spatial pixel/spot by deeper integrating different modalities. In turn, these refined embeddings sharpen the attention scores between cross-modality features, yielding more accurate inference of cross-modality regulation. In addition, the analysis of the cross-modality regulatory relationship enabled by MultiGATE provides the unique insight in studying transcriptional regulation in the native tissue context powered by spatial multi-omics data. We demonstrate the superior performance of MultiGATE through various spatial multi-omics datasets generated from different tissues and technologies. Our results highlight the ability of MultiGATE to effectively capture the regulatory relationships across different molecular modalities while providing enhanced latent embeddings of the pixels for accurate spatial clustering."

The revised Method Overview paragraph now reads:

"The core of MultiGATE is a two-level graph attention autoencoder that simultaneously embeds the pixels/spots in a low-dimensional space and models the cross-modality feature regulatory relationships (e.g., peak–gene, protein–gene and metabolite–gene). At the first level, a cross-modality attention mechanism is employed to model the cross-modality regulatory relationship; At the second level, a within-modality attention mechanism is

utilized to incorporate spatial information, which encourages the embedding of neighboring pixels to be similar. To further improve the cross-modality data integration, MultiGATE also incorporates a Contrastive Language-Image Pretraining (CLIP) loss, which encourages the alignment of the embeddings from different modalities."

Response to Question 2: Meanwhile, the final output (is it embedding?) and downstream tasks presented in Figure 1 are not clear.

We have revised the Overview Figure of MultiGATE (Fig. R6) and the end of Section "Method Overview" to clarify that the *final outputs* of MultiGATE are:

1. A low-dimensional *latent embedding* for each spatial pixel, which is used for clustering and spatial domain identification.
2. A matrix of *attention scores* between features (e.g., peak-gene pairs), which represent the strength of regulatory interactions.

Also, we have explicitly listed the *downstream analyses* enabled by these outputs:

- Spatial clustering/domain identification using latent embeddings extracted by MultiGATE
- Cis-regulatory inference using cross-modality attention scores
- Trans-regulatory inference using cross-modality attention scores
- Protein-gene interactions using cross-modality attention scores

The revised Method Overview paragraph now reads:

"The holistic structure of MultiGATE enables the aggregation and alignment of spatial information and molecular features from each modality. MultiGATE effectively extracts the low-dimensional representations of the pixels for clustering/spatial domain identification and data visualization, and simultaneously unravels the regulatory relationship of the cross-modality features for cis-regulation (e.g., enhancer-promoter interactions), trans-regulation

(e.g., transcription factor–enhancer–gene interactions) and protein–gene association (Fig. 1)."

Figure R6: Overview of MultiGATE.

MultiGATE is a two-level graph-attention auto-encoder designed for spatial multi-omics analysis. It extracts the latent embeddings of the pixels/spots in spatial multi-omics data, while simultaneously incorporating the regulatory relationship of the cross-modality features through the cross-modality attention mechanism and the spatial relationship of the pixels/spots through the within-modality attention mechanism. In addition to reconstruction loss, a CLIP contrastive loss aligns embeddings across modalities. MultiGATE yields (i) latent representations of pixels for clustering and visualization and (ii) cross-modality attention scores for cross-modality regulatory inference.

Comment 2

The method of using H&E images is not clear. Is it only to determine the radius of the neighbor when constructing a graph? In addition, the description of line 346 is vague, “When the pixel size decreases from 50 μm to 20 μm , spots become spatially denser, and we set the radius parameter d such that the numbers of neighbors increase from four to eight on average.” This claim needs data to support it.

Response:

Thank you for pointing out this point on defining spatial neighborhoods. We apologize for not making this clearer in the original manuscript.

MultiGATE does not use the H&E images to determine spatial neighborhood. In fact, the H&E images are not available for the human hippocampus and P22 mouse brain datasets. These datasets were both generated using Spatial-ATAC-RNA-seq [13], only bright field images taken by 10X objective (Thermo Fisher EVOS fl microscope) are available, where the nuclei are not stained. So we did not use H&E images to determine the neighborhood.

For each spot/pixel i with spatial coordinates (x_i, y_i) , we define its neighborhood $\mathcal{N}(i)$ as the set of its immediately adjacent neighbors, similar to BayesSpace [14]:

- Murine spleen data (SPOTS [12] technology):

Spots lie on a roughly hexagonal grid. We define each spot’s neighborhood to include exactly the six immediately adjacent spots—identical to the neighborhood definition used by BayesSpace [14] and other Visium-based tools (Fig. R7A).

- Spatial ATAC–RNA–seq datasets:

Both the adult human hippocampus and the P22 mouse brain datasets use the same Spatial ATAC–RNA–seq protocol, but differ in pixel size (50 μm vs. 20 μm):

- In the 50 μm human data, each pixel (other than the boundary pixels) has four neighbors (the four immediately adjacent pixels; Fig. R7B). When the neighbors change from 4 to 8 (four additional diagonals), the clustering results are similar (Fig. R8).

- In the 20 μm P22 mouse brain data, since pixel size is smaller than the adult human hippocampus data (50 μm), we used eight neighbors for each pixel (the full 3×3 pixel grid). A further increase in the number of neighbors for the mouse data would have captured a second ring of up to 24 neighbors, which could be too large (Fig. R7C).

We acknowledge that the sentence "When the pixel size decreases from 50 μm to 20 μm , spots become spatially denser, and we set the radius parameter d such that the numbers of neighbors increase from four to eight on average" is confusing and we have removed it in the revised manuscript. We have added a new Supplementary Fig. R7 showing the distribution of neighbor counts per spot/pixel across all datasets. Because boundary pixels have different number of neighbors, the average number of neighbors are around 6 for SPOTS data, 4 for 50 μm ATAC–RNA–seq data and 8 for 20 μm ATAC–RNA–seq data.

The revised manuscript now reads:

"The spatial neighborhood graph captures the similarity of neighboring spots using an adjacency matrix denoted as \mathbf{S} , where $S_{ij} = 1$ if spot j is an immediate spatial neighbor of spot i (and $S_{ii} = 1$ for all i). Neighborhoods are defined based on direct spatial adjacency [14], [15]. The spatial RNA+protein [12] technique uses the 10X Visium platform, each spot has six neighbors, following a hexagonal grid layout [14]. For Spatial ATAC–RNA–seq data, neighborhood size reflects pixel resolution: in the 50 μm human hippocampus dataset, each pixel has four adjacent neighbors (up/down/left/right), while in the 20 μm P22 mouse brain dataset, each pixel has eight neighbors in a 3×3 grid. The average number of neighbors per spot across datasets is visualized in Supplementary Note S6."

Figure R7: Definition of spatial neighbors.

Figure R8: Clustering of the human hippocampus dataset with 4 and 8 neighbors per pixel.

Comment 3

For the benchmarking on mouse brain data, the parameter settings for SpatialGlue and Seurat WNN appear to be missing from the Methods section. Could authors specify these parameters to ensure a fair comparison among approaches? Furthermore, regarding the attention mechanism, would authors please provide data or a discussion illustrating which modality carries more weight in identifying the mouse brain tissue architecture?

Response:

Response to Question 1: For the benchmarking on mouse brain data, the parameter settings for SpatialGlue and Seurat WNN appear to be missing from the Methods section. Could authors specify these parameters to ensure a fair comparison among approaches?

SpatialGlue parameters setting:

Following the official SpatialGlue tutorial (Tutorial 4: Data Integration for Mouse Brain Spatial-epigenome-transcriptome; available at <https://spatialglue-tutorials.readthedocs.io/en/latest/index.html>), we set the data type as:

```
data_type = 'Spatial-epigenome-transcriptome'
```

For this data type, we employed the following hyperparameters:

- Training epochs: 1600.
- Weight factors: [1, 5, 1, 1].

RNA data preprocessing involved:

- Filtering genes with `min_cells = 10` and cells with `min_genes = 200`.
- Selecting the top 3000 highly variable genes using the “`seurat_v3`” method.
- Normalizing to a target sum of $1e4$, followed by log-transformation and scaling.
- Reducing dimensionality via PCA with 50 components.

For the ATAC data, after subsetting to match the RNA observations, we applied LSI with 51 components. Finally, clustering was performed using the `mclust` algorithm with the number of clusters set to 18.

We strictly followed the analysis procedure and all the hyperparameter settings in the official SpatialGlue tutorial website.

Seurat WNN parameters setting:

We employed the default Seurat WNN pipeline (as implemented in Seurat v4). Specifically, we:

- Constructed a weighted nearest neighbor graph using the `FindMultiModalNeighbors` function based on the top 50 principal components from the RNA modality and principal components 2–50 from the ATAC modality (excluding the first PCA dimension from the ATAC modality, as it is typically correlated with sequencing depth), as recommended by the official Seurat WNN tutorial.
- Performed Louvain clustering on the resulting “wsnn” graph with a resolution parameter of 0.5, yielding 18 clusters.

These settings adhere to the recommendations in the Seurat documentation.

Manuscript revisions

We have updated the manuscript as follows:

1. Added a new subsection "Parameter Settings for Comparison Methods" to the Methods section (lines 560–565).
2. Added Supplementary Note S7 to present the parameter configurations used for benchmarking.

Response to Question 2: Furthermore, regarding the attention mechanism, would authors please provide data or a discussion illustrating which modality carries more weight in identifying the mouse brain tissue architecture?

Our MultiGATE incorporates two types of attention mechanisms: The first cross-modality attention mechanism infers cross-modality regulatory relationships; The second within-modality attention mechanism is utilized to incorporate spatial information, which encourages the embedding of neighboring pixels to be similar. MultiGATE does not, by design, report an explicit “modality weight” for each pixel or cluster. Nonetheless, we can gauge the relative influence of the ATAC and RNA modalities by comparing the MultiGATE’s clustering results

(which uses both modalities) with STAGATE's [16] clustering results on each modality alone (Fig. R9).

- In the region highlighted by the *red box*, the MultiGATE spatial domains are more similar to the ATAC-only clustering results (Fig. R9B). This close correspondence suggests that, in this anatomical region, chromatin accessibility carries more weight in identifying the mouse brain tissue architecture.
- Conversely, in the region marked by the *black box*, the MultiGATE result is more similar to the RNA-only clustering, indicating that transcriptomic information carries more weight in identifying the mouse brain tissue architecture.

These observations illustrate that MultiGATE adaptively leverages information from different molecular modalities.

Figure R9: P22 mouse brain spatial clustering results.

A Spatial clustering results of MultiGATE using both ATAC and RNA.

B Spatial clustering results using ATAC information only by STAGATE.

C Spatial clustering results using RNA information only by STAGATE.

Comment 4

The description of how the attention score is applied to predict peak–gene associations is unclear. Why did the authors choose not to validate these predictions using peak–motif–TF and gene relationships? Additionally, the insights from Figures C and D are not evident. Specifically, what is the innovation or importance of identifying RPS27L as a regulator, and how does it contribute to the study’s significance? Presenting more substantial biological examples would help demonstrate the practical value of this approach.

Response:

Response to Question 1: The description of how the attention score is applied to predict peak–gene associations is unclear.

We thank the reviewer for pointing this out. In the revised manuscript (Methods, “MultiGATE attention-based *cis*-regulatory inference and validation”), we have expanded the description and now provide a detailed description of how raw attention scores are transformed into high-confidence peak–gene links:

1. Cross-modality attention computation.

In the cross-modality autoencoder, for each peak f and gene g we first compute an unnormalized attention coefficient

$$e_{fg} = \text{Sigmoid}(\mathbf{v}_1^T \mathbf{X}_{(f)}^T + \mathbf{v}_2^T \mathbf{X}_{(g)}^T),$$

where $\mathbf{X}_{(f)}^T$ and $\mathbf{X}_{(g)}^T$ are the observed data for peak f and gene g , and $\mathbf{v}_1, \mathbf{v}_2$ are learned vectors.

We then incorporate a genomic-distance prior

$$A_{fg} = \exp\left(\left(\frac{\text{dist}_{fg} + \text{bp_width}}{\text{bp_width}}\right)^{-0.75}\right) \quad (\text{bp_width} = 400 \text{ bp}),$$

and normalize e_{fg} to obtain the cross-modality attention score

$$\text{att}_{fg} = \frac{A_{fg} \exp(e_{fg})}{\sum_{h \in \mathcal{N}_f} A_{fh} \exp(e_{fh})},$$

where \mathcal{N}_f is the set of features connected to peak f in the feature-connectivity graph.

2. Rescaling and thresholding.

First, we pooled all cross-modality attention scores (i.e., excluding self-attention) and performed a linear transformation to map them onto the interval $[0, 1]$. The resulting distribution (Fig. R10) deviated from unimodality, suggesting two components. To quantify this, we fitted one- and two-component Gaussian mixture models (GMMs) and compared their fits using both a likelihood-ratio test (LRT, $p < 10^{-16}$) and the Bayesian Information Criterion ($\Delta\text{BIC} \geq 1.9 \times 10^4$). In both human and mouse datasets, these criteria favored the two-component mixture model (Table 3).

Within this two-component mixture model, we define our threshold θ as the intersection point of the two Gaussian densities (Fig. R10). This data-driven cutoff separates low, noise-driven attention scores from high, biologically meaningful ones. Empirically, we obtained

$$\theta_{\text{Human hippocampus}} = 0.204, \quad \theta_{\text{Mouse P22}} = 0.141.$$

These thresholds were then used to call peak–gene links.

Table 3: Mixture–model statistics for rescaled attention scores

Dataset	$\Delta\text{BIC} (1 \rightarrow 2)$	LRT p -value	Threshold θ
Human	19 729	$< 10^{-16}$	0.204
Mouse P22	32 455	$< 10^{-16}$	0.141

Figure R10: Gaussian mixture model (GMM) density plots for rescaled attention scores. Right: Mouse P22 brain dataset with identified threshold at 0.14. Left: Human hippocampus dataset with identified threshold at 0.20. In both cases, the data density (black), GMM components (orange and green), and the identified threshold (vertical dashed line) are shown.

3. Calling peak–gene links.

Any peak–gene pair whose rescaled attention exceeds this threshold is reported as a *cis*-regulatory interaction. We then validate these predictions against external resources:

- eQTL [8]. A pair is supported if an eQTL locus for the gene falls within the peak region.
- EnhancerAtlas [9] or EGAS [10]. A pair is supported if the peak overlaps a regulatory region annotated for that gene.

This expanded description makes explicit how MultiGATE converts raw cross-modality attention outputs into a final set of peak–gene associations. And we have added it into the revised manuscript (Methods, “MultiGATE attention-based *cis*-regulatory inference and validation”).

The following presents the updated portion of the Methods in the revised manuscript:

"We use the following procedure to get peak-gene regulatory pairs from MultiGATE’s cross-modality attention scores and evaluate the peak-gene regulatory pairs:

1. *Cross-modality attention computation.*

In the cross-modality autoencoder, for each peak f and gene g we first compute an unnormalized attention coefficient

$$e_{fg} = \text{Sigmoid}(\mathbf{v}_1^T \mathbf{X}_{(f)}^T + \mathbf{v}_2^T \mathbf{X}_{(g)}^T),$$

where $\mathbf{X}_{(f)}^T$ and $\mathbf{X}_{(g)}^T$ are the observed data for peak f and gene g , and $\mathbf{v}_1, \mathbf{v}_2$ are learned vectors.

We then incorporate a genomic-distance prior

$$A_{fg} = \exp\left(\left(\frac{\text{dist}_{fg} + \text{bp_width}}{\text{bp_width}}\right)^{-0.75}\right) \quad (\text{bp_width} = 400 \text{ bp}),$$

and normalize e_{fg} to obtain the cross-modality attention score

$$\text{att}_{fg} = \frac{A_{fg} \exp(e_{fg})}{\sum_{h \in \mathcal{N}_f} A_{fh} \exp(e_{fh})},$$

where \mathcal{N}_f is the set of features connected to peak f in the feature-connectivity graph.

2. Rescaling and thresholding.

First, we pooled all inter-feature attention scores (i.e., excluding self-attention) and performed a linear transformation to map them onto the interval $[0, 1]$. The resulting distribution was bimodal (Supplementary Note S5). We fitted a two-component Gaussian mixture model to these normalized scores and defined our threshold θ as the intersection point of the two Gaussian densities—i.e., the value at which the posterior probability of belonging to either component is equal (Supplementary Note S5). This data-driven cutoff separates low, noise-driven attention scores from high, biologically meaningful ones. Empirically, we obtained

$$\theta_{\text{Human hippocampus}} = 0.204, \quad \theta_{\text{Mouse P22}} = 0.141.$$

These thresholds were then used to call peak–gene links.

To validate the peak-gene regulatory pairs identified in the human hippocampus data, we used the human hippocampus eQTL data [8]. To validate the peak-gene regulatory pairs identified in the mouse brain data, we used the Enhancer Atlas [9] and EGAS [10] data obtained from the mouse cortex and striatum: The Enhancer Atlas obtains regulatory information by predicting consensus enhancers based on high-throughput datasets, including histone modification, CAGE, GRO-seq, transcription factor binding, and DHS; The EGAS data utilizes single-cell sequencing data and a non-parametric permutation-based procedure to predict the genome-wide enhancer-gene associations. However, mouse eQTL data was not employed due to the absence of the corresponding tissues of interest.

A peak-gene pair is considered to be supported by Enhancer Atlas or EGAS if the peak overlaps with a genomic region that regulates the corresponding gene. A peak-gene pair is considered to be supported by eQTL if there is an eQTL locus within the peak and the locus is associated with the corresponding gene."

Response to Question 2: Why did the authors choose not to validate these predictions using peak–motif–TF and gene relationships? Additionally, the insights from Figures C and D are not evident. Specifically, what is the innovation or importance of identifying RPS27L as a regulator, and how does it contribute to the study’s significance? Presenting more substantial biological examples would help demonstrate the practical value of this approach.

On validation strategy and use of motifs

We thank the reviewer for the insightful comment. In the original version of MultiGATE, the attention mechanism was primarily designed to capture cis-regulatory interactions, modeling the influence of proximal regulatory elements (defined by spatially resolved ATAC-seq peaks) on gene expression. To validate these cis-regulatory predictions, Figures 2C and 2D leveraged independent eQTL data. Specifically, Figure 2C shows that peak–gene pairs supported by eQTLs

received significantly higher attention scores from MultiGATE, while Figure 2D demonstrates that MultiGATE outperforms baseline methods in identifying eQTL-associated regulatory interactions, as reflected by ROC curves. Together, these results suggest that MultiGATE effectively prioritizes biologically meaningful cis-regulatory relationships.

We did not use peak–motif–TF–gene relationships to validate these cis-regulatory links because individual peaks often contain multiple transcription factor binding motifs, and each motif may occur frequently across the regulatory elements. This high redundancy and lack of specificity make it difficult to confidently assign a particular transcription factor to a given peak. As a result, motif-based validation is not a reliable strategy in this setting.

In the revised manuscript, we have extended MultiGATE to capture trans-regulatory and long-range regulatory interactions, including transcription factor–peak–gene relationships and distal enhancer–promoter interactions, supported by HiChIP data. These enhancements broaden the model’s regulatory inference capabilities and improve the biological interpretability of the results in both cis- and trans-regulatory contexts.

Extension to trans-regulation

To extend MultiGATE toward trans-regulation, we modified the model by introducing a TF binding potential (TFBP) as an additional prior in the cross-modality autoencoder. Formally, for a given TF k and peak i , the prior is defined as:

$$\text{TFBP}_{k,i} = M_{k,i} \times X_i \quad (5)$$

where $M_{k,i} \in \{0, 1\}$ is the binarized motif binding score for TF k on peak i . The motifs were scanned using HOMER [17], with motif-to-TF mapping curated from [18]. It is set to 1 if at least one binding site for TF k is found in peak i , and 0 otherwise. X_i denotes the average chromatin accessibility of peak i across all spots. This score integrates both motif presence and chromatin openness, serving as a biologically informed prior for TF binding potential. The original peak–gene (excluding the TFs) attention mechanism remains unchanged, allowing MultiGATE to simultaneously learn both cis and trans regulations within a unified attention framework. To quantify predicted TF binding to peaks, we define the TF–peak integrated

attention score as:

$$S_{k,i} = \alpha_{k,i} \times \sum_{j \in \mathcal{G}(i)} Att_{i,j} \quad (6)$$

where $\alpha_{k,i}$ is the learned attention score from TF k to peak i , $Att_{i,j}$ is the attention score from peak i to gene j , and $\mathcal{G}(i)$ is the set of genes linked to peak i . This metric prioritizes TF–peak pairs that are both accessible and functionally linked to gene expression.

We tested this framework using the spatial ATAC–RNA-seq dataset from the adult human hippocampus and evaluated the predictions for SOX2, a well-studied brain-specific TF. The ChIP-seq data for SOX2 in brain tissue were downloaded from the Cistrome database [11] and served as the ground truth for evaluating predicted TF binding peaks. We compared our model against several baseline methods: The motif-only baseline was constructed using the binary motif TF binding score (defined as M in Eq. (5)) identified via HOMER [17], with motif-to-TF mapping curated from [18]; The TF binding potential baseline was defined as the product of binary motif binding and chromatin openness (defined as TFBP in Eq. (5)); The cosine similarity baseline was computed as the cosine similarity between the expression profile of the TF (from spatial RNA data) and the accessibility profile of each peak (from spatial ATAC data) across all the spots.

Our method’s learned attention scores (defined as $S_{k,i}$ in Eq. (6)) were highly predictive of TF binding events, with an AUC of 0.8669 and AUPR of 0.4906, substantially outperforming motif-only (AUC = 0.4906, AUPR = 0.1934), TF binding potential baseline (AUC = 0.6326, AUPR = 0.2845), and cosine similarity between TFs and peaks (AUC = 0.6280, AUPR = 0.2752), as shown in Fig. R11.

However, we acknowledge that large-scale validation of trans-regulation remains challenging due to the limited availability of high-quality ChIP-seq datasets in brain tissues such as the hippocampus. While we demonstrate the potential of our approach using SOX2, a brain-specific TF with reliable ChIP-seq data, broader validation across multiple TFs is currently constrained. We consider this an important direction for future work and plan to incorporate broader validation using emerging tissue-specific datasets. We have clarified this process in the result and discussion in the revised manuscript accordingly.

Figure R11: Comparison of methods for predicting SOX2 binding in the adult human hippocampus. Motif_Binding denotes the binary motif TF binding score. TFBinding_Potential is computed as the product of binary motif binding score and chromatin openness (defined as TFBP in Eq. (5)). The Cosine_TF_Peak is computed as the cosine similarity between the expression profile of the TF and the accessibility profile of each peak across all the cells. Attention_Score demonstrates the TF–peak integrated attention score defined as $S_{k,i}$ in Eq. (6).

On the role of RPS27L and biological examples

We also thank the reviewer for the insightful comment regarding the biological significance of identifying RPS27L and for suggesting that more substantial examples would strengthen the manuscript. While RPS27L relates to neuronal stress during aging, we agree it may not be the most striking example. To address this, we further analyzed genes such as CA12 and PRKD3, which have been implicated in hippocampal function and synaptic regulation. MultiGATE successfully identified peak–gene interactions for these targets that are supported by hippocampus eQTL data, highlighting its capability to recover meaningful regulatory associations.

Revised paragraph (lines 139-146):

"To further showcase the biological relevance of the peak-gene interactions identified by MultiGATE, we examined genes with well-established functional importance in the hippocampus, including CA12 and PRKD3. CA12 is a carbonic anhydrase in hippocampal neurons, regulates pH and supports neuronal excitability [19], [20]. PRKD3 is associated with synaptic function and Alzheimer's disease [21], [22]. For each of these genes, MultiGATE successfully identified peak-gene associations that are supported by hippocampus-specific eQTLs (Fig. 2E), whereas non-significant associations are generally not supported by eQTL signals."

Long-range enhancer–gene interactions and biological examples

To demonstrate MultiGATE's ability to detect long-range interactions and gain more biological insight of the distal gene regulation, we reanalyzed the adult human hippocampus Spatial ATAC–RNA-seq dataset. Under the 150 kb genomic distance prior alone, we obtained 16,347 short-range peak–gene candidates. We then augmented this prior by adding 205,627 enhancer–promoter contacts curated from HiChIP experiments database [23] across 24 diverse human tissues (Table 4). Specifically, any ATAC peak overlapping a HiChIP-annotated enhancer

that loops to a gene promoter was introduced as a candidate edge in the peak-gene prior network, regardless of genomic distance.

After retraining MultiGATE with these augmented connections, we stratified all candidate edges into eight distance bins (0–150 kb, 150–300 kb, . . . , >1.25 Mb) and compared the attention scores assigned to brain-specific versus non-brain HiChIP loops in each bin. In every distance category, brain-specific HiChIP loops received significantly higher attention scores than other loops (Mann–Whitney U test, $P < 1 \times 10^{-3}$; Fig. R12), suggesting that MultiGATE successfully prioritizes tissue-relevant distal regulation.

Notably, the enhancer chr1:7669258-7670119 and its target gene CAMTA1, located 883.9 kb apart, were identified by MultiGATE with an attention score of 0.2194 and is supported by the eQTL pair chr1_7669531_G_C - CAMTA1. CAMTA1, a transcription factor enriched in the hippocampus, is crucial for Purkinje cell survival and cerebellar function, with its loss leading to ataxia, motor deficits, and neurodegeneration due to dysregulated neuronal gene expression [24]. Additionally, MultiGATE identified another enhancer at chr5:150000116-150001000 and its target gene CSNK1A1, with a distance of 507.9 kb. This enhancer-gene pair is supported by the hippocampus eQTL chr5_150000595_C_T - CSNK1A1, along with a risk SNP (rs4705403, $p=1.00e-8$) associated with migraine in GWAS study. CK1 α plays a crucial role in the pathogenesis of Alzheimer’s and Parkinson’s diseases by regulating key proteins involved in neurodegeneration [25]. These examples illustrate how MultiGATE uncovers functionally important regulatory connections, including both long-range chromatin loops and trans-regulatory TF-peak-gene interactions that are missed by previous models.

Manuscript revisions

We have revised the manuscript to:

1. Add the new biological examples in the Results section (e.g., CA12, PRKD3; lines 139-143).
2. Expand on the implications of trans-regulation (lines 149–154 in Results; lines 333-337 in Discussion; Supplementary Notes S2) and long-range interactions modeling (lines 146–149 in Results; Supplementary Notes S4).
3. Update Figure 2E with more biological examples.

Table 4: Number of peak–gene pairs per tissue (FitHiChIP, 5 kb resolution).

Tissue	Candidate peak-gene pairs
Aorta	5399
Blood	53084
Brain	10644
Breast	18694
Colon	765
Embryo	8958
Endometrioid endometrial tumor	51
Endometrium	11
Esophagus	16026
Eye	1062
Heart	8221
Kidney	3
Lung	14744
Lymph node	1163
Lymphocyte	28106
Muscle	1616
Ovary	1714
Prostate	227
Skin	26163
Stem cell	44
Stomach	5675
Thyroid	2666
Uterus	591

Figure R12: Tissue-specific trends in gene–peak attention scores across genomic distances. This line plot shows median attention scores for gene–peak pairs in Brain (red) and Other Tissues (blue) across eight distance bins (0–150kb to >1250kb). Shaded regions indicate 95% confidence intervals around each median. Statistical significance between Brain and Other Tissues was assessed using Mann–Whitney U tests, with asterisks above each bin denoting significance levels ($*p < 0.05$, $**p < 0.01$, $***p < 0.001$). “ES” marks the effect size.

Figure R13: MultiGATE enables precise detection of spatial domains and authentic integrative regulatory inference in the human hippocampus.

A Brightfield image and manually annotated segmentation of hippocampus layers and white matter (WM) in the human hippocampus.

B Spatial clustering of hippocampal regions using MultiGATE, SpatialGlue, and Seurat WNN. Clustering performance is assessed using the Adjusted Rand Index (ARI), with higher values indicating greater clustering accuracy.

C Boxplots representing attention scores for peak-gene pairs across different genomic distances, grouped based on whether they are supported by expression quantitative trait loci (eQTL) evidence.

D Receiver operating characteristic (ROC) curves comparing the performance of MultiGATE and other methods in predicting eQTL-associated regulatory interactions.

E Visualization of MultiGATE-predicted cis-regulatory interactions for the target gene CA12 and PRKD3 along with eQTL evidence.

Comment 5

It is hard to disguise the overall performance between multiGATE and SpatialGlue on tissue architecture prediction; the quantitative score is needed. In addition, why do those three clustering results share the same color legend, which is unclear to me. In line 151, “SpatialGlue misassigned pixels in the inner layer to the same cluster as the outermost layer” is not convincing and confuses the pixels with spots.

Response:

Response to Question 1: It is hard to disguise the overall performance between multiGATE and SpatialGlue on tissue architecture prediction; the quantitative score is needed.

We thank the reviewer’s important comment. In the revised manuscript, we introduced multiple quantitative metrics to enable a more rigorous comparison between MultiGATE, SpatialGlue and Seurat WNN for tissue architecture prediction. In particular, for the two datasets with ground-truth labels, the adult human hippocampus dataset (Spatial ATAC-RNA-seq) and the mouse brain dataset (spatial transcriptomics and metabolomics), we computed the Rand Index, Adjusted Rand Index, Adjusted Mutual Information, Normalized Mutual Information, Homogeneity, Completeness, V-measure, and the Fowlkes–Mallows Index. As shown in Fig. R14A and Fig. R14B, MultiGATE achieves superior performance relative to SpatialGlue and Seurat WNN across all these metrics in both datasets.

For the two datasets without ground-truth labels, the P22 mouse brain dataset (Spatial ATAC-RNA-seq) and the mouse spleen dataset (SPOTS), we calculated the Intraclass Correlation Coefficient (ICC) [26] to evaluate the clustering performance. The ICC measures the consistency of observations within clusters. A higher ICC value indicates increased homogeneity within clusters for a given modality, suggesting an improvement in clustering quality[27]. Again, MultiGATE outperforms both SpatialGlue and Seurat WNN. (Figs. R14 C,D).

Intraclass Correlation Coefficient (ICC).

For each cluster i in modality k , within-cluster agreement of the latent features (top 50 principal components of the pre-processed data for ATAC + RNA, UMAP coordinates for

RNA + protein) is quantified by

$$\text{ICC}_i^{(k)} = \frac{\sigma_m^2}{\sigma_i^2 + \sigma_m^2},$$

where σ_i^2 denotes the within-cluster variance among observations belonging to cluster i and σ_m^2 is the variance of the corresponding cluster means across all clusters in modality k . The statistic takes values in $[0, 1]$, with larger values indicating greater homogeneity (i.e. more coherent clusters) for that modality. For every method–modality combination, $\text{ICC}_i^{(k)}$ is computed for all clusters.

Manuscript Revisions

We have updated the manuscript as follows:

1. Added a new subsection “Quantitative Metrics Calculation” to the Methods (lines 566–578).
2. Included detailed calculation procedures in Supplementary Note S9.
3. Added the results of quantitative metrics to Extended Data Figure 6.

Figure R14: Quantitative Evaluation of MultiGATE, SpatialGlue and Seurat WNN for clustering. MultiGATE outperforms alternative approaches (SpatialGlue and Seurat WNN) on every metric across 4 datasets.

A Bar plot of clustering metrics (Rand Index, Adjusted Rand Index, Adjusted Mutual Information, Normalized Mutual Information, Homogeneity, Completeness, V-measure, and Fowlkes–Mallows Index) for the adult human hippocampus dataset.

B Bar plot of the same clustering metrics for the mouse brain dataset.

C Bar plot of the combined ICC (ATAC+RNA) values across methods in the P22 mouse brain dataset, with MultiGATE achieving the highest score.

D Bar plot of the combined ICC (RNA+Protein) values across methods in the mouse spleen dataset.

Response to Question 2: why do those three clustering results share the same color legend, which is unclear to me.

Regarding those three clustering results share the same color legend, we used a unified color scheme across the three clustering methods (MultiGATE, SpatialGlue, and Seurat WNN) to facilitate direct visual comparisons of spatial patterns (Fig. R15). Each method naturally produces integer labels for its clusters (e.g., 1–7 in the hippocampus dataset, Fig. R15A shows the spatial clustering results for MultiGATE, and Fig. R15C show the results for SpatialGlue); however, it is challenging to compare the clustering results between the two methods if similar spatial domains are assigned different colors. Therefore, we employed the Hungarian algorithm to determine the optimal one-to-one mapping between cluster labels from different methods, maximizing the overlap of spots between the clusters that are mapped.

Once these mappings were established, each integer label was assigned a unique color. For instance, if cluster 1 from MultiGATE maps to cluster 4 from SpatialGlue, both cluster panels were visualized using the same color (Figs. R15 A and B). This approach enables readers to immediately discern regions that are consistently classified across methods. We only shuffle the indexes of cluster labels; we do not change the clustering results, and we have added an explanation in the revised manuscript (lines 117-119) and expanded in Supplementary Note S10.

Figure R15: Uniform Color Matching for Comparative Clustering Analysis.

We only shuffle the indexes of cluster labels; we do not change the clustering results.

A MultiGATE clustering results.

B SpatialGlue clustering results with colors matched to MultiGATE clustering results.

C SpatialGlue clustering results.

Response to Question 3: In line 151, “SpatialGlue misassigned pixels in the inner layer to the same cluster as the outermost layer” is not convincing and confuses the pixels with spots.

We thank the reviewer for raising this important point concerning our terminology. In the Spatial ATAC–RNA-seq dataset of Zhang et al. [13], each *pixel* denotes a spatially resolved unit in which both chromatin accessibility (ATAC) and gene expression (RNA) are jointly profiled—i.e., one pixel corresponds to one data point. Our usage aligns with the authors’ original definition in [13].

For the sentence in line 151, we were referring to cluster 5 in SpatialGlue. To clarify, the pixels in SpatialGlue cluster 5 may contain a heterogeneous mixture of multiple cell types. The following are the analyses to substantiate this statement.

1. Global marker analysis: SpatialGlue Cluster 5 vs. MultiGATE Cluster 5

- **SpatialGlue Cluster 5** shows no cluster-enriched markers. Differential expression analysis (Wilcoxon test, BH-adjusted $p > 0.05$ for all genes) failed to identify any

significant DEGs that are enriched in cluster 5, indicating a heterogeneous mixture of multiple cell types (Fig. R17 left).

- **MultiGATE Cluster 5**, in contrast, yields biologically meaningful markers (Fig. R17 center), including:

- *Myh11* (smooth muscle myosin heavy chain 11; \log_2 FC \approx 3.24, BH-adjusted $p < 1 \times 10^{-7}$)
- *Cald1* (high-molecular-weight caldesmon; \log_2 FC \approx 1.3, BH-adjusted $p < 3 \times 10^{-6}$)
- *Mylk* (smooth muscle myosin light chain kinase; \log_2 FC \approx 2.5, BH-adjusted $p < 1 \times 10^{-4}$)

MYH11 is the hallmark marker of differentiated smooth muscle cells (SMCs) [28], [29], CALD1 is restricted to fully differentiated SMCs and absent in myofibroblasts or pericytes [30], [31], and MYLK encodes a Ca^{2+} /calmodulin-dependent kinase essential for smooth muscle contraction [32]. The co-expression of these three genes indicates that MultiGATE Cluster 5 represents arterial vascular smooth muscle cells.

2. Local analysis in the LS nucleus region

- **MultiGATE Cluster13 represents lateral septal (LS) nucleus.**

The region of MultiGATE cluster 13 (bottom left cluster) was annotated as lateral septal (LS) nucleus in the postnatal day 22 (P22) mouse brain [13] (Fig. R16A). Differential expression of cluster 13 identified *Zic1* (\log_2 FC \approx 2.4, FDR $< 1 \times 10^{-10}$) and *Zic4* (\log_2 FC \approx 1.9, FDR $< 1 \times 10^{-10}$; Fig. R17 right). Orthogonal datasets support that *Zic1* and *Zic4* are marker genes of lateral septal (LS) nucleus: in a septum-focused snRNA-seq and MERFISH atlas [33], *Zic1* and *Zic4* are the most enriched transcripts in LS cells at P21, with expression confined to the LS; similarly, Bgee v15.2 reports a tissue-specificity score of 99.91 for *Zic1* and 74.11 for *Zic4* in the LS [34]. Developmental lineage studies further confirm that *Zic*-expressing progenitors give rise predominantly to LS neurons [35]–[38].

- **The bottom left pixels are misassigned to cluster 5 in SpatialGlue.**

Pixels that SpatialGlue assigned to SpatialGlue Cluster 5 still exhibit LS-specific *Zic1/4* expression indistinguishable from MultiGATE Cluster 13 pixels (Fig. R16B), which is different from other SpatialGlue Cluster 5 pixels' expression level, demonstrating that SpatialGlue misassigns some pixels in the LS nucleus region to SpatialGlue Cluster 5 (Fig. R16B).

Together, these results show that Cluster 5 in SpatialGlue shows no cluster-enriched markers (Fig. R17 left) and combines pixels from multiple cell types.

To clarify the specific misassignment by SpatialGlue, we have revised the sentence as follows:

"The cluster 5 in SpatialGlue shows no cluster-enriched markers (Supplementary Note S11), indicating a heterogeneous mixture of multiple cell types. More specifically, the cluster 5 in SpatialGlue contains pixels in the LS nucleus region, which are molecularly different from the other pixels in cluster 5 (Supplementary Note S11)."

Figure R16: Further analysis of the clustering results of MultiGATE and SpatialGlue.

A Spatial maps of cluster assignments by MultiGATE (left) and SpatialGlue (right). Colors denote clusters; the black rectangle on the SpatialGlue plot highlights lateral septal (LS) pixels misassigned to cluster 5.

B Violin plots of Zic1, Zic4, Trpc4, and Cacna2d2 expression. Comparison across MultiGATE Cluster 13, SpatialGlue Cluster 5 in the LS region, and SpatialGlue Cluster 5 elsewhere shows SpatialGlue’s erroneous split of the LS nucleus versus MultiGATE’s coherent clustering.

Figure R17: Volcano plot for MultiGATE Cluster 5, SpatialGlue Cluster 5 and MultiGATE Cluster 13.

Comment 6

The biology story and evidence in Figure 4 are not convincing and come across as somewhat confusing. First, SPOT’s measure is at the cellular level, which explains why the authors label tissue structures by cell types. However, it would be advisable to incorporate a spatial deconvolution step rather than making cell-level conclusions without single-cell resolution. Additionally, it is unclear what “Macrophages I, II, and III” (i.e., not typical Macrophage name) specifically refer to from a biological standpoint. Figure 4C should include a metric or value to indicate the significance level. Lastly, the way of presenting Figure 4E heatmap is weak in illustrating the variation in protein intensities for each cluster.

Response:

Response to Question 1: First, SPOT’s measure is at the cellular level, which explains why the authors label tissue structures by cell types. However, it would be advisable to incorporate a spatial deconvolution step rather than making cell-level conclusions without single-cell resolution.

We thank the reviewer for raising this important point regarding incorporating a spatial deconvolution step. We agree that spatial platforms like SPOTS provide multi-cellular

measurements in each pixel/spot, and that any downstream interpretation of cell-type composition or tissue structure should consider this resolution limitation of the SPOTS technology [12].

Spatial deconvolution methods (CARD [39], RCTD [40], Cell2location [41], Stereoscope [42], SPOTlight [43], DSTG [44], SpatialDWLS [45], SpatialDeX [46], and STdeconvolve [47]) typically focus on cell type deconvolution and estimate cell-type proportions for each pixel/spot; these estimated cell-type proportions can not be taken as input by MultiGATE or SpatialGlue. MultiGATE and SpatialGlue take the raw molecular features or principal components of raw molecular features as input.

Furthermore, existing spatial deconvolution methods have been primarily developed for single-modality spatial transcriptomics data. Reference-based methods such as CARD [39], RCTD [40], Cell2location [41], Stereoscope [42], SPOTlight [43], DSTG [44] and SpatialDWLS [45] require a single-cell RNA reference data and are designed for transcriptomic data; Reference-free methods such as SpatialDeX [46] and STdeconvolve [47] also operate on RNA modality only. To the best of our knowledge, there are currently no deconvolution methods that are designed for spatial multi-omics data, which combines transcriptomics with other modalities such as protein or chromatin accessibility.

Nevertheless, to demonstrate that MultiGATE's clusters align with deconvolution results, we compared MultiGATE's cluster results with the protein-based deconvolution results provided by the original SPOTS authors [12]. The SPOTS authors only provided the deconvolution results based on protein modality; no RNA-based deconvolution results are available in the SPOTS spleen dataset. The cell subtypes defined by the MultiGATE clustering and protein-based deconvolution results are different, so we merge into three broad cell types (T cell enriched, B cell enriched and Macrophage enriched) as shown in Table 5 and Table 6.

We then compared the distribution of each deconvolution cell type proportion across MultiGATE's clusters (Fig. R18). In each case, the corresponding deconvolution cell type proportion is significantly higher in the matching cluster (Wilcoxon $p < 10^{-5}$ for all pairwise tests), confirming that MultiGATE's T cell, B cell, and macrophage clusters recover the same coarse cell-type enrichments as protein-only deconvolution provided by the SPOTS authors.

Table 5: Mapping of MultiGATE subclusters to broad cluster categories.

Original Subcluster	Merged Category
Red Pulp Macrophages (RPM)	Macrophage enriched
White Pulp Macrophages (WPM)	Macrophage enriched
Marginal Zone Macrophages (MZM)	Macrophage enriched

Table 6: Mapping of SPOTS author-provided protein-based deconvolution results to broad cluster categories.

Original Subcluster	Merged Category
CD4 T-cells (%)	T cell enriched
CD8 T-cells (%)	T cell enriched
GCB naïve/activated (%)	B cell enriched
Mature B (%)	B cell enriched
Antigen-presenting Macs (%)	Macrophage enriched
Red pulp Macs (scavenging) (%)	Macrophage enriched
Monocytes (%)	Macrophage enriched

We note that the deconvolution results based only on the protein modality may not be accurate. We also validated MultiGATE’s clustering results using other independent data/information: marker genes/proteins of each cluster (Fig. R19A and Fig. R20), and the spatial distribution of each cluster relative to germinal centers (GCs) (Fig. R19B). Details about the validation of MultiGATE’s clustering results are in the response below.

Manuscript revisions

We have added a comparison of MultiGATE’s clustering results with protein-based deconvolution outcomes in Supplementary Note S13 (see also main text, lines 264–265).

Figure R18: Comparison of MultiGATE clustering results with deconvolution-derived cell-type proportions provided by the SPOTS authors [12]. MultiGATE-defined clusters show significantly higher enrichment for the expected cell types (T cells, B cells, and macrophages).

Response to Question 2: Additionally, it is unclear what “Macrophages I, II, and III” (i.e., not typical Macrophage name) specifically refer to from a biological standpoint.

We thank the reviewer for pointing out the need to clarify the biological identities of Macrophages I, II, and III. We have mapped these 3 clusters to well-established macrophage subtypes based on integrative analysis of protein expression, gene expression, and spatial localization, as detailed below.

Macrophage I has been identified as Red Pulp Macrophages (RPM) due to its high protein expression of F4-80 and CD163 (Fig. R20), along with elevated RNA levels of erythroid and hemoglobin-related genes such as *Hbb-bt* and *Hba-a1* (Fig. R19A). These markers are characteristic of RPMs, which specialize in clearing senescent red blood cells and processing hemoglobin [48].

Macrophage II corresponds to White Pulp Macrophages (WPM), based on increased protein expression of F4-80 and CD68 (Fig. R20), which are characteristic markers of white pulp macrophages [49], [50]. Although canonical WPM markers were not strongly enriched at the RNA level, the protein data, together with their spatial localization near germinal centers (Fig. R19B), support their annotation as WPMs. This is consistent with WPM's function as antigen-presenting cells within the T cell-rich zone of the white pulp [49]–[51]. The use of spatial information was essential for distinguishing this population from others with overlapping marker profiles.

Macrophage III has been annotated as Marginal Zone Macrophages (MZM), given its strong protein expression of CD169 (*Siglec1*) (Fig. R20) and RNA expression of *Marco* (Fig. R19A), both hallmark markers of MZMs involved in capturing blood-borne pathogens in the marginal zone [50], [52].

To further validate these annotations, we analyzed the spatial distribution of these macrophage populations relative to germinal centers (GCs). As shown in Fig. R19B, the spatial distances of RPM, WPM, and MZM are concordant with their known anatomical locations in the spleen: WPM are localized closest to the GCs, followed by MZM in the marginal zone, and RPM situated furthest in the red pulp (Fig. R19B). This spatial arrangement further supports the biological annotation of these macrophage subsets [51].

A Marker gene expression

B Spatial Distribution of Cell Populations

Figure R19: Further analysis of cell types in Spleen dataset.

A Marker gene expression in Spleen dataset.

B Spatial distribution of different cell types in Spleen dataset.

Protein Z-score by Cluster

Figure R20: ADT (Antibody-Derived Tags) signatures associated with each spatial cluster, revealing distinct immune cell identities.

We revised the manuscript to replace "Macrophages I, II, and III" with RPM, WPM, and MZM, respectively, ensuring our results link to well-characterized macrophage populations in the literature. Detailed validation of these annotations has been added in Supplementary Note S14.

Response to Question 3: Figure 4C should include a metric or value to indicate the significance level.

Thank you for pointing out that Figure 4C lacked explicit statistical metrics. We have now added a new Table 7, which reports the p-values from two-sided Wilcoxon rank-sum tests as well as two complementary effect-size measures (rank-biserial correlation r and Cliff's delta δ) for all four pairwise comparisons of CD3 expression. In the revised Figure 4C, we also annotate the comparison with significance stars corresponding to the p-value thresholds: each comparison is annotated with significance stars (* $p < 0.05$, ** $p < 0.01$, *** $p < 0.001$, **** $p < 0.0001$) (Fig. R21), supporting the comparison of clustering performance between MultiGATE and SpatialGlue.

The revised clustering results comparison between MultiGATE and SpatialGlue now reads:

"Notably, MultiGATE demonstrates a more precise clustering of T cells and B cells compared to SpatialGlue. We evaluated this by comparing the ADT expression of the CD3 protein, a canonical T cell marker, across T cell and B cell clusters identified by each method (Fig. 4C). Since CD3 is not expressed in B cells, a method that better separates these populations should yield a greater difference in CD3 expression between the two clusters. Indeed, the clustering of MultiGATE shows significantly greater separation in CD3 expression (Wilcoxon $p < 1 \times 10^{-6}$, rank-biserial $r = -0.953$, and Cliff's $\delta = 0.953$) than SpatialGlue (Wilcoxon $p < 1 \times 10^{-6}$, rank-biserial $r = -0.866$, and Cliff's $\delta = 0.866$; Supplementary Note S16). Additionally, within the T cell cluster itself, CD3 expression is significantly higher in the clustering produced by MultiGATE than SpatialGlue (Wilcoxon $p = 1.2 \times 10^{-5}$, Supplementary Note S16), suggesting more precise identification of T cells."

Figure R21: CD3 expression in different clusters.

Table 7: Comparison of CD3 expression across clusters. P-values are from two-sided Wilcoxon rank-sum tests. r is the rank-biserial correlation; δ is Cliff's delta.

Comparison	p-value	rank-biserial r	Cliff's δ
B cell cluster_ MultiGATE vs B cell cluster_ SpatialGlue	0.280711	0.034942	-0.034942
T cell cluster_ MultiGATE vs T cell cluster_ SpatialGlue	1.2×10^{-5}	-0.205942	0.205942
T cell cluster_ MultiGATE vs B cell cluster_ MultiGATE	$< 1 \times 10^{-6}$	-0.952941	0.952941
T cell cluster_ SpatialGlue vs B cell cluster_ SpatialGlue	$< 1 \times 10^{-6}$	-0.865607	0.865607

Response to Question 4: Lastly, the way of presenting Figure 4E heatmap is weak in illustrating the variation in protein intensities for each cluster.

We thank the reviewer for this insightful comment. To better illustrate the variation in protein expression across spatial clusters, we have revised the figure by replotting the heatmap using

z-score normalized protein intensities (ADT counts), as shown in Fig. R22 below. In the figure, representative proteins highly expressed in each cluster are annotated: F4-80 and CD163 for Red Pulp Macrophages (RPM); CD68 and F4-80 for White Pulp Macrophages (WPM); CD169-siglec for Marginal Zone Macrophages (MZM); CD19, IgD, and B220-CD45R for B cells; and CD3, CD4, and CD8 for T cell clusters. These annotations reveal the distinct molecular identities of each immune cell population and enhance the interpretability of spatial protein expression patterns in the mouse spleen.

Protein Z-score by Cluster

Figure R22: ADT (Antibody-Derived Tags) signatures associated with each spatial cluster, revealing distinct immune cell identities. Representative proteins highly expressed in each cluster are annotated: F4-80 and CD163 for Red Pulp Macrophages (RPM); CD68 and F4-80 for White Pulp Macrophages (WPM); CD169-siglec for Marginal Zone Macrophages (MZM); CD19, IgD, and B220-CD45R for B cells; and CD3, CD4, and CD8 for T cell clusters.

Comment 7

The biology insight from the Slide-Tags data set is not clear, and Extended Data Figure 2D is not cited.

Response:

We revised the corresponding paragraph to provide clearer biological insights into the melanoma dataset analyzed using MultiGATE and also cited Extended Data Figure 2D accordingly. The updated paragraph emphasizes the distinct cell states, immune interactions, and therapeutic implications of the two tumor clusters, while maintaining the original structure. The updated paragraph was revised as follows:

"Next, we employed MultiGATE to analyze a Slide-tags dataset [53] generated from metastatic melanoma samples, where open chromatin and RNA expression were profiled in single nuclei, along with spatial barcodes providing the spatial location of the nuclei. This dataset primarily consisted of two tumor clusters, as annotated in the original study [53] (Extended Data Fig. 2A). Consistent with these annotations, MultiGATE, along with SpatialGlue, and SeuratWNN accurately divided the tumor cells into two clusters (Extended Data Fig. 2B and 2D). Further analysis of differentially expressed genes identified by MultiGATE between tumor Cluster 1 and Cluster 2 revealed distinct expression patterns (Extended Data Fig. 2C and 2E). Cluster 1 was characterized by a mesenchymal-like state, with elevated expression and chromatin accessibility of TNC, a marker of invasion and metastasis, alongside PLCB4 and CPEB2, which support YAP signaling—a key driver of epithelial-mesenchymal transition and cellular plasticity in melanoma [54]–[58]. Cluster 1 also showed downregulation of MHC genes (CTSB, HSP90AA1, HSP90AB1, and LGMN; Extended Data Fig. 2E), suggesting immune evasion and potential resistance to immunotherapy, a trait often linked to aggressive tumor behavior [56]. In contrast, Cluster 2

displayed a melanocytic-like state with high expression of DCT and APOE, both implicated in treatment resistance via maintenance of melanocyte identity and ferroptosis evasion [59], [60]. This melanocytic state was further supported by the significant enrichment of Cluster 2 upregulated genes (in Extended Data Fig. 2E) in pigmentation (GO:0043473; Hypergeometric $P = 1.38 \times 10^{-4}$) and in MITF-regulated pathways (R-HSA-9730414; Hypergeometric $P = 6.47 \times 10^{-6}$), where MITF is a master regulator of melanocyte identity and drives melanocyte development. Additionally, Cluster 2 showed a relatively upregulation of MHC genes within the KEGG antigen processing and presentation pathway, with 7 out of 27 genes showing a fold change greater than 1.2 (Hypergeometric test, $P = 4.63 \times 10^{-6}$), indicating enhanced antigen presentation. This increased immune visibility likely promotes greater interaction with cytotoxic T cells, contributing to a more proliferative and therapy-sensitive phenotype in Cluster 2 [61], [62]. Together, these findings reveal that heterogeneity in melanoma identified by MultiGATE reflects distinct cell states with divergent therapeutic vulnerabilities."

Figure R23: Supplementary materials for Slide-tags tumor and Spots spleen.

- A** Annotated spatial distribution of Tumor 1 and Tumor 2 within the Slide-tags dataset.
- B** Spatial clustering results by MultiGATE, SpatialGlue, and Seurat WNN within the Slide-tags dataset.
- C** Expression patterns of differentially expressed genes (DEGs) between Cluster 1 and Cluster 2 as analyzed by MultiGATE.
- D** UMAP plots generated by MultiGATE, SpatialGlue, and Seurat WNN in Slide-tags dataset.
- E** Heatmap of the differentially expressed genes between Cluster 1 and Cluster 2, identified by MultiGATE clustering in the Slide-tags dataset.
- F** Overlay of histology images with spatial clustering results from MultiGATE and SpatialGlue in SPOTS dataset.

Comment 8

The color legend in Figure 3F should show the value rather than low and high.

Response:

We thank the reviewer for this helpful suggestion. In the revised Figure 3F, the color bar has been updated to display explicit numeric values (log-normalized expression). The updated panel is shown below (Fig. R24).

Figure R24: Spatial expression patterns of differentially expressed genes (DEGs) in specific brain clusters of the P22 mouse brain.

Comment 9

Color legend in Figures 3F, 4D, and 4E should claim which values are used (e.g., score, normalized, and raw data).

Response:

Thank you for pointing this out. We have updated the color legends in Figures 3F, 4D, and 4E to specify exactly which values are shown:

- **Fig. 3F:** log-normalized gene expression.
- **Fig. 4D:** centered log-ratio (CLR)–normalized protein counts.
- **Fig. 4E:** z-scores of ADT signatures across spatial clusters.

References

- [1] J. Braisted, A. Patt, C. Tindall, *et al.*, “Ramp-db 2.0: A renovated knowledgebase for deriving biological and chemical insight from metabolites, proteins, and genes,” *Bioinformatics*, vol. 39, no. 1, btac726, Nov. 2022. DOI: 10.1093/bioinformatics/btac726.
- [2] Y. Long, K. S. Ang, R. Sethi, *et al.*, “Deciphering spatial domains from spatial multi-omics with SpatialGlue,” *nature methods*, vol. 21, no. 9, pp. 1658–1667, Sep. 2024, Publisher: Nature Publishing Group TLDR: SpatialGlue, a graph neural network model with a dual-attention mechanism that deciphers spatial domains by intra-omics integration of spatial location and omics measurement followed by cross-omics integration, is introduced. DOI: 10.1038/s41592-024-02316-4.
- [3] Z.-J. Cao and G. Gao, “Multi-omics single-cell data integration and regulatory inference with graph-linked embedding,” *Nature Biotechnology*, vol. 40, no. 10, pp. 1458–1466, 2022.
- [4] Y. Hao, S. Hao, E. Andersen-Nissen, *et al.*, “Integrated analysis of multimodal single-cell data,” *Cell*, vol. 184, no. 13, pp. 3573–3587, 2021.
- [5] R. Argelaguet, D. Arnol, D. Bredikhin, *et al.*, “Mofa+: A statistical framework for comprehensive integration of multi-modal single-cell data,” *Genome biology*, vol. 21, pp. 1–17, 2020.
- [6] A. Gayoso, Z. Steier, R. Lopez, *et al.*, “Joint probabilistic modeling of single-cell multi-omic data with totalvi,” *Nature methods*, vol. 18, no. 3, pp. 272–282, 2021.
- [7] T. Ashuach, M. I. Gabitto, R. V. Koodli, G.-A. Saldi, M. I. Jordan, and N. Yosef, “Multivi: Deep generative model for the integration of multimodal data,” *Nature Methods*, vol. 20, no. 8, pp. 1222–1231, 2023.
- [8] F. Aguet *et al.*, “Genetic effects on gene expression across human tissues,” *Nature*, vol. 550, no. 7675, pp. 204–213, 2017.

- [9] T. Gao and J. Qian, “Enhancer atlas 2.0: An updated resource with enhancer annotation in 586 tissue/cell types across nine species,” *Nucleic acids research*, vol. 48, no. D1, pp. D58–D64, 2020.
- [10] F. Xie, E. J. Armand, Z. Yao, *et al.*, “Robust enhancer-gene regulation identified by single-cell transcriptomes and epigenomes,” *Cell Genomics*, vol. 3, no. 7, 2023.
- [11] R. Zheng, C. Wan, S. Mei, *et al.*, “Cistrome data browser: Expanded datasets and new tools for gene regulatory analysis,” *Nucleic acids research*, vol. 47, no. D1, pp. D729–D735, 2019.
- [12] N. Ben-Chetrit, X. Niu, A. D. Swett, *et al.*, “Integration of whole transcriptome spatial profiling with protein markers,” *Nature Biotechnology*, vol. 41, no. 6, pp. 788–793, 2023.
- [13] D. Zhang, Y. Deng, P. Kukanja, *et al.*, “Spatial epigenome–transcriptome co-profiling of mammalian tissues,” *Nature*, vol. 616, no. 7955, pp. 113–122, 2023.
- [14] E. Zhao, M. R. Stone, X. Ren, *et al.*, “Spatial transcriptomics at subspot resolution with bayesspace,” *Nature biotechnology*, vol. 39, no. 11, pp. 1375–1384, 2021.
- [15] J. Hu, X. Li, K. Coleman, *et al.*, “Spagen: Integrating gene expression, spatial location and histology to identify spatial domains and spatially variable genes by graph convolutional network,” *Nature methods*, vol. 18, no. 11, pp. 1342–1351, 2021.
- [16] K. Dong and S. Zhang, “Deciphering spatial domains from spatially resolved transcriptomics with an adaptive graph attention auto-encoder,” *Nature communications*, vol. 13, no. 1, p. 1739, 2022.
- [17] S. Heinz, C. Benner, N. Spann, *et al.*, “Simple combinations of lineage-determining transcription factors prime cis-regulatory elements required for macrophage and b cell identities,” *Molecular cell*, vol. 38, no. 4, pp. 576–589, 2010.
- [18] Z. Duren, X. Chen, J. Xin, Y. Wang, and W. H. Wong, “Time course regulatory analysis based on paired expression and chromatin accessibility data,” *Genome research*, vol. 30, no. 4, pp. 622–634, 2020.

- [19] L. Ciccone, C. Cerri, S. Nencetti, and E. Orlandini, "Carbonic anhydrase inhibitors and epilepsy: State of the art and future perspectives," *Molecules*, vol. 26, no. 21, p. 6380, 2021.
- [20] N. Lemon, E. Canepa, M. A. Ilies, and S. Fossati, "Carbonic anhydrases as potential targets against neurovascular unit dysfunction in alzheimer's disease and stroke," *Frontiers in aging neuroscience*, vol. 13, p. 772 278, 2021.
- [21] A. Esteban-Martos, A. M. Brokate-Llanos, L. M. Real, *et al.*, "A functional pipeline of genome-wide association data leads to midostaurin as a repurposed drug for alzheimer's disease," *International Journal of Molecular Sciences*, vol. 24, no. 15, p. 12 079, 2023.
- [22] M. Jorfi, A. Maaser-Hecker, and R. E. Tanzi, "The neuroimmune axis of alzheimer's disease," *Genome Medicine*, vol. 15, no. 1, p. 6, 2023.
- [23] W. Zeng, Q. Liu, Q. Yin, R. Jiang, and W. H. Wong, "Hichipdb: A comprehensive database of hichip regulatory interactions," *Nucleic Acids Research*, vol. 51, no. D1, pp. D159–D166, Oct. 2022. DOI: 10.1093/nar/gkac859.
- [24] C. Long, C. E. Grueter, K. Song, *et al.*, "Ataxia and purkinje cell degeneration in mice lacking the camta1 transcription factor," *Proceedings of the National Academy of Sciences*, vol. 111, no. 31, pp. 11 521–11 526, 2014.
- [25] S. Jiang, M. Zhang, J. Sun, and X. Yang, "Casein kinase 1 α : Biological mechanisms and theranostic potential," *Cell Communication and Signaling*, vol. 16, pp. 1–24, 2018.
- [26] T. K. Koo and M. Y. Li, "A guideline of selecting and reporting intraclass correlation coefficients for reliability research," *Journal of chiropractic medicine*, vol. 15, no. 2, pp. 155–163, 2016.
- [27] K. Coleman, A. Schroeder, M. Loth, *et al.*, "Resolving tissue complexity by multimodal spatial omics modeling with miso," *Nature methods*, pp. 1–9, 2025.
- [28] C. S. Madsen, C. P. Regan, J. E. Hungerford, S. L. White, I. Manabe, and G. K. Owens, "Smooth muscle-specific expression of the smooth muscle myosin heavy chain gene in transgenic mice requires 5-flanking and first intronic dna sequence," *Circulation research*, vol. 82, no. 8, pp. 908–917, 1998.

- [29] J. M. Miano, P. Cserjesi, K. L. Ligon, M. Periasamy, and E. N. Olson, "Smooth muscle myosin heavy chain exclusively marks the smooth muscle lineage during mouse embryogenesis.," *Circulation research*, vol. 75, no. 5, pp. 803–812, 1994.
- [30] H. Goikuria, M. d. M. Freijo, R. Vega Manrique, *et al.*, "Characterization of carotid smooth muscle cells during phenotypic transition," *Cells*, vol. 7, no. 3, p. 23, 2018.
- [31] K. Watanabe, T. Tajino, M. Sekiguchi, and T. Suzuki, "H-caldesmon as a specific marker for smooth muscle tumors: Comparison with other smooth muscle markers in bone tumors," *American journal of clinical pathology*, vol. 113, no. 5, pp. 663–668, 2000.
- [32] W.-Q. He, Y.-J. Peng, W.-C. Zhang, *et al.*, "Myosin light chain kinase is central to smooth muscle contraction and required for gastrointestinal motility in mice," *Gastroenterology*, vol. 135, no. 2, pp. 610–620, 2008.
- [33] Y. Xie, C. M. Reid, A. A. Granados, *et al.*, "Developmental origin and local signals cooperate to determine septal astrocyte identity," *bioRxiv*, 2023.
- [34] F. B. Bastian, J. Roux, A. Niknejad, *et al.*, "The bgee suite: Integrated curated expression atlas and comparative transcriptomics in animals," *Nucleic acids research*, vol. 49, no. D1, pp. D831–D847, 2021.
- [35] T. Inoue, M. Ota, M. Ogawa, K. Mikoshiba, and J. Aruga, "Zic1 and zic3 regulate medial forebrain development through expansion of neuronal progenitors," *Journal of Neuroscience*, vol. 27, no. 20, pp. 5461–5473, 2007.
- [36] A. N. Rubin, F. Alfonsi, M. P. Humphreys, C. K. Choi, S. F. Rocha, and N. Kessaris, "The germinal zones of the basal ganglia but not the septum generate gabaergic interneurons for the cortex," *Journal of Neuroscience*, vol. 30, no. 36, pp. 12 050–12 062, 2010.
- [37] M. T. García, S. K. Stegmann, T. E. Lacey, *et al.*, "Transcriptional profiling of sequentially generated septal neuron fates," *Elife*, vol. 10, e71545, 2021.
- [38] B. Wei, Z. Huang, S. He, *et al.*, "The onion skin-like organization of the septum arises from multiple embryonic origins to form multiple adult neuronal fates," *Neuroscience*, vol. 222, pp. 110–123, 2012.

- [39] Y. Ma and X. Zhou, “Spatially informed cell-type deconvolution for spatial transcriptomics,” *Nature biotechnology*, vol. 40, no. 9, pp. 1349–1359, 2022.
- [40] D. M. Cable, E. Murray, L. S. Zou, *et al.*, “Robust decomposition of cell type mixtures in spatial transcriptomics,” *Nature biotechnology*, vol. 40, no. 4, pp. 517–526, 2022.
- [41] V. Kleshchevnikov, A. Shmatko, E. Dann, *et al.*, “Cell2location maps fine-grained cell types in spatial transcriptomics,” *Nature biotechnology*, vol. 40, no. 5, pp. 661–671, 2022.
- [42] A. Andersson, J. Bergenstr hle, M. Asp, *et al.*, “Single-cell and spatial transcriptomics enables probabilistic inference of cell type topography,” *Communications biology*, vol. 3, no. 1, p. 565, 2020.
- [43] M. Elosua-Bayes, P. Nieto, E. Mereu, I. Gut, and H. Heyn, “Spotlight: Seeded nmf regression to deconvolute spatial transcriptomics spots with single-cell transcriptomes,” *Nucleic acids research*, vol. 49, no. 9, e50–e50, 2021.
- [44] Q. Song and J. Su, “Dstg: Deconvoluting spatial transcriptomics data through graph-based artificial intelligence,” *Briefings in bioinformatics*, vol. 22, no. 5, bbaa414, 2021.
- [45] R. Dong and G.-C. Yuan, “Spatialdws: Accurate deconvolution of spatial transcriptomic data,” *Genome biology*, vol. 22, no. 1, p. 145, 2021.
- [46] X. Liu, G. Tang, Y. Chen, Y. Li, H. Li, and X. Wang, “Spatialdex is a reference-free method for cell-type deconvolution of spatial transcriptomics data in solid tumors,” *Cancer research*, vol. 85, no. 1, pp. 171–182, 2025.
- [47] B. F. Miller, F. Huang, L. Atta, A. Sahoo, and J. Fan, “Reference-free cell type deconvolution of multi-cellular pixel-resolution spatially resolved transcriptomics data,” *Nature communications*, vol. 13, no. 1, p. 2339, 2022.
- [48] M. Kohyama, W. Ise, B. T. Edelson, *et al.*, “Role for spi-c in the development of red pulp macrophages and splenic iron homeostasis,” *Nature*, vol. 457, no. 7227, pp. 318–321, 2009.

- [49] B. Chen, R. Li, A. Kubota, L. Alex, and N. G. Frangogiannis, “Identification of macrophages in normal and injured mouse tissues using reporter lines and antibodies,” *Scientific reports*, vol. 12, no. 1, p. 4542, 2022.
- [50] A. Noelia, A. Castrillo, *et al.*, “Origin and specialization of splenic macrophages,” *Cellular immunology*, vol. 330, pp. 151–158, 2018.
- [51] L. C. Davies, S. J. Jenkins, J. E. Allen, and P. R. Taylor, “Tissue-resident macrophages,” *Nature immunology*, vol. 14, no. 10, pp. 986–995, 2013.
- [52] O. A. Perez, S. T. Yeung, P. Vera-Licona, *et al.*, “Cd169+ macrophages orchestrate innate immune responses by regulating bacterial localization in the spleen,” *Science immunology*, vol. 2, no. 16, eaah5520, 2017.
- [53] A. J. Russell, J. A. Weir, N. M. Nadaf, *et al.*, “Slide-tags enables single-nucleus barcoding for multimodal spatial genomics,” *Nature*, vol. 625, no. 7993, pp. 101–109, 2024.
- [54] X. Zhang, L. Yang, P. Szeto, *et al.*, “The hippo pathway oncoprotein yap promotes melanoma cell invasion and spontaneous metastasis,” *Oncogene*, vol. 39, no. 30, pp. 5267–5281, 2020.
- [55] C. Savorani, M. Malinverno, R. Seccia, *et al.*, “A dual role of yap in driving $\text{tgf}\beta$ -mediated endothelial-to-mesenchymal transition,” *Journal of Cell Science*, vol. 134, no. 15, jcs251371, 2021.
- [56] M. Fukunaga-Kalabis, G. Martinez, T. Nguyen, *et al.*, “Tenascin-c promotes melanoma progression by maintaining the abcb5-positive side population,” *Oncogene*, vol. 29, no. 46, pp. 6115–6124, 2010.
- [57] H. T. Phan, N. H. Kim, W. Wei, G. G. Tall, and A. V. Smrcka, “Uveal melanoma-associated mutations in $\text{plc}\beta 4$ are constitutively activating and promote melanocyte proliferation and tumorigenesis,” *Science signaling*, vol. 14, no. 713, eabj4243, 2021.
- [58] P. Zeng, F. Wang, X. Long, *et al.*, “Cpeb2 enhances cell growth and angiogenesis by upregulating arpc5 mRNA stability in multiple myeloma,” *Journal of Orthopaedic Surgery and Research*, vol. 18, no. 1, p. 384, 2023.

- [59] B. J. Pak, J. Lee, B. L. Thai, *et al.*, “Radiation resistance of human melanoma analysed by retroviral insertional mutagenesis reveals a possible role for dopachrome tautomerase,” *Oncogene*, vol. 23, no. 1, pp. 30–38, 2004.
- [60] S. More, J. Bonnereau, D. Wouters, *et al.*, “Secreted apoe rewires melanoma cell state vulnerability to ferroptosis,” *Science Advances*, vol. 10, no. 42, eadp6164, 2024.
- [61] S. J. Rodig, D. Gusenleitner, D. G. Jackson, *et al.*, “Mhc proteins confer differential sensitivity to ctla-4 and pd-1 blockade in untreated metastatic melanoma,” *Science translational medicine*, vol. 10, no. 450, eaar3342, 2018.
- [62] D. B. Johnson, M. V. Estrada, R. Salgado, *et al.*, “Melanoma-specific mhc-ii expression represents a tumour-autonomous phenotype and predicts response to anti-pd-1/pd-l1 therapy,” *Nature communications*, vol. 7, no. 1, p. 10 582, 2016.

Authors' Response to Reviewer 3

General Comments. The authors present MultiGATE, a novel method for integrating spatial multi-omics data that utilizes a two-level graph attention autoencoder. The key innovation is that it combines CLIP loss and two-level GATE to model the cross-modality regulatory relationship for regulatory inference, and to do spatial embedding for clustering/spatial domain identification. The method seems reasonable for this task and is well-described. The authors benchmarked against several popular methods on 3 datasets, showing MultiGATE's better performance.

Response:

Thank you for your insightful feedback and constructive suggestions. We have carefully addressed each of your comments in the revised manuscript as detailed below.

Remarks on code availability. The authors provide an easily installable package, accompanied by comprehensive documentation. This includes step-by-step instructions for installing the MultiGATE package, along with tutorials on how to analyze various spatial multi-omics datasets using the tool. These resources enable users to replicate the provided results and examples with ease.

Response:

We thank the reviewer for the positive feedback on our code organization and tutorial website.

Comment 1

All the benchmarking datasets are from brain or spleen, both of which exhibit a stereotypical structure. Can it be applied to tissues without such stereotypical structure, such as tumors, skeletal muscle, colon, liver, lung, and adipose tissue? Benchmarking on a dataset from a tissue without a stereotypical structure would make a stronger case, to be more representative of the full breadth of spatial multi-omics datasets

Response:

We thank the reviewer for raising this important point. To demonstrate that MultiGATE generalizes beyond tissues with stereotypical laminar organization, we have now benchmarked MultiGATE on both a human breast cancer dataset [1] profiled with the 10× Visium CytAssist protocol (spatial RNA + protein) and a simulated breast cancer–patterned spatial ATAC + RNA dataset.

1. Human breast cancer (spatial RNA + protein).

Breast cancer tissue does not exhibit a stereotypical structure (Fig. R25A). To quantitatively evaluate the clustering results of MultiGATE and other comparison methods, we calculated the Intraclass Correlation Coefficient (ICC) [2] to evaluate the clustering performance. The ICC measures the consistency of observations within clusters. A higher ICC value indicates increased homogeneity within clusters for a given modality, suggesting an improvement in clustering quality [3]. MultiGATE achieved the highest combined Intraclass Correlation Coefficient (ICC) for RNA and protein, outperforming SpatialGlue, Seurat WNN, MOFA+, and totalVI (MultiGATE: 0.568; SpatialGlue: 0.462; Seurat WNN: 0.539; MOFA+: 0.509; totalVI: 0.421) (Fig. R25B).

Figure R25: Benchmarking MultiGATE and alternative methods on an FFPE human breast cancer sample profiled with 10x Visium CytAssist (spatial RNA and protein).

A Hematoxylin and eosin (H&E) stained section of the breast cancer tissue.

B Bar plot of the combined ICC (RNA+Protein) values across methods, with MultiGATE achieving the highest score.

C Spatial cluster assignments from MultiGATE, SpatialGlue, Seurat WNN, totalVI, and MOFA+.

2. Simulation dataset of breast cancer-patterned spatial ATAC + RNA.

We could not find spatial ATAC + RNA datasets available for tumor or other non-stereotypically structured tissues (e.g., skeletal muscle, colon, liver, lung or adipose), so we generated a simulation dataset by projecting paired hippocampal spatial ATAC+RNA profiles [4] onto the Visium coordinates of a breast cancer section [5] (Simulation Algorithms 1; Fig. R26A), this simulated dataset preserves the spatial heterogeneity of the breast cancer section by assigning genuine hippocampal ATAC and RNA profiles to each breast cancer Visium spot.

When evaluated on this simulated dataset, MultiGATE outperformed four integration methods—SpatialGlue, Seurat WNN, MultiVI and MOFA+—across eight clustering metrics (Fig. R26B). In particular, MultiGATE achieved an adjusted Rand index of 0.46, compared with 0.39 for SpatialGlue, 0.23 for Seurat WNN, 0.26 for MultiVI and 0.06 for MOFA+. Notably, while the ground-truth annotation comprised 5 clusters, MOFA+ was unable to get 5 clusters. Even at a very low Louvain resolution of 0.01, MOFA+ produced 6 clusters, with the vast majority of pixels collapsing into a single cluster (Fig. R26D). We additionally evaluated the MOFA+ at a higher resolution of 0.1 (yielding eight clusters) and used these 8-cluster results for quantitative comparisons (Fig. R26D). These results underscore the robustness of MultiGATE in integrating spatial ATAC + RNA data from tissues that lack a stereotypical structure.

Together, these analyses confirm that MultiGATE can be applied to analyze spatial multi-omics data from tissues without stereotypical structure, such as breast cancer, and can handle both spatial RNA+protein and spatial ATAC+RNA modalities.

Manuscript Revisions

1. Added benchmarking of MultiGATE on tissues without stereotypical structure (human breast cancer spatial RNA + protein data and simulated breast cancer-patterned spatial ATAC + RNA data) to the Results section (lines 312–315).
2. Described the simulation algorithm for generating the breast cancer-patterned spatial ATAC + RNA dataset in Supplementary Note S16.
3. Added MultiGATE's spatial clustering results in Extended Data Figure 4 (breast cancer RNA + protein dataset) and Extended Data Figure 5 (simulated breast cancer-patterned spatial ATAC + RNA dataset).

Simulated Dataset (Spatial ATAC+RNA)

Figure R26: Benchmarking MultiGATE and alternative methods on a simulation dataset (spatial ATAC and RNA).

- A** Simulated dataset (spatial ATAC+RNA) by projecting spatial ATAC + RNA profiles from the human hippocampus atlas onto the breast cancer section. Each dot represents one Visium spot, colored by its original breast cancer cell type in the Visium section (Cell type; left legend) and annotated with the corresponding hippocampal cluster (Clusters 1–5; right legend).
- B** Quantitative comparison of clustering performance across methods. Bars show eight metrics—Adjusted Rand Index (ARI), Rand Index, Adjusted Mutual Information (AMI), Normalized Mutual Information (NMI), Homogeneity, Completeness, V-measure, and Fowlkes–Mallows score—for MultiGATE (red), SpatialGlue (blue), Seurat WNN (teal), MultiVI (orange), and MOFA+ (purple).
- C** Spatial cluster assignments for each method. Insets report the ARI achieved by MultiGATE (0.46), SpatialGlue (0.39), Seurat WNN (0.23), MultiVI (0.26), and MOFA+ (0.06).

Simulation algorithm of breast cancer-patterned spatial ATAC+RNA data.

We simulated a spatial ATAC + RNA data using molecular data (ATAC + RNA) from human hippocampal spatial ATAC-RNA-seq dataset [4] and spatial coordinates from breast cancer Visium dataset [5] through the following process:

Data inputs

- **Human hippocampus multi-omic atlas.** Paired spatial ATAC + RNA profiles with expert-annotated cell-type labels [4].
- **Breast cancer Visium data.** Spatial coordinates and 5 cell-type groups were obtained from a spatially resolved atlas of human breast cancers [5] (10X visium technology).

Spatial assignment pipeline

- (1) **Cell-type mapping.** Establish a one-to-one correspondence between each of the five cell-type categories in the breast cancer Visium data and the annotated clusters in the hippocampus ATAC+RNA data.
- (2) **Spatial coordinates assignment.** For each mapped cell type, let n_{visium} be the number of spatial spots in Visium breast cancer data and $n_{\text{spatialmulti}}$ the number of spatial pixels in human hippocampus spatial ATAC+RNA data. We then assigned RNA+ATAC profiles to Visium spots' spatial coordinates according to:

$n_{\text{spatialmulti}} = n_{\text{visium}}$: Establish a one-to-one mapping that preserves local neighborhood structure, so that nearby pixels in the hippocampus data are assigned to adjacent Visium spots.

$n_{\text{spatialmulti}} < n_{\text{visium}}$: Sample with replacement from the set of pixels until each spatial spot is assigned one ATAC+RNA profile.

$n_{\text{spatialmulti}} > n_{\text{visium}}$: Randomly downsample the set of pixels so that each pixel is assigned a spatial coordinate.

Comment 2

The authors include spatialGlue and Seurat WNN in their benchmark. However, several other popular methods have been proposed to integrate multi-omics datasets. I would like to suggest the authors introduce more state-of-the-art methods in the introduction such as totalVI, MultiVI, MOFA etc. Also, more methods should be included in the benchmarking analyses.

Response:

We thank the reviewer for this suggestion. In the revised manuscript, we have now:

1. **Introduced totalVI, MultiVI and MOFA+ in the Introduction.** We briefly describe:

- *totalVI* [6], a variational-inference framework for analyzing joint RNA+protein (CITE-seq) data, modeling transcriptome counts with negative-binomial (NB) distribution and the protein counts as NB mixture of foreground and background signal;
- *MultiVI* [7], a conditional variational autoencoder extending scVI to ATAC+RNA single-cell multi-omics, modeling raw gene expression counts with a negative-binomial distribution and binarized chromatin-accessibility observations with a Bernoulli distribution;
- *MOFA+* [8], a Bayesian factor-analysis model that decomposes multiple omics input matrices into shared and view-specific low-rank factors.

2. **Expanded our benchmarking** to include these three additional methods:

- *totalVI* on the SPOTS (Spatial RNA+protein) mouse spleen dataset;
- *MultiVI* on the human hippocampus (Spatial ATAC+RNA) and mouse P22 (Spatial ATAC+RNA) datasets;
- *MOFA+* on all three datasets.

We compared the spatial clustering results of MultiGATE and the three methods and evaluated the performance using quantitative integration metric (adjusted random index(ARI)/combined ICC). In the human hippocampus dataset (Fig. R27A), MultiGATE achieves the highest Adjusted Rand

Index (ARI = 0.58), surpassing the other methods, with MOFA+ and MultiVI obtaining ARIs of 0.10 and 0.14, respectively (Fig. R27A). For the mouse P22 brain (spatial ATAC+RNA) and SPOTS spleen (spatial RNA+protein) datasets (Figs. R27 B, C), where ground-truth labels are unavailable, we evaluate the clustering results via the combined ICC scores across modalities; in both cases, MultiGATE attains the highest combined ICC (Fig. R28). Together, these results demonstrate that MultiGATE consistently outperforms alternative methods (including SpatialGlue, Seurat WNN, totalVI, MultiVI, and MOFA+) in clustering accuracy.

Manuscript Revisions

We have updated the manuscript as follows:

1. Added totalVI, MultiVI, and MOFA+ to the Introduction (lines 50–61).
2. Incorporated results for totalVI, MultiVI, and MOFA+ into the Results section.
3. Added totalVI, MultiVI, and MOFA+ results to Extended Data Figures 7 and 8.

Figure R27: Clustering Results of MultiGATE,totalVI,MultiVI and MOFA+.

A Human hippocampus dataset spatial clustering results of MultiGATE, MOFA+, and MultiVI.

B Mouse P22 dataset spatial clustering results of MultiGATE, MOFA+, and MultiVI.

C Mouse Spleen dataset spatial clustering results of MultiGATE, MOFA+, and totalVI.

Figure R28: Bar plot of the combined ICC values across methods.

- A** Bar plot of the combined ICC (ATAC+RNA) values across methods in Mouse P22 dataset.
- B** Bar plot of the combined ICC (RNA+protein) values across methods in Mouse Spleen dataset.

Comment 3

It's interesting that attention mechanism in MultiGATE can capture the cis-regulatory interactions and identify the peak-pair regulation. Is it possible to model trans-regulation, and if so, why or why not? Additionally, could transcriptional regulation be modeled by incorporating gene-protein associations (i.e., how does gene expression affect protein level)?

Response:

Response to Question 1: Is it possible to model trans-regulation, and if so, why or why not?

We thank the reviewer for highlighting the interpretability of the attention mechanism in modeling cis-regulation and for asking about its extension to trans-regulation. Indeed, one

of the strengths of MultiGATE is its ability to represent regulatory relationships as weighted connections in the attention layers, which naturally generalizes to trans-regulation when additional context is provided.

To enable this, we revised the model to explicitly incorporate TF binding information, thereby modeling trans-regulation mediated by TFs. This is accomplished by introducing a TF binding potential (TFBP) as an additional prior in the cross-modality autoencoder. Formally, for a given TF k and peak i , the prior is defined as:

$$\text{TFBP}_{k,i} = M_{k,i} \times X_i \quad (7)$$

where $M_{k,i} \in \{0, 1\}$ is the binarized motif binding score for TF k on peak i . The motifs were scanned using HOMER [9], with motif-to-TF mapping curated from [10]. It is set to 1 if at least one binding site for TF k is found in peak i , and 0 otherwise. X_i denotes the average chromatin accessibility of peak i across all spots. This score integrates both motif presence and chromatin openness, serving as a biologically informed prior for TF binding potential. The original peak–gene (excluding the TFs) attention mechanism remains unchanged, allowing MultiGATE to simultaneously learn both cis and trans regulations within a unified attention framework. To quantify predicted TF binding to peaks, we define the TF–peak integrated attention score as:

$$S_{k,i} = \alpha_{k,i} \times \sum_{j \in \mathcal{G}(i)} \text{Att}_{i,j} \quad (8)$$

where $\alpha_{k,i}$ is the learned attention score from TF k to peak i , $\text{Att}_{i,j}$ is the attention score from peak i to gene j , and $\mathcal{G}(i)$ is the set of genes linked to peak i . This metric prioritizes TF–peak pairs that are both accessible and functionally linked to gene expression.

We tested this framework using the spatial ATAC–RNA–seq dataset from the adult human hippocampus and evaluated the predictions for SOX2, a well-studied brain-specific TF. The ChIP-seq data for SOX2 in brain tissue were downloaded from the Cistrome database [11] and served as the ground truth for evaluating predicted TF binding peaks. We compared our model against several baseline methods: The motif-only baseline was constructed using the binary motif TF binding score (defined as M in Eq. (7)) identified via HOMER [9], with motif-to-TF mapping curated from [10]; The TF binding potential baseline was defined as the product

of binary motif binding and chromatin openness (defined as TFBP in Eq. (7)); The cosine similarity baseline was computed as the cosine similarity between the expression profile of the TF (from spatial RNA data) and the accessibility profile of each peak (from spatial ATAC data) across all spots.

Figure R29: Comparison of methods for predicting SOX2 binding in the adult human hippocampus. Motif_Binding denotes the binary motif TF binding score. TFBinding_Potential is computed as the product of binary motif binding score and chromatin openness (defined as TFBP in Eq. (7)). The Cosine_TF_Peak is computed as the cosine similarity between the expression profile of the TF and the accessibility profile of each peak across all the cells. Attention_Score demonstrates the TF–peak integrated attention score defined as $S_{k,i}$ in Eq. (8).

Our method’s learned attention scores (defined as $S_{k,i}$ in Eq. (8)) were highly predictive of TF binding events, with an AUC of 0.8669 and AUPR of 0.4906, substantially outperforming motif-only (AUC = 0.4906, AUPR = 0.1934), TF binding potential baseline (AUC = 0.6326,

AUPR = 0.2845), and cosine similarity between TFs and peaks (AUC = 0.6280, AUPR = 0.2752), as shown in Fig. R29.

These findings show that the attention mechanism, when extended with TF inputs, is capable of modeling trans-regulatory interactions effectively. However, we acknowledge that the limited availability of high-quality ChIP-seq datasets for many TFs in brain regions like the hippocampus makes it difficult to validate the generalizability of our method across TFs. We consider this an important direction for future work and plan to incorporate broader validation using emerging tissue-specific datasets. These capabilities and limitations are discussed in the revised manuscript (lines 333-337).

Response to Question 2: could transcriptional regulation be modeled by incorporating gene-protein associations (i.e., how does gene expression affect protein level)?

MultiGATE can model transcriptional regulation by incorporating gene-protein associations into the cross-modality attention mechanism. While the original framework only links each protein feature to the genes encoding the protein, we have generalized this mechanism so that any protein feature can be linked to a curated set of related genes—and vice versa—thereby capturing broader regulatory relationships. To illustrate this, we applied MultiGATE to the SPOTS dataset, which jointly profiles spatial RNA and protein. Besides the original links between proteins and the genes encoding the protein in the cross-modality feature connectivity graph, we curated a *CD3-related* gene set and added an edge between the CD3 protein and each gene in the *CD3-related* gene set. To test whether MultiGATE can accurately recover known associations, we also defined three negative-control gene sets (B cell-related, macrophage-related, and randomly selected). We added identical edges between CD3 and the genes in these sets into the cross-modality feature connectivity graph. We then compared the learned attention scores between CD3 and each gene across the four sets.

1. ***CD3-related* gene set:** CD3 is a T cell marker, We queried RAMP-DB [12] for all pathways containing any of the four CD3 chains (CD3 zeta (ζ), delta (δ), epsilon (ϵ), and gamma (γ)). From those pathways (TCR signaling; Modulators of TCR signaling and T cell activation; T-cell activation SARS-CoV-2; T-cell antigen receptor (TCR) pathway

during *Staphylococcus aureus* infection; T-cell receptor signaling pathway and other pathways), we randomly extracted 48 member genes to form our *CD3-related* set.

2. **Negative control gene sets:** We queried RAMP-DB [12] for two immune-related gene sets and one randomly selected gene set from all human protein-coding genes. Each set contains 16 genes (so that all 3 sets total 48 genes):

- **B cell-related gene set:** From four B cell–related pathways—B cell receptor signaling pathway; Extrafollicular and follicular B cell activation by SARS-CoV-2; FBXL10 enhancement of MAP/ERK signaling in diffuse large B-cell lymphoma; and Antigen activates B Cell Receptor (BCR) leading to generation of second messengers, we randomly selected 16 genes.
- **Macrophage-related gene set:** From six macrophage–related pathways—ER-Phagosome pathway; Induction of autophagy and toll-like receptor signaling pathways by graphene oxide; Toll-like receptor signaling pathway; Toll-like receptor signaling related to MyD88; Cytokine–cytokine receptor interaction; and NOD-like receptor signaling pathway, we randomly selected 16 genes.
- **Randomly selected gene set:** We randomly drew 16 genes from human protein-coding genes.

Each control set contains 16 genes, matching the total of 48 genes in the *CD3-related* set.

In our prior cross-modality feature-connectivity graph, besides the original connections between proteins and their encoding genes, we connected each gene in all four sets to the CD3 protein, and recorded the learned attention scores between these genes and the CD3 protein.

Figure R30: Distribution of attention scores between the CD3 protein and four groups of genes: (1) genes from the CD3-Related pathway, (2) genes from the B cell pathway, (3) genes from the macrophage pathway, and (4) randomly selected genes. The central box in each group represents the interquartile range (IQR), with the median indicated by the horizontal line. Whiskers extend to $1.5 \times$ IQR, and outliers are shown as individual points, and the yellow marker indicates the mean value.

To evaluate that MultiGATE can distinguish the genes related to the CD3 protein, we compare the attention-score distributions for four gene sets: the *CD3-related* genes show the highest median and mean attention scores, followed by B cell-related genes, macrophage-related genes, and random-selected genes (Fig. R30). This suggests that MultiGATE can learn CD3-related genes.

To further validate that high attention score reflects functional proximity in *CD3-related* gene set, we mapped each gene's position onto the KEGG T cell receptor signaling pathway

(hsa04660). We color-coded the ten genes with the highest attention score (red) and the ten with the lowest attention score (blue), revealing that genes with high attention scores are clustered near CD3 nodes, whereas genes with low attention scores tend to lie further away (Fig. R31). Quantitatively, the top ten have an average shortest-path distance of 2.1 to any CD3 chain, versus 3.5 for the bottom ten, suggesting that our attention scores reflect functional proximity in the biological pathway.

The above examples demonstrate that MultiGATE can learn and prioritize gene–protein associations, accurately distinguishing true CD3-related genes from negative controls.

Manuscript revisions

We have updated the manuscript as follows:

1. We have included trans-regulatory modeling and results to the Results section (lines 149–154) and Supplementary Note S2. We also discussed the capabilities and limitations of modeling trans-regulation in the Discussion (lines 333–337).
2. We have added a detailed description of how gene–protein associations are modeled via cross-modality attention, including the curation of CD3-related and negative-control gene sets, their incorporation into the cross-modality feature connectivity graph, and the evaluation of learned attention scores, to Supplementary Note S3 (see also main text, lines 278–282).

Figure R31: Visualization of gene proximity in the T cell receptor signaling pathway. Red indicates the top ten genes with the highest attention scores with CD3 in the T cell receptor signaling pathway. The bottom ten genes, with the lowest attention scores relative to the CD3 family, are shown in blue.

Comment 4

In Figure 2A, the authors calculated ARI for each method and compared their clustering accuracy. What parameters are used for clustering (i.e. resolution of Louvain clustering)? Is the ARI presented the highest value each method could achieve? Also, the number of spots appears to be inconsistent across the different methods. Could the authors also clarify the clustering parameters used for the resulting clusters in Figure 3B?

Response:

Response to Question 1: In Figure 2A, the authors calculated ARI for each method and compared their clustering accuracy. What parameters are used for clustering (i.e. resolution of Louvain clustering)? Is the ARI presented the highest value each method could achieve?

We thank the reviewer for this question. Below we clarify how the clustering parameters were chosen.

1. Louvain clustering for MultiGATE and Seurat WNN

Both MultiGATE and Seurat's WNN pipeline employ the Louvain community-detection algorithm. We performed a fine-grained grid search over the resolution parameter, retaining only those settings that yield exactly seven clusters (to match the expert-annotated number). Specifically:

- *MultiGATE*: Seven clusters arise for resolution values in the interval [0.515, 0.615] (Fig. R33). Across this window, the ARI varies by less than 0.01.
- *Seurat WNN*: Seven clusters occur for resolutions in [0.25, 0.50] (Fig. R34). Within this range, the ARI likewise fluctuates by under 0.01.

Thus, the ARI curves in Fig. R32 report the possible ARI for each method under the constraint of seven clusters; The ARI changes a little (<0.01) across different resolutions. For both methods based on Louvain clustering, the value in Figure 2A represents the best achievable ARI under the constraint of seven clusters (any other resolution giving seven clusters produces virtually the same ARI, with $\Delta < 0.01$).

2. Gaussian mixture clustering for SpatialGlue

SpatialGlue does not expose a Louvain resolution parameter. Instead, it fits a Gaussian mixture model (via the `mclust` [13] package) on the combined embeddings, with the only user-specified input being the number of mixture components. We fixed $K = 7$ to align with the expert annotation.

Manuscript revisions

We have added a sensitivity analysis of the Louvain clustering resolution in Supplementary Note S8 (see also lines 550-551 in the main text).

Figure R32: Clustering accuracy (Adjusted Rand Index, ARI) vs the Louvain resolution parameter on the human hippocampus spatial multi-omics data. Shaded bands mark the resolution ranges yielding 6 (green), 7 (orange) and 8 (blue) clusters. **A** ARI versus resolution for the MultiGATE method. **B** ARI versus resolution for the Seurat WNN method.

Figure R33: Spatial clustering results of MultiGATE across different Louvain resolution parameters on the human hippocampus spatial multi-omics data.

Figure R34: Spatial clustering results of Seurat WNN across different Louvain resolution parameters on the human hippocampus spatial multi-omics data.

Response to Question 2: In Figure 2A, Also, the number of spots appears to be inconsistent across the different methods.

We thank the reviewer for noting the discrepancy in the number of spots shown in Figure 2A. In our analysis of the human hippocampus dataset, both MultiGATE and Seurat WNN were applied to the same set of 2,500 spatial pixels. By contrast, SpatialGlue has a filtering step for the pixels as described in the official SpatialGlue tutorial for spatial epigenome–transcriptome data, which reduces the number of retained spots to 2,211. In particular, we applied the following preprocessing steps in Scanpy following the official tutorial in SpatialGlue website https://spatialglue-tutorials.readthedocs.io/en/latest/Tutorial%204_data%20integration%20for%20mouse%20brain%20Spatial-epigenome-transcriptome.html:

```
# RNA modality filtering
sc.pp.filter_genes(adata_omics1, min_cells=10)
sc.pp.filter_cells(adata_omics1, min_genes=200)
# Match ATAC modality to filtered RNA cells
adata_omics2 = adata_omics2[adata_omics1.obs_names].copy()
```

These commands remove low-coverage genes and cells (containing at least 200 genes) from the RNA data, then restrict the ATAC dataset to the same cell barcodes, yielding 2,211 pixels. To ensure a fair comparison, the Adjusted Rand Index (ARI) reported in Figure 2A was computed only on these 2,211 spatial locations for all methods.

Manuscript revisions

We have clarified the number of pixels in the Results section (lines 111–113).

Response to Question 3: Could the authors also clarify the clustering parameters used for the resulting clusters in Figure 3B?

In Figure 3B, we compare spatial clustering results on the P22 mouse brain dataset across MultiGATE, SpatialGlue and Seurat’s WNN, each constrained to recover eighteen clusters (to match the SpatialGlue cluster numbers reported in this dataset for a fair comparison). The clustering parameters (resolution in the Louvain algorithm) and spatial clustering results for each method are as follows:

- **MultiGATE & Seurat WNN (Louvain clustering):** We performed a grid search over the Louvain resolution parameter and retained only those values yielding 18 clusters.
 - *MultiGATE* produces eighteen clusters for resolutions in the range [1.95, 2.10], with the resulting spatial clustering results remaining highly consistent throughout this window (Fig. R35A).
 - *Seurat WNN* achieves eighteen clusters for a resolution interval of [0.45, 0.48], again yielding near-identical spatial clustering results across this band (Fig. R35B).
- *SpatialGlue*: SpatialGlue does not use Louvain clustering but fits a Gaussian mixture model via the `mclust` [13] package. We fixed the number of mixture components to $K = 18$ to match the results in the SpatialGlue paper results, following the hyperparameter recommendations on the SpatialGlue official tutorial website https://spatialglue-tutorials.readthedocs.io/en/latest/Tutorial%204_data%20integration%20for%20mouse%20brain%20Spatial-epigenome-transcriptome.html.

Manuscript revisions

We have added a sensitivity analysis of the Louvain clustering resolution in Supplementary Note S8 (see also lines 550-551 in the main text).

A**MultiGATE spatial clustering results vs resolution****B****Seurat WNN spatial clustering results vs resolution**
Figure R35: Spatial clustering results of MultiGATE and Seurat WNN across different Louvain resolution parameters on the Mouse P22 ¹⁰⁶ multi-omics data.

(A: MultiGATE, B: Seurat WNN)

Comment 5

Figure 3F illustrates several marker genes that are highly spatially enriched in certain brain regions, as reported in previous studies (line 183-196 in the manuscript). However, the identified clusters do not perfectly align with these brain regions (Figure 3A, B). How, then, is the marker gene Pde10a (Figure 3F), which is highly expressed in the CP region, identified through DEG analysis, given that the CP region appears to be separated into several distinct clusters?

Response:

We thank the reviewer for this insightful question. MultiGATE's joint modeling of gene expression and chromatin accessibility reveals heterogeneity within the caudoputamen (CP), splitting it into three subclusters (Clusters 1, 4, and 14; Fig. R36A, B). We performed three differential expression analyses. For each analysis, we compared one cluster (Clusters 1, 4, or 14) against all other clusters using the Wilcoxon rank-sum test with Benjamini–Hochberg correction. In every case, Pde10a ranked among the very top up-regulated genes (see volcano plots in Fig. R37):

- **Cluster 1 vs. others:** \log_2 fold-change = 1.62, adjusted $p < 7.96 \times 10^{-150}$.
- **Cluster 4 vs. others:** \log_2 fold-change = 1.83, adjusted $p < 8.76 \times 10^{-153}$.
- **Cluster 14 vs. others:** \log_2 fold-change = 1.63, adjusted $p < 1.17 \times 10^{-73}$.

This consistent enrichment explains why Pde10a emerges in our marker-gene analysis of the CP region (Fig. 3F), even though CP is subdivided in the clustering.

Manuscript revisions

We have added details of the differential expression analyses to Supplementary Note S12 (see also main text, lines 215–216).

Figure R36: Spatial clustering results in P22 mouse brain dataset. **A** Annotated coronal section of a P56 mouse brain from the Allen Mouse Brain Atlas. **B** Spatial clustering of brain regions in a P22 mouse brain using MultiGATE.

Figure R37: Volcano plots for DEGs of CP subclusters (Cluster 1, Cluster 4, and Cluster 14) versus all other clusters.

Comment 6

The Methods section states that in the cross-modality autoencoder, the dimensions are set to match the number of spots. What is the rationale behind this choice? As is known, high-resolution spatial multi-omics techniques can generate tens of thousands of spots in a single slice. How can the autoencoder dimensions be set to match such a large number?

Response:

Thank you for pointing out this question. We apologize for the confusion due to the description "the dimensions are set to match the number of spots" in the original version of the manuscript. To clarify, we use the spatial ATAC+RNA data as an example. In the cross-modality attention mechanism, we integrate the information of each gene with its neighboring peaks according to the cross-modality feature connectivity graph, and each peak with its neighboring genes. As a result, the cross-modality encoder in MultiGATE produces two enhanced data matrices: one of size (Genes \times Spots) and one of size (Peaks \times Spots). In the within-modality graph attention autoencoder, the model performs spatial smoothing over neighboring spots (Fig. R39).

As for the memory usage, MultiGATE does not store intermediate variables such as the enhanced data matrices (output of the cross-modality attention encoder). Since the spatial multi-omics data itself is the same for all the computing methods and MultiGATE does not store intermediate variables, the primary difference in memory usage comes from the number of parameters. We therefore focus on the parameters we used in the cross-modality autoencoder.

In the cross-modality encoder of MultiGATE, the only learnable parameters are two vectors of length N (the number of spatial spots), for a total of $2N$ parameters. Thus, even when N reaches tens of thousands, the parameter memory remains very modest and scales linearly with N (Fig. R38).

Figure R38: GPU memory consumed by the cross-modality module as a function of the number of spatial spots N . Only the learnable parameters are profiled; the input matrices are identical across competing methods.

The following explains the number of parameters used in MultiGATE's cross-modality attention autoencoder.

Parameterisation of the cross-modality encoder.

For every genomic feature f , the cross-modality encoder computes its representation by

$$\bar{\mathbf{X}}_{(f)}^\top = \sigma\left(\sum_{g \in \mathcal{A}_f} \alpha_{fg} \mathbf{X}_{(g)}^\top + \alpha_{ff} \mathbf{X}_{(f)}^\top\right), \quad (9)$$

where $\sigma(\cdot)$ is the ReLU activation function, \mathcal{A}_f is the neighbourhood of feature f , and the attention scores are

$$e_{fg} = \text{Sigmoid}(\mathbf{v}_1^\top \mathbf{X}_{(f)}^\top + \mathbf{v}_2^\top \mathbf{X}_{(g)}^\top), \quad (10)$$

$$\alpha_{fg} = \frac{A_{fg} \exp(e_{fg})}{\sum_{h \in \mathcal{A}_f} A_{fh} \exp(e_{fh})}, \quad (11)$$

with trainable vectors $\mathbf{v}_1, \mathbf{v}_2 \in \mathbb{R}^N$. Here A_{fg} represents the prior knowledge that quantifies the connection between features f and g based on their genomic distance, and its value decays when the genomic distance increases.

MultiGATE *does not* introduce a dense $N \times N$ weight matrix to project the input matrix; it uses only two trainable vectors, $\mathbf{v}_1, \mathbf{v}_2 \in \mathbb{R}^N$, which are shared by *all* features and reused by the decoder through parameter sharing. Consequently, the cross-modality module adds just $2N$ parameters in total. As Fig. R38 shows, GPU memory grows linearly with N and remains a small parameter memory usage even when $N \approx 20,000$.

To make this more explicit, we have removed the sentence "the dimensions are set to match the number of spots" and added the following paragraph in the revised manuscript:

"In all datasets, the cross-modality mechanism consists of one GAT (Graph Attention Network) layer with two learnable vectors to compute attention scores between cross-modality features. In the cross-modality attention mechanism, we integrate the information of each gene with its neighboring peaks (or proteins) according to attention scores, and each peak (or protein) with its neighboring genes. The cross-modality encoder then outputs two enhanced data matrices—one of size (Peaks \times Spots) and one of size (Genes \times Spots)—which integrate information from the other modality and are passed on to the within-modality attention step.

For the within-modality attention autoencoders, each encoder has two hidden layers (dimensions 512 and 30), and the decoder is symmetric to the encoder."

Figure R39: Overview of MultiGATE.

MultiGATE is a two-level graph-attention auto-encoder designed for spatial multi-omics analysis. It integrates chromatin accessibility or protein information with gene expression data using an attention mechanism. Cross-modality attention (level 1) captures relationships between modalities, while within-modality attention (level 2) incorporates spatial proximity. In addition to reconstruction loss, a CLIP contrastive loss aligns embeddings across modalities. MultiGATE yields (i) latent representations of pixels for joint clustering and visualization and (ii) cross-modality attention scores for regulatory inference.

Comment 7

Line 399 says the dimension of the hidden layers of encoders are 512, 30 respectively. Does this mean the latent dimension is set to be 30? Why is this number chosen? How does this affect downstream results?

Response:

Thank you for pointing this out. Indeed, the latent dimensionality of MultiGATE is set to 30 for all datasets. We chose 30 for the following reasons:

1. **Empirical precedent.** A latent space of roughly 30 dimensions is widely adopted in recent graph-based embedding methods for spatial transcriptomics. For example, STAGATE [14] uses a 30-dimensional embedding by default [14], STMGCN [15] employs 32 dimensions, and DeepST [16] uses 28 dimensions. SEDR [17] uses 32 latent dimensions. Open-ST [18] uses 30 latent dimensions.
2. **Robustness to hyperparameter choice.** We performed a sensitivity analysis on the human hippocampus dataset (spatial ATAC + RNA) and mouse brain dataset (spatial transcriptomics + metabolomics), varying the latent dimension from 20 to 50 in increments of 5. As shown in Fig. R40 and Fig. R41, the clustering results are stable across this range.

Manuscript revisions

We have incorporated a latent-dimension sensitivity analysis (20–50) into Supplementary Note S20 (see also lines 462-463 in the main text).

Human hippocampus dataset (Spatial ATAC-RNA-seq)

Figure R40: Impact of latent dimensionality on spatial clustering performance in the human hippocampus dataset.

Mouse Brain dataset (Spatial transcriptomics + metabolomics)

Figure R41: Impact of latent dimensionality on spatial clustering performance in the mouse brain dataset.

Comment 8

For peak-gene connection, why is the base pair distance restricted within 150 k? Line 311 says “ $A_{i,j} = 1$ if and only if the gene name i matches the protein name of protein j ”. How is the case handled when a single protein is encoded by multiple genes?

Response:

Response to Question 1: Why is the base pair distance restricted within 150 kb?

We thank the reviewer for this insightful question. We selected a genomic-distance threshold of *150 kb* to delimit candidate peak–gene associations for three main reasons:

1. **Empirical prior in single-cell multi-omics.** A 150 kb window is a well-established cutoff in analogous frameworks—for example, GLUE applies the same threshold to define candidate peak-gene interactions [19].
2. **Computational efficiency.** Restricting peak–gene pairs to within 150 kb reduces the number of edges in our cross-modality attention graph, speeding up computational time.
3. **Biological validity.** High-resolution chromatin-conformation studies show that most functional enhancer–promoter loops occur within roughly 120 kb of a transcription start site, with contact frequency declining sharply beyond this range. A 150 kb cutoff thus enriches for true cis-regulatory interactions while excluding distal contacts that are likely noise [20].

While this prior restricts connections within the 150 kb window, the attention mechanism in MultiGATE can detect long-range enhancer-promoter interactions, even those exceeding 500 kb, when supported by additional data: across eight distance bins (0–150 kb, 150–300 kb, . . . , > 1.25 Mb), MultiGATE can distinguish the brain-specific enhancer–promoter contacts from those in other tissues across 24 diverse human tissues. Details are provided below:

The cross-modality attention framework in MultiGATE places no hard limit on distance (150 kb) and can incorporate more distal links when supported by external data.

To demonstrate MultiGATE's ability to detect long-range interactions, we reanalyzed the adult human hippocampus Spatial ATAC-RNA-seq dataset. Under the 150 kb genomic distance prior alone, we obtained 16,347 short-range peak-gene candidates. We then augmented this prior by adding 205,627 enhancer-promoter contacts curated from HiChIP experiments database [21] across 24 diverse human tissues (Table 8). Specifically, any ATAC peak overlapping a HiChIP-annotated enhancer that loops to a gene promoter was introduced as a candidate edge in the peak-gene prior network, regardless of genomic distance.

After retraining MultiGATE with these augmented connections, we stratified all candidate edges into eight distance bins (0–150 kb, 150–300 kb, . . . , >1.25 Mb) and compared the attention scores assigned to brain-specific versus non-brain HiChIP loops in each bin. In every distance category, brain-specific HiChIP loops received significantly higher attention scores than other loops (Mann–Whitney U test, $P < 1 \times 10^{-3}$; Fig. R42).

Notably, the enhancer chr1:7669258-7670119 and its target gene CAMTA1, located 883.9 kb apart, were identified by MultiGATE with an attention score of 0.2194 and is supported by the eQTL pair chr1_7669531_G_C - CAMTA1. CAMTA1, a transcription factor enriched in the hippocampus, is crucial for Purkinje cell survival and cerebellar function, with its loss leading to ataxia, motor deficits, and neurodegeneration due to dysregulated neuronal gene expression [22]. Additionally, MultiGATE identified another enhancer at chr5:150000116-150001000 and its target gene CSNK1A1, with a distance of 507.9 kb. This enhancer-gene pair is supported by the hippocampus eQTL chr5_150000595_C_T - CSNK1A1, along with a risk SNP (rs4705403, $p=1.00e-8$) associated with migraine in GWAS study. CK1 α plays a crucial role in the pathogenesis of Alzheimer's and Parkinson's diseases by regulating key proteins involved in neurodegeneration [23].

These observations demonstrate that MultiGATE can detect long-range interactions when supplied with appropriate priors.

Table 8: Number of peak–gene pairs per tissue (FitHiChIP, 5 kb resolution).

Tissue	Candidate peak-gene pairs
Aorta	5399
Blood	53084
Brain	10644
Breast	18694
Colon	765
Embryo	8958
Endometrioid endometrial tumor	51
Endometrium	11
Esophagus	16026
Eye	1062
Heart	8221
Kidney	3
Lung	14744
Lymph node	1163
Lymphocyte	28106
Muscle	1616
Ovary	1714
Prostate	227
Skin	26163
Stem cell	44
Stomach	5675
Thyroid	2666
Uterus	591

Response to Question 2: How is the case handled when a single protein is encoded by multiple genes?

We apologize for not making this explicit in the original manuscript. To clarify, we visualize the cross-modality feature-connectivity graph in SPOTS dataset (Fig. R45). In this graph, each protein is represented by a blue node, and each gene by a green node labeled with its gene symbol. It contains:

- **Single-gene encoded proteins** (e.g., CD19) have a one-to-one connection: the CD19 protein node connects only to the *Cd19* gene node.

- **Multi-subunit proteins** (e.g., CD3) connect to every gene that encodes CD3 protein: in Fig. R45, the CD3 protein node links to the *Cd3d*, *Cd3e*, *Cd3g*, and *Cd247* gene nodes, each with $A_{i,j} = 1$.

MultiGATE is capable of handling cases where multiple genes encode one protein. Accordingly, we have updated the manuscript to read:

" $A_{ij} = 1$ if and only if the gene i encodes the protein j or the subunits of protein j ."

In addition, MultiGATE can model gene-protein associations, which is added in the revised manuscript:

MultiGATE can model transcriptional regulation by incorporating gene-protein associations into the cross-modality attention mechanism. While the original framework only links each protein feature to the genes encoding the protein, we have generalized this mechanism so that any protein feature can be linked to a curated set of related genes—and vice versa—thereby capturing broader regulatory relationships. To illustrate this, we applied MultiGATE to the SPOTS dataset, which jointly profiles spatial RNA and protein. Besides the original links between proteins and the genes encoding the protein in the cross-modality feature connectivity graph, we curated a *CD3-related* gene set and added an edge between the CD3 protein and each gene in the *CD3-related* gene set. To test whether MultiGATE can accurately recover known associations, we also defined three negative-control gene sets (B cell-related, macrophage-related, and randomly selected). We added identical edges between CD3 and the genes in these sets into the cross-modality feature connectivity graph. We then compared the learned attention scores between CD3 and each gene across the four sets.

1. ***CD3-related* gene set:** CD3 is a T cell marker, We queried RAMP-DB [12] for all pathways containing any of the four CD3 chains (CD3 zeta (ζ), delta (δ), epsilon (ϵ), and gamma (γ)). From those pathways (TCR signaling; Modulators of TCR signaling and T cell activation; T-cell activation SARS-CoV-2; T-cell antigen receptor

(TCR) pathway during *Staphylococcus aureus* infection; T-cell receptor signaling pathway and other pathways), we randomly extracted 48 member genes to form our *CD3-related* set.

2. **Negative control gene sets:** We queried RAMP-DB [12] for two immune-related gene sets and one randomly selected gene set from all human protein-coding genes. Each set contains 16 genes (so that all 3 sets total 48 genes):

- **B cell-related gene set:** From four B cell-related pathways—B cell receptor signaling pathway; Extrafollicular and follicular B cell activation by SARS-CoV-2; FBXL10 enhancement of MAP/ERK signaling in diffuse large B-cell lymphoma; and Antigen activates B Cell Receptor (BCR) leading to generation of second messengers, we randomly selected 16 genes.
- **Macrophage-related gene set:** From six macrophage-related pathways—ER-Phagosome pathway; Induction of autophagy and toll-like receptor signaling pathways by graphene oxide; Toll-like receptor signaling pathway; Toll-like receptor signaling related to MyD88; Cytokine-cytokine receptor interaction; and NOD-like receptor signaling pathway, we randomly selected 16 genes.
- **Randomly selected gene set:** We randomly drew 16 genes from human protein-coding genes.

Each control set contains 16 genes, matching the total of 48 genes in the *CD3-related* set.

In our prior cross-modality feature-connectivity graph, besides the original connections between proteins and their encoding genes, we connected each gene in all four sets to the CD3 protein, and recorded the learned attention scores between these genes and the CD3 protein.

Figure R43: Distribution of attention scores between the CD3 protein and four groups of genes: (1) genes from the CD3-Related pathway, (2) genes from the B cell pathway, (3) genes from the macrophage pathway, and (4) randomly selected genes. The central box in each group represents the interquartile range (IQR), with the median indicated by the horizontal line. Whiskers extend to $1.5 \times$ IQR, and outliers are shown as individual points, and the yellow marker indicates the mean value.

To evaluate that MultiGATE can distinguish the genes related to the CD3 protein, we compare the attention-score distributions for four gene sets: the *CD3-related* genes show the highest median and mean attention scores, followed by B cell-related genes, macrophage-related genes, and random-selected genes (Fig. R43). This suggests that MultiGATE can learn CD3-related genes.

To further validate that high attention score reflects functional proximity in *CD3-related* gene set, we mapped each gene's position onto the KEGG T cell receptor signaling pathway

(hsa04660). We color-coded the ten genes with the highest attention score (red) and the ten with the lowest attention score (blue), revealing that genes with high attention scores are clustered near CD3 nodes, whereas genes with low attention scores tend to lie further away (Fig. R44). Quantitatively, the top ten have an average shortest-path distance of 2.1 to any CD3 chain, versus 3.5 for the bottom ten, suggesting that our attention scores reflect functional proximity in the biological pathway.

The above examples demonstrate that MultiGATE can learn and prioritize gene-protein associations, accurately distinguishing true CD3-related genes from negative controls.

Figure R44: Visualization of gene proximity in the T cell receptor signaling pathway. Red indicates the top ten genes with the highest attention scores with CD3 in the T cell receptor signaling pathway. The bottom ten genes, with the lowest attention scores relative to the CD3 family, are shown in blue.

Figure R45: Protein-gene encoding relationships in the SPOTS dataset.

Comment 9

Is there a weighting factor to balance the components in the loss function?

Response:

Thank you for this insightful question. In our current MultiGATE formulation, we did not introduce weight parameters between the reconstruction loss and the CLIP contrastive learning loss. We optimize

$$\mathcal{L} = \underbrace{\sum_{i=1}^N \|\mathbf{X}_{1i} - \hat{\mathbf{X}}_{1i}\|_2 + \sum_{i=1}^N \|\mathbf{X}_{2i} - \hat{\mathbf{X}}_{2i}\|_2}_{\text{reconstruction}} + \underbrace{\mathcal{L}_{\text{CLIP}}}_{\text{CLIP}}$$

$$- \frac{1}{2N} \sum_{i=1}^N \log \frac{\exp(\text{sim}(f_i, g_i))}{\sum_j \exp(\text{sim}(f_i, g_j))}$$

$$- \frac{1}{2N} \sum_{i=1}^N \log \frac{\exp(\text{sim}(g_i, f_i))}{\sum_j \exp(\text{sim}(g_i, f_j))}$$

Empirically, this unweighted formulation performs well across all benchmark datasets and a simulated cancer-patterned spatial ATAC+RNA dataset. Accordingly, we adopted the unweighted sum to ensure ease of use.

Comment 10

It would be helpful to mention the computational time and memory usage in the paper.

Response: Thank you for the comment.

To assess the scalability of MultiGATE, we evaluated its runtime on the spatial multi-omics datasets with varying sizes and complexities (Table 9). Training time increases with the number of spots and the dimensions of the features (genes, peaks, or protein markers). Nonetheless, our results demonstrate that even for larger datasets, the runtime remains feasible on standard high-performance hardware.

Hardware Configuration:

- **GPU:** 1 × Tesla V100-PCIE-32GB
- **CPU:** Intel Xeon Gold 6254 @ 3.10GHz (20 cores)
- **Operating System:** CentOS Linux 7 (Core)

Manuscript revisions

We have updated the detailed runtime and memory usage metrics in Supplementary Note S17 (see also lines 477–478 in the main text).

Table 9: Runtime and GPU Memory Usage for MultiGATE on Spatial Multi-Omics Datasets

Dataset	Dimensions	Memory Usage	Avg. Training Time
Adult Human Hippocampus (Spatial ATAC-RNA-seq [4])	RNA: 2500 pixels \times 7666 genes ATAC: 2500 pixels \times 28270 peaks	8726 MB	5.5 minutes
P22 Mouse Brain (Spatial ATAC-RNA-seq [4])	RNA: 9215 pixels \times 16252 genes ATAC: 9215 pixels \times 120400 peaks	31108 MB	1 hr 50 min
Mouse Spleen (Spatial RNA + ADT) (SPOTS [24])	RNA: 2563 spatial barcodes \times 14371 genes Protein: 2563 spatial barcodes \times 21 markers	1550 MB	2 minutes
Metastatic Melanoma (Spatial ATAC + RNA) (Slide-tags [25])	RNA: 2535 pixels \times 14807 genes ATAC: 2535 pixels \times 13665 peaks	4630 MB	8.5 minutes
Mouse Brain (Spatial RNA + metabolomics) (SMA [26])	RNA: 2820 spots \times 1538 genes Metabolomics: 2820 spots \times 1538 m/z's	1550 MB	1 minute

References

- [1] 10xGenomics, *Visium cytassist gene and protein expression library of human breast cancer, if, 6.5mm (ffpe). cytassist spatial gene and protein expression dataset analyzed using space ranger 2.1.0*, <https://www.10xgenomics.com/resources/datasets/gene-and-protein-expression-library-of-human-breast-cancer-cytassist-ffpe-2-standard>, 2023.
- [2] T. K. Koo and M. Y. Li, “A guideline of selecting and reporting intraclass correlation coefficients for reliability research,” *Journal of chiropractic medicine*, vol. 15, no. 2, pp. 155–163, 2016.
- [3] K. Coleman, A. Schroeder, M. Loth, *et al.*, “Resolving tissue complexity by multimodal spatial omics modeling with miso,” *Nature methods*, pp. 1–9, 2025.
- [4] D. Zhang, Y. Deng, P. Kukanja, *et al.*, “Spatial epigenome–transcriptome co-profiling of mammalian tissues,” *Nature*, vol. 616, no. 7955, pp. 113–122, 2023.
- [5] 10x Genomics, *Human Breast Cancer: Ductal Carcinoma In Situ, Invasive Carcinoma (FFPE) Spatial Gene Expression dataset*, Spatial Gene Expression dataset analyzed using Space Ranger 1.3.0, Licensed under the Creative Commons Attribution 4.0 International (CC BY 4.0) license, Jul. 2019.
- [6] A. Gayoso, Z. Steier, R. Lopez, *et al.*, “Joint probabilistic modeling of single-cell multi-omic data with totalvi,” *Nature methods*, vol. 18, no. 3, pp. 272–282, 2021.
- [7] T. Ashuach, M. I. Gabitto, R. V. Koodli, G.-A. Saldi, M. I. Jordan, and N. Yosef, “Multivi: Deep generative model for the integration of multimodal data,” *Nature Methods*, vol. 20, no. 8, pp. 1222–1231, 2023.
- [8] R. Argelaguet, D. Arnol, D. Bredikhin, *et al.*, “Mofa+: A statistical framework for comprehensive integration of multi-modal single-cell data,” *Genome biology*, vol. 21, pp. 1–17, 2020.

- [9] S. Heinz, C. Benner, N. Spann, *et al.*, “Simple combinations of lineage-determining transcription factors prime cis-regulatory elements required for macrophage and b cell identities,” *Molecular cell*, vol. 38, no. 4, pp. 576–589, 2010.
- [10] Z. Duren, X. Chen, J. Xin, Y. Wang, and W. H. Wong, “Time course regulatory analysis based on paired expression and chromatin accessibility data,” *Genome research*, vol. 30, no. 4, pp. 622–634, 2020.
- [11] R. Zheng, C. Wan, S. Mei, *et al.*, “Cistrome data browser: Expanded datasets and new tools for gene regulatory analysis,” *Nucleic acids research*, vol. 47, no. D1, pp. D729–D735, 2019.
- [12] J. Braisted, A. Patt, C. Tindall, *et al.*, “Ramp-db 2.0: A renovated knowledgebase for deriving biological and chemical insight from metabolites, proteins, and genes,” *Bioinformatics*, vol. 39, no. 1, btac726, Nov. 2022. DOI: 10.1093/bioinformatics/btac726.
- [13] L. Scrucca, M. Fop, T. B. Murphy, and A. E. Raftery, “mclust 5: Clustering, Classification and Density Estimation Using Gaussian Finite Mixture Models,” *The R Journal*, vol. 8, no. 1, pp. 289–317, 2016. DOI: 10.32614/RJ-2016-021.
- [14] K. Dong and S. Zhang, “Deciphering spatial domains from spatially resolved transcriptomics with an adaptive graph attention auto-encoder,” *Nature communications*, vol. 13, no. 1, p. 1739, 2022.
- [15] X. Shi, J. Zhu, Y. Long, and C. Liang, “Identifying spatial domains of spatially resolved transcriptomics via multi-view graph convolutional networks,” *Briefings in Bioinformatics*, vol. 24, no. 5, bbad278, 2023.
- [16] C. Xu, X. Jin, S. Wei, *et al.*, “Deepst: Identifying spatial domains in spatial transcriptomics by deep learning,” *Nucleic Acids Research*, vol. 50, no. 22, e131–e131, 2022.
- [17] H. Xu, H. Fu, Y. Long, *et al.*, “Unsupervised spatially embedded deep representation of spatial transcriptomics,” *Genome Medicine*, vol. 16, no. 1, p. 12, 2024.
- [18] M. Schott, D. León-Periñán, E. Splendiani, *et al.*, “Open-st: High-resolution spatial transcriptomics in 3d,” *Cell*, vol. 187, no. 15, pp. 3953–3972, 2024.

- [19] Z.-J. Cao and G. Gao, “Multi-omics single-cell data integration and regulatory inference with graph-linked embedding,” *Nature Biotechnology*, vol. 40, no. 10, pp. 1458–1466, 2022.
- [20] J. Dekker, M. A. Marti-Renom, and L. A. Mirny, “Exploring the three-dimensional organization of genomes: Interpreting chromatin interaction data,” *Nature Reviews Genetics*, vol. 14, no. 6, pp. 390–403, 2013.
- [21] W. Zeng, Q. Liu, Q. Yin, R. Jiang, and W. H. Wong, “Hichipdb: A comprehensive database of hichip regulatory interactions,” *Nucleic Acids Research*, vol. 51, no. D1, pp. D159–D166, Oct. 2022. DOI: 10.1093/nar/gkac859.
- [22] C. Long, C. E. Grueter, K. Song, *et al.*, “Ataxia and purkinje cell degeneration in mice lacking the camta1 transcription factor,” *Proceedings of the National Academy of Sciences*, vol. 111, no. 31, pp. 11 521–11 526, 2014.
- [23] S. Jiang, M. Zhang, J. Sun, and X. Yang, “Casein kinase 1 α : Biological mechanisms and theranostic potential,” *Cell Communication and Signaling*, vol. 16, pp. 1–24, 2018.
- [24] N. Ben-Chetrit, X. Niu, A. D. Swett, *et al.*, “Integration of whole transcriptome spatial profiling with protein markers,” *Nature Biotechnology*, vol. 41, no. 6, pp. 788–793, 2023.
- [25] A. J. C. Russell, J. A. Weir, N. M. Nadaf, *et al.*, “Slide-tags enables single-nucleus barcoding for multimodal spatial genomics,” *Nature*, vol. 625, no. 7993, pp. 101–109, Jan. 2024. DOI: 10.1038/s41586-023-06837-4.
- [26] M. Vicari, R. Mirzazadeh, A. Nilsson, *et al.*, “Spatial multimodal analysis of transcriptomes and metabolomes in tissues,” *Nature Biotechnology*, vol. 42, no. 7, pp. 1046–1050, 2024.